# A fast and efficient colocalization algorithm for identifying shared genetic risk factors across multiple traits

Christopher N. Foley [1,2✉], James R. Staley[2,3], Philip G. Breen[4], Benjamin B. Sun[2], Paul D. W. Kirk [1], Stephen Burgess [1,2] & Joanna M. M. Howson [2,5,6]

Genome-wide association studies (GWAS) have identified thousands of genomic regions affecting complex diseases. The next challenge is to elucidate the causal genes and mechanisms involved. One approach is to use statistical colocalization to assess shared genetic aetiology across multiple related traits (e.g. molecular traits, metabolic pathways and complex diseases) to identify causal pathways, prioritize causal variants and evaluate pleiotropy. We propose HyPrColoc (Hypothesis Prioritisation for multi-trait Colocalization), an efficient deterministic Bayesian algorithm using GWAS summary statistics that can detect colocalization across vast numbers of traits simultaneously (e.g. 100 traits can be jointly analysed in around 1 s). We perform a genome-wide multi-trait colocalization analysis of coronary heart disease (CHD) and fourteen related traits, identifying 43 regions in which CHD colocalized with ≥1 trait, including 5 previously unknown CHD loci. Across the 43 loci, we further integrate gene and protein expression quantitative trait loci to identify candidate causal genes.

[1] MRC Biostatistics Unit, Cambridge Institute of Public Health, University of Cambridge, Cambridge CB2 0SR, UK. [2] Cardiovascular Epidemiology Unit, Department of Public Health and Primary Care, University of Cambridge, Cambridge CB1 8RN, UK. [3] MRC Integrative Epidemiology Unit, Population Health Sciences, Bristol Medical School, University of Bristol, Bristol, UK. [4] School of Mathematics, University of Edinburgh, Kings Buildings, Edinburgh EH9 3JZ, UK. [5] National Institute for Health Research Cambridge Biomedical Research Centre, University of Cambridge and Cambridge University Hospitals, Cambridge, UK. [6] Department of Genetics, Novo Nordisk Research Centre Oxford, Oxford, UK. ✉email: chris.neal.foley@gmail.com

Genome-wide association studies (GWAS) have identified thousands of genomic loci associated with complex traits and diseases (https://www.ebi.ac.uk/gwas/). However, identification of the causal mechanisms underlying these associations and subsequent biological insights have not been as forthcoming, due to issues such as linkage disequilibrium (LD) and incomplete genomic coverage. One approach to aid biological insight following GWAS is to make use of functional data. For example, candidate causal genes can be proposed when the overlap in association signals between a complex trait and functional data (e.g. gene expression) is a consequence of both traits sharing a causal variant, i.e. the association signals for both traits colocalize. The abundance of significant associations identified by GWAS means that chance overlap between association signals for different traits is likely[1]. Consequently, overlap does not by itself allow us to identify causal variants[1,2]. Statistical colocalization methodologies seek to resolve this. By constructing a formal statistical model, colocalization approaches have been successful in identifying whether a molecular trait (e.g. gene expression) and a disease trait share a causal variant in a genomic region[3–7], and potentially prioritise a candidate causal gene. Recently it has been proposed that colocalization methodologies can be further enhanced by integrating additional information from multiple intermediate traits linked to disease, e.g. protein expression, metabolite levels[8]. The underlying hypothesis of multi-trait colocalization is that if a variant is associated with multiple related traits then this provides stronger evidence that the variant may be causal[8]. Thus, multi-trait colocalization aims to increase power to identify causal variants. We show that by using multi-level functional datasets in this way can reveal candidate causal genes and pathways underpinning complex disease.

A number of statistical methods have been developed to assess whether association signals across a pair of traits colocalize[3–7]. Some methods assess colocalization between a pair of traits using individual participant data[9,10], limiting their applicability. In contrast, the COLOC algorithm uses GWAS summary statistics[2]. This approach works by systematically exploring putative causal configurations, where each configuration locates a causal variant for one or both traits, under the assumption that there is at most one causal variant per trait. COLOC was recently extended to the multi-trait framework, MOLOC[8]. The authors achieved a 1.5-fold increase in candidate causal gene identification when a third relevant trait was included in colocalization analyses relative to results from two traits. However, the approach is computationally impractical beyond 4 traits due to prohibitive computational complexity arising from the exponential growth in the number of causal configurations that must be explored with each additional trait analysed.

Here we present a computationally efficient method, hypothesis prioritisation for multi-trait colocalization (HyPrColoc), to identify colocalized association signals using summary statistics on large numbers of traits. The approach extends the underlying methodology of COLOC and MOLOC. Our major result is that the posterior probability of colocalization at a single causal variant can be accurately approximated by enumerating only a small number of putative causal configurations. Moreover, HyPrColoc identifies subsets (which we refer to as clusters) of traits which colocalize at distinct causal variants in the genomic locus by employing a novel branch and bound divisive clustering algorithm. We show that the multi-trait clustering method of HyPrColoc has several performance advantages over alternative colocalization approaches and apply HyPrColoc genome-wide to coronary heart disease (CHD) and many related traits[11,12], identifying known and previously unknown candidate CHD genetic risk loci with colocalized associations across these traits.

## Results

**Overview**. HyPrColoc is a Bayesian method for identifying shared genetic associations between complex traits in a particular gene region using summary GWAS results. HyPrColoc provides two principal novelties: (i) Efficient computation of the posterior probability that all $m$ traits share a causal variant (which we refer to as the posterior probability of full colocalization, PPFC); and (ii) partitioning of traits into clusters, such that each cluster comprises traits sharing a causal variant. HyPrColoc only requires regression coefficients and their corresponding standard errors from summary GWAS (for binary traits these can be on the log-odds scale, see Methods). The approach makes three key assumptions: (i) for non-independent studies, that the GWAS results are from the same underlying population, i.e. that the LD pattern is the same across studies, (ii) that there is at most one causal variant in the genomic region for each trait (we assess limitations of this assumption when there are multiple causal variants below), and (iii) that these causal variants are either directly typed or well imputed in all of the GWAS datasets[2,8].

**Description of the HyPrColoc method**. We define a putative causal configuration matrix $S$ to be a binary $m \times Q$ matrix, where $m$ is the number of traits and $Q$ is the number of variants. To increase the probability of identifying any underlying causal variant(s) in the region, the number of SNPs $Q$ included in analyses should be maximised, i.e. the region should be well imputed. $S_{ij}$ is 1 if the $j^{th}$ variant is causal for the $i^{th}$ trait and 0 otherwise (Supplementary Information). A hypothesis uniquely identifies traits which share a causal variant, traits which have distinct causal variants and traits which do not have a causal variant. Except for the null hypothesis ($H_0$) of no causal variant for any trait, hypotheses such as $H_m$: all $m$ traits share a causal variant correspond to multiple configuration matrices, $S$ (Fig. 1). By considering the set of configurations to which a hypothesis corresponds, the posterior odds of the hypothesis against the null hypothesis can be computed. For example, let $\mathcal{S}_m$ denote the set of configurations for hypothesis $H_m$ and $S_0$ denote the single configuration for $H_0$, then the posterior odds for the hypothesis that all traits colocalize to a single causal variant is given by,

$$\frac{P(H_m|D)}{P(H_0|D)} = \sum_{S \in \mathcal{S}_m} \frac{P(D|S)}{P(D|S_0)} \times \frac{p(S)}{p(S_0)} \quad (1)$$

where $D$ represents the combined trait data, the first term in the summation is a Bayes factor and the second term is a prior odds[2,8]. To identify a candidate causal variant across the $m$ traits, i.e. to perform multi-trait fine-mapping, we locate the configuration $S^*$ satisfying $\max_{S \in \mathcal{S}_m} P(S|D) = P(S^*|D)$. If the summary data for the genetic associations between traits are independent, then the Bayes factor for each configuration $S$ can be computed by combining Wakefield's approximate Bayes factors[13] for each trait in the configuration ('Methods'). If the summary data between traits are correlated because a subset of the participant data was used in at least two of the GWAS analyses, then an extension to Wakefield's approximate Bayes factors, which jointly models the trait associations, can be employed ('Methods'). For a given hypothesis $H$ and set of corresponding configurations $\mathcal{S}_H$, the prior probability of configuration $S$, $p(S)$, can either be equal for all $S \in \mathcal{S}_H$, or can be defined as a product of variant-level priors ('Methods'). Our variant-level prior extends that of COLOC[2] and MOLOC[8] to a framework that is suitable for the analysis of large numbers of traits. We adopt an approach which requires the specification of a partition of the traits into clusters, together with two interpretable parameters: $p$, the probability that a variant is causal for one trait; and $p_c$, the conditional probability

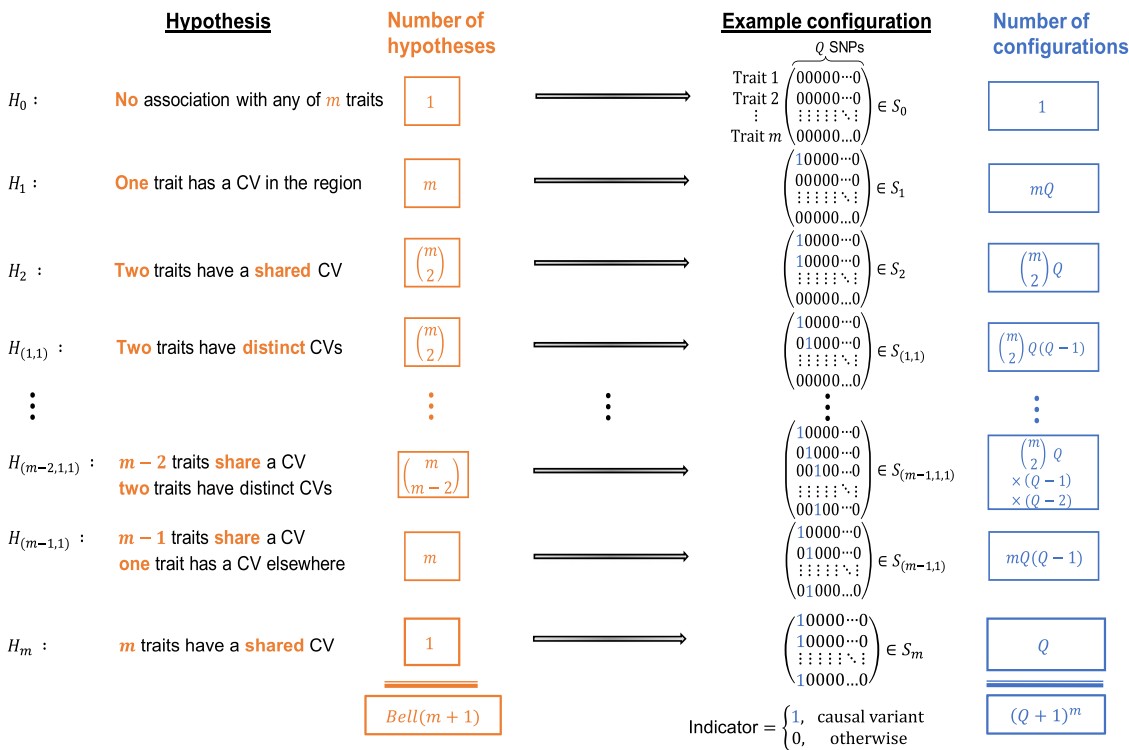

**Fig. 1 Colocalization hypotheses and causal configurations.** Statistical colocalization hypotheses and examples of their associated SNP configurations that allow for at most one causal variant for each of $m$ traits in a region containing $Q$ genetic variants. For clarity, the hypotheses and a single configuration associated with each hypothesis are shown for $m \geq 4$ traits, but the column totals $Bell\,(m+1)$ and $(Q+1)^m$ are correct for $m \geq 2$.

that a variant is causal for a second trait given it is causal for one trait ('Methods'). As it will be helpful later, we refer to $p_c$ as the conditional colocalization prior. COLOC[2] requires specification of three prior parameters $\{p_1, p_2, p_{12}\}$ and, while the scope of the configuration priors in HyPrColoc is different for more than a pair of traits, it is instructive to note that $p \equiv pi$, for $i \in \{1, 2\}$, and $p_c \equiv \left(\frac{p_{12}}{p_{12}+p_1}\right)$ when $m = 2$. To help users of the COLOC[2] software, our software allows users to specify the parameter $p$ and one of either (i) $p_c$; or (ii) $p_{12}$, from which $p_c$ is computed. For simplicity and as a conservative measure, we assume a priori that the genetic association probability $p$ and the conditional colocalization probability $p_c$ are equal for all traits. This approach allows sensitivity analyses assessing robustness of posterior inference to be routinely performed. However, it implicitly assumes traits are a priori exchangeable, e.g. assumes $p_1 = p_2$; this is supported across a range of designs (case/control or quantitative trait) but may lead to poorer performance in specific datasets[14].

**Efficient computation of the posterior probability of full colocalization (PPFC).** For a pre-specified genomic region comprising $Q$ variants, the aim is to evaluate the $PPFC$, $P(H_m|D)$, that all $m$ traits share a causal variant within that region, given the summarized data $D$. According to Bayes' rule, this is given by:

$$PPFC : P(H_m|D) = \frac{\sum_{S \in S_m} P(D|S) \times p(S)}{p(D)}. \quad (2)$$

Brute-force computation of the denominator, $p(D)$, requires the exhaustive enumeration of $(Q+1)^m$ causal configurations, which is computationally prohibitive for $m > 4$, e.g. MOLOC[8]. HyPrColoc overcomes this challenge by approximating $p(D)$ in a way that is both computationally efficient, i.e. has fast computational time, and tightly bounds the approximation error.

As we show in the 'Methods', the PPFC can be approximated as

$$\widehat{PPFC} = P_R P_A, \quad (3)$$

where $P_R$, $P_A > 0$ are rapidly computable values that quantify the probability that two criteria necessary for colocalization are satisfied (Fig. 2). The first of these criteria is that all the traits must share an association with one or more variants within the region. $P_R$, which we refer to as the regional association probability, is the probability that this criterion is satisfied. By itself, this criterion does not guarantee that there is a single causal variant shared by all traits, because it could be the case that two or more traits have distinct causal variants in strong LD with one another. To safeguard against this, we have a second criterion that ensures the shared associations between all traits are owing to a single shared putative causal variant. $P_A$ is the probability that this second criterion is satisfied. We refer to $P_A$ as the alignment probability as it quantifies the probability of alignment at a single causal variant between the shared associations. Both $P_R$ and $P_A$ have linear computational cost in the number of traits $m$, making a calculation of $\widehat{PPFC}$ possible when analysing vast numbers of traits. If the first criterion is satisfied, but the second is not, this may be because it is possible to partition the traits into clusters, such that each cluster has a distinct causal variant. HyPrColoc additionally seeks to identify these clusters.

**Identification of clusters of colocalized traits.** If $\widehat{PPFC}$ falls below a threshold value, $\tau$, we reject the hypothesis $H_m$ that all $m$ traits colocalize to a shared causal variant. In practice, this threshold is specified by defining separate thresholds, $P_R^*$ and $P_A^*$, for $P_R$ and $P_A$, such that $\tau = P_R^* P_A^*$ ('Methods'). If $H_m$ is rejected, HyPrColoc seeks to determine if there are values $\ell < m$ such that $H_\ell$ cannot be rejected; i.e. if there exist subsets of the

## Visualisation of colocalization criteria

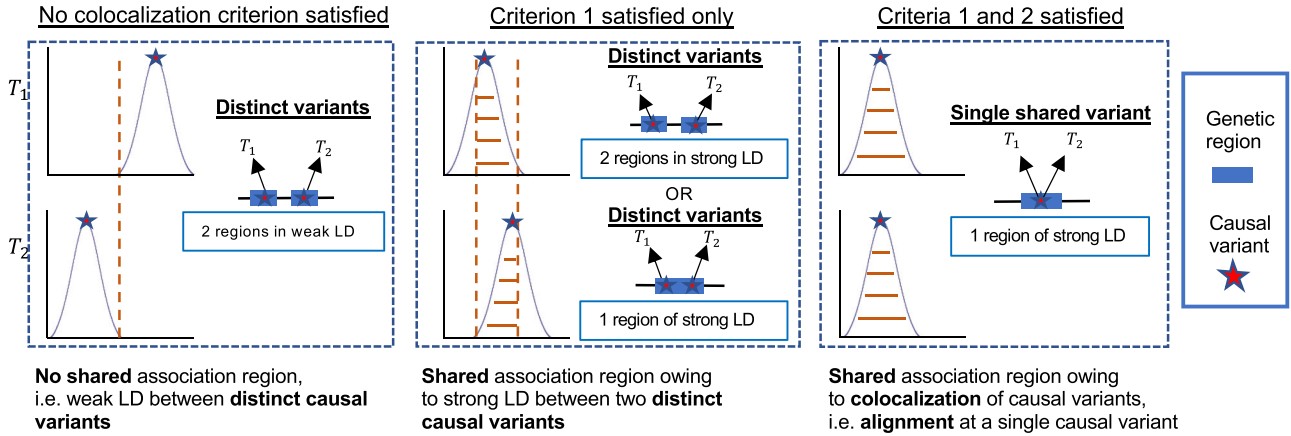

**No shared** association region, i.e. weak LD between **distinct causal variants**

**Shared** association region owing to strong LD between two **distinct causal variants**

**Shared** association region owing to **colocalization** of causal variants, i.e. **alignment** at a single causal variant

## Outline of the main *HyPrColoc* approximation

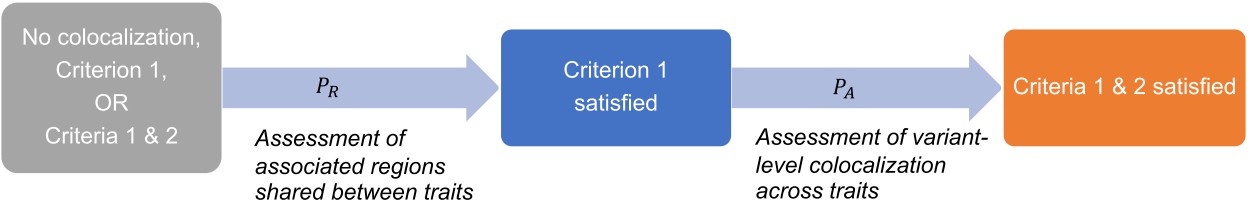

**Fig. 2 Illustration of the HyPrColoc approximation.** We illustrate the HyPrColoc approach with $m = 2$ traits. Statistical colocalization between traits which do not share an association region, i.e. do not have shared genetic predictors, is not possible (no colocalization criteria satisfied). However, traits which do (satisfying criterion 1) possess the possibility. HyPrColoc first assesses evidence supporting all $m$ traits sharing an association region, which quickly identifies utility in a colocalization framework. HyPrColoc then assesses whether any shared association region is due to colocalization between the traits (criteria 1 and 2) or due to a region of strong LD between two distinct causal variants, one for each trait (criterion 1 only). Results from these two calculations are combined to accurately approximate the *PPFC*.

traits such that all traits within the same subset colocalize to a shared causal variant. Starting with a single cluster containing all $m$ traits, our branch and bound divisive clustering algorithm (Supplementary Figs. S1a, b) iteratively partitions the traits into larger numbers of clusters, stopping the process of partitioning a cluster of two or more traits when all traits in a cluster satisfy both $P_R > P_R^*$ and $P_A > P_A^*$. The process of partitioning a cluster into two smaller clusters is performed using one of two criteria: (i) regional ($P_R$) or (ii) alignment ($P_A$) selection ('Methods' and Supplementary Note). For $k \leq m$ traits in a cluster, the regional selection criterion has $\mathcal{O}(kQ)$ computational cost and is computed from a collection of hypotheses that assume not all traits in a cluster colocalize because one of the traits does not have a causal variant in the region. The alignment selection criterion has $\mathcal{O}(kQ^2)$ computational cost and is computed from hypotheses that assume not all traits in a cluster colocalize because one of the traits has a causal variant elsewhere in the region (Supplementary Note). By default, the HyPrColoc software uses the more computationally efficient regional selection criterion to partition a cluster.

**Model validation using simulations**. We created simulated datasets by resampling phased haplotypes from the European samples in 1000 Genomes[15] and for each dataset we randomly selected one of the first 50 regions confirmed to be associated with CHD[16] ('Methods'). For each simulation scenario, 1000 replicates were performed.

**Computational efficiency**. The posterior probability of colocalization, across $m$ traits and in a region of $Q$ variants, can be accurately approximated by computing $\mathcal{O}(mQ^2)$ causal configurations. Figure 3 illustrates this for varying numbers of independent studies and variants, demonstrating a close linear relationship between computation time and the number of traits. Consequently, HyPrColoc is able to assess 100 traits, in a region of 1000 SNPs, in under 1 second compared to MOLOC which takes approximately 1 hour to analyse five traits. For $m \leq 4$, traits the median absolute relative difference between the HyPrColoc and MOLOC[8] posterior probabilities was found to be $\lessapprox 0.5\%$ (Fig. 3).

**Performance of HyPrColoc to detect multi-trait colocalization**. We used simulated datasets in which all traits colocalize to assess the accuracy of HyPrColoc in detecting colocalization across varying numbers of traits and study sample sizes. We simulated independent datasets with sample sizes of 5,000, 10,000, and 20,000 individuals for up to 100 quantitative traits and for which all traits share a single causal variant explaining either 0.5%, 1% or 2% of trait variance. For each simulated dataset, we used HyPrColoc to approximate the PPFC. The distribution of PPFC across the simulated datasets was narrower in the analysis of two traits relative to a larger number of traits, as the probability of random misalignment of the lead variant between traits increases as the number of traits increases (top Fig. 4). However, the estimated PPFC is always close to 1 for 5, 10 and 20 traits illustrating that the distribution of the estimate is stable across a broad

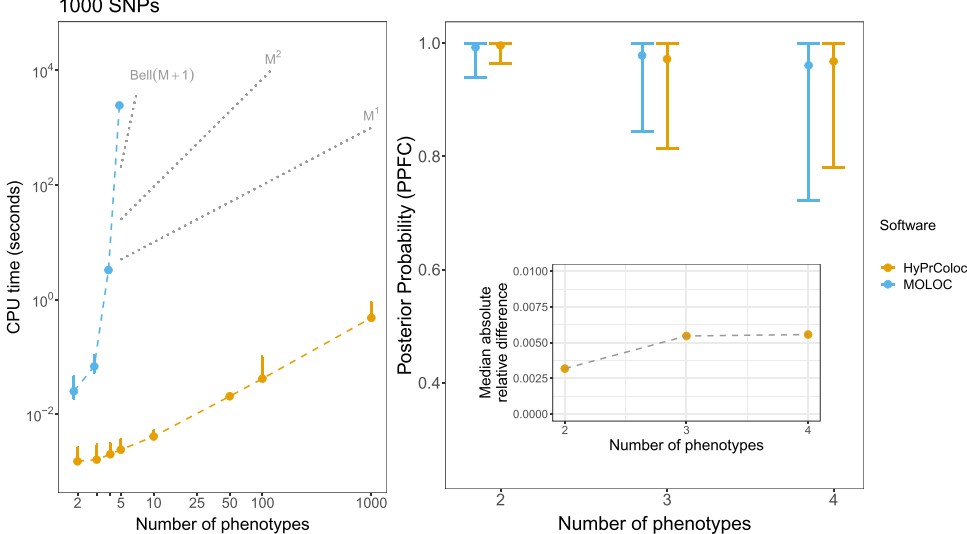

**Fig. 3 Comparison of HyPrColoc and MOLOC computation time and posterior probability of colocalization.** (Left panel) Computation time (seconds) for HyPrColoc (yellow) and MOLOC (blue) to assess full colocalization across $M \leq 1000$ traits in a region containing $Q = 1000$ SNPs. MOLOC was restricted to $M \leq 5$ traits owing to the computational and memory burden of the MOLOC algorithm when $M > 5$. When $M = 5$, we summarise the computation time of MOLOC from 10 datasets - as it took around 1 hour to analyse a single dataset, in all other scenarios performance was summarised from 1000 datasets. Three reference lines are plotted: (i) $Bell(M + 1)$, which denotes the theoretical cost of exhaustively enumerating all hypotheses; (ii) $M^2$, denoting quadratic cost and; (ii) $M^1$, denoting the linear complexity of the HyPrColoc algorithm. (Right panel) Distribution of the posterior probability of colocalization between all traits, i.e. the posterior probability of full colocalization (PPFC), using HyPrColoc (yellow) and MOLOC (blue) across $M \in \{2,3,4\}$ traits. Error bars denote the 1st and 9th deciles and a point denotes the median value. Despite differences in the prior set-up between the methods, the median absolute relative difference between the two posterior probabilities was $\lesssim 0.005$.

number of traits and sample sizes. For 100 traits there is a small decrease in power due to the growth in the number of hypotheses in which only a subset of the traits colocalize. This is expected when sample size is fixed and the shared causal variant explains only a small fraction of trait variation for each trait, as combined evidence supporting hypotheses in which a subset of the traits colocalize are eventually greater than evidence supporting full colocalization.

When at least one trait did not have a causal variant in the region the false detection rate was negligible. For example, we generated 100 quantitative traits, each from a study with sample size 10,000, in which 99 traits share a causal variant and the remaining trait had either: (i) a distinct causal variant or (ii) no causal variant in the region. In each scenario a causal variant explained 1% of trait variation. The 1st, 5th (median) and 9th deciles of the PPFC were $(4 \times 10^{-24}, 1 \times 10^{-17}, 5 \times 10^{-8})$ in scenario (i) and (0.02, 0.05, 0.10) in scenario (ii). There is a considerable difference between the results from each scenario, but the PPFC is below the threshold for declaring colocalization in both situations.

**Fine mapping the causal variant with HyPrColoc.** The proportion which HyPrColoc correctly identified the true causal variant increased as the number of colocalized traits included in the analyses increased up to 2-fold, irrespective of sample size and variance explained by the causal variant (middle Fig. 4), highlighting a major benefit of performing multi-trait fine-mapping. If HyPrColoc identified a variant that was not the true causal variant, we computed the LD between the true causal variant and the identified variant. In cases where the identified variant was not the causal variant, the variant was typically in very strong LD (median $r^2 \geq 0.99$) with the true causal variant and for large numbers of traits, i.e. $m \geq 20$, with sample size 20,000, the two variants were in perfect LD, i.e. $r^2 = 1$ (bottom Fig. 4).

**Branch and bound divisive clustering algorithm.** Here we assess the performance of the branch and bound (BB) divisive clustering algorithm to identify clusters of colocalized traits over a range of scenarios, several specifications of the conditional colocalization prior $p_c$ and using three classification criteria. The criteria were: accuracy, which is an overall measure of the classification of traits into clusters; true positive rate (TPR) and false positive rate (FPR), see Methods for more details. We simulated 10 traits from non-overlapping datasets under three scenarios: (i) a single cluster of 10 colocalized traits; (ii) 2 clusters of 3 colocalized traits, the remaining 4 traits do not have a causal variant (reflecting hypothesis free colocalization searches) and (iii) 4 clusters of colocalized traits, comprising 2 clusters of 3 traits and 2 clusters of 2 traits. Scenarios (ii) and (iii) are designed to simultaneously investigate potential false and true positive findings. Each cluster of colocalized traits share a single causal variant and causal variants between clusters are distinct, but can be in perfect LD, i.e. $r^2 = 1$, with one another—we assess results when the single causal variant assumption is violated later. To mirror real scenarios in which data are taken from studies with different sample sizes, we take the number of individuals in each study ($N_i$) as a random draw from the set $N_i \in \{1k, 5k, 10k, 15k, 20k\}$. For comparison, we additionally present results when all studies have a large sample size by also performing an analysis in which $N_i = 15k$ for all traits. In all scenarios, the causal variant for each trait explained 1% of trait variance and the probability parameters were set to $P_R^* = P_A^* = 0.5$ ('Methods'). Following the approach of Wallace[14], we assess sensitivity to the choice of colocalization prior $p_c$, i.e. $(1 - \gamma)$. Across a wide range of simulated data, Wallace[14] demonstrated that setting $p_{12} = 5 \times 10^{-6}$ in COLOC (approximately $p_c = 0.05$ in HyPrColoc) was generally a robust choice. Starting from this value, we evaluated results with more conservative choices of $p_c$ by performing three separate analyses for each dataset using $p_c \in \{0.05, 0.02, 0.01\}$, equivalent to $p_{12} \approx \{5 \times 10^{-6}, 2 \times 10^{-6}, 1 \times 10^{-6}\}$ with $p = 10^{-4}$ fixed[14], in order to

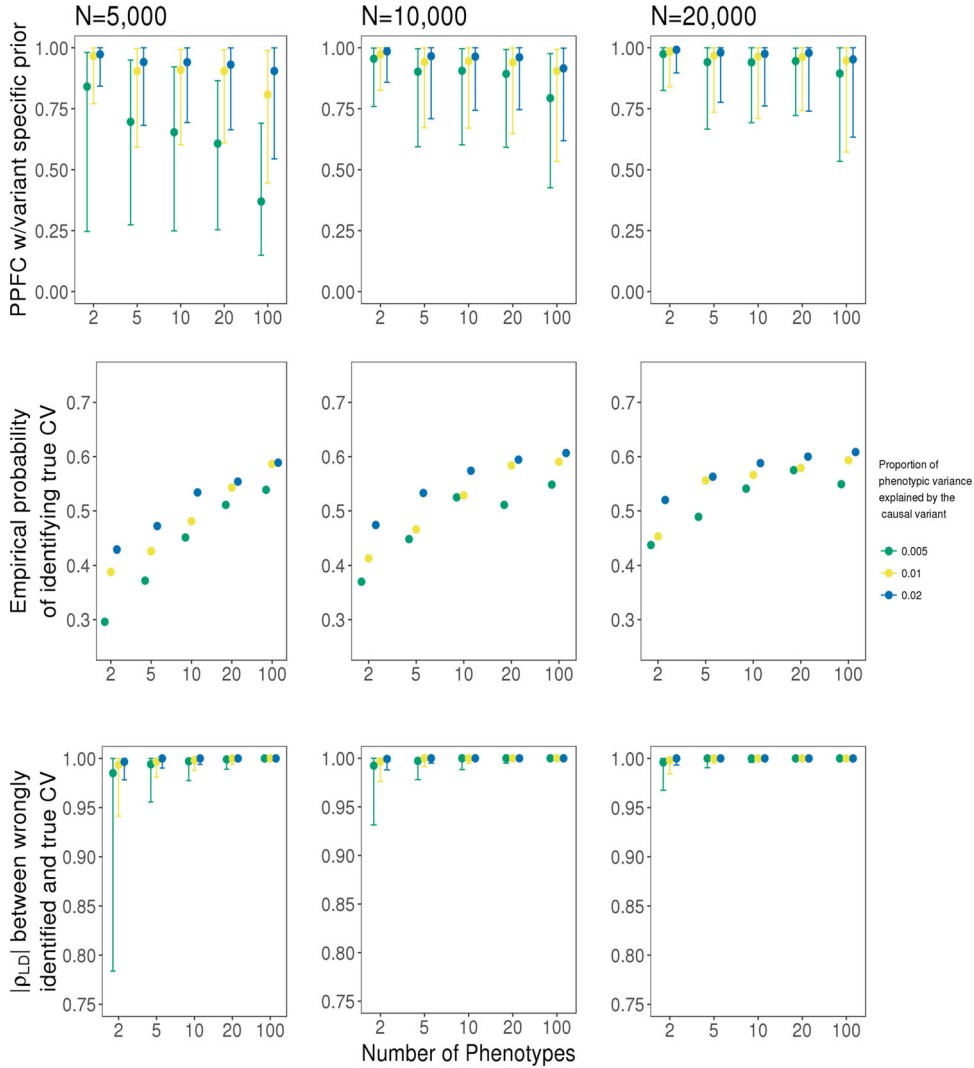

**Fig. 4 Assessment of the HyPrColoc posterior probability.** Simulation results for a sample size $N \in \{5,000, 10,000, 20,000\}$ and a causal variant explaining {0.5%, 1%, 2%} of variation across $m \in \{2, 5, 10, 20, 100\}$ traits. Presented is the distribution of the HyPrColoc posterior probability of full colocalization (PPFC) for variant-level priors only (top); the probability of correctly identifying the causal variant (middle) and; linkage disequilibrium between an incorrectly identified causal variant and the true causal variant (bottom). Error bars denote the 1st and 9th deciles and a point denotes the median value and performance was summarised from 1000 simulated datasets. Comparing performance across increasing study sample size and variance explained by the causal variant, power to detect all colocalized traits is reduced when including studies with smaller sample sizes (top row), however, including these studies can still boost the probability of correctly identifying the shared causal variant irrespective of variance explained (middle row).

identify a robust choice of $p_c$. These values can result in substantial differences in the prior probability of colocalization as the number of traits in a cluster increases ('Methods'). For comparison, we compare HyPrColoc against the alternative of performing pairwise colocalization analyses using COLOC[2], which restricts clusters' sizes to two traits only. Results are presented in Figs. 5, 6 and Supplementary Fig. S2.

We observed that both HyPrColoc and pairwise COLOC perform reasonably well across all three scenarios. The median accuracy and TPR is generally ≥0.75, for all three choices of $p_c$, improving to around 1 when the sample size of each study is large (Supplementary Fig. S2a, b; Supplementary Table S6)—indicating that including studies with smaller sample sizes decreases the TPR. Accuracy was more sensitive to the choice of $p_c$ when all traits colocalized into a single cluster, i.e. scenario (i), relative to scenarios (ii) and (iii) where we observed little sensitivity to $p_c$ (Supplementary Fig. S2a). We noted increased variability in the TPR when traits that do not have a causal variant were included in analyses, i.e. scenario (ii), particularly using the more stringent

colocalization prior $p_c = 0.01$ (Supplementary Fig. S2b). The FPR was generally low across all scenarios and prior choices: the 1st decile and median values were all zero. However, in scenario (iii), when there are 4 clusters of traits and 4 causal variants in the region, the 9th decile of the FPR increased for both methods, from around zero in scenario (ii) up to 0.16, 0.1 and 0.08 when $p_c$ was 0.05, 0.02 and 0.01, respectively (Supplementary Fig. S2c). The increase in FPR in scenario (iii) was a consequence of HyPrColoc occasionally wrongly including an extra trait in one of the clusters (Fig. 5b), and the pairwise approach overestimating the number of clusters (Supplementary Fig. S1c). This was because the causal variants from distinct clusters were in strong LD, i.e. $r^2 > 0.95$, the FPR of both methods reduced when excluding causal variants in strong LD (Supplementary Fig. S3). Over all scenarios, HyPrColoc regularly identified both the correct number of clusters of colocalized traits in the data (Fig. 5a) as well as the correct number of colocalized traits within each cluster (Fig. 5b). The pairwise approach resulted in more variation in the number of clusters identified (Supplementary Fig. S1c). HyPrColoc can

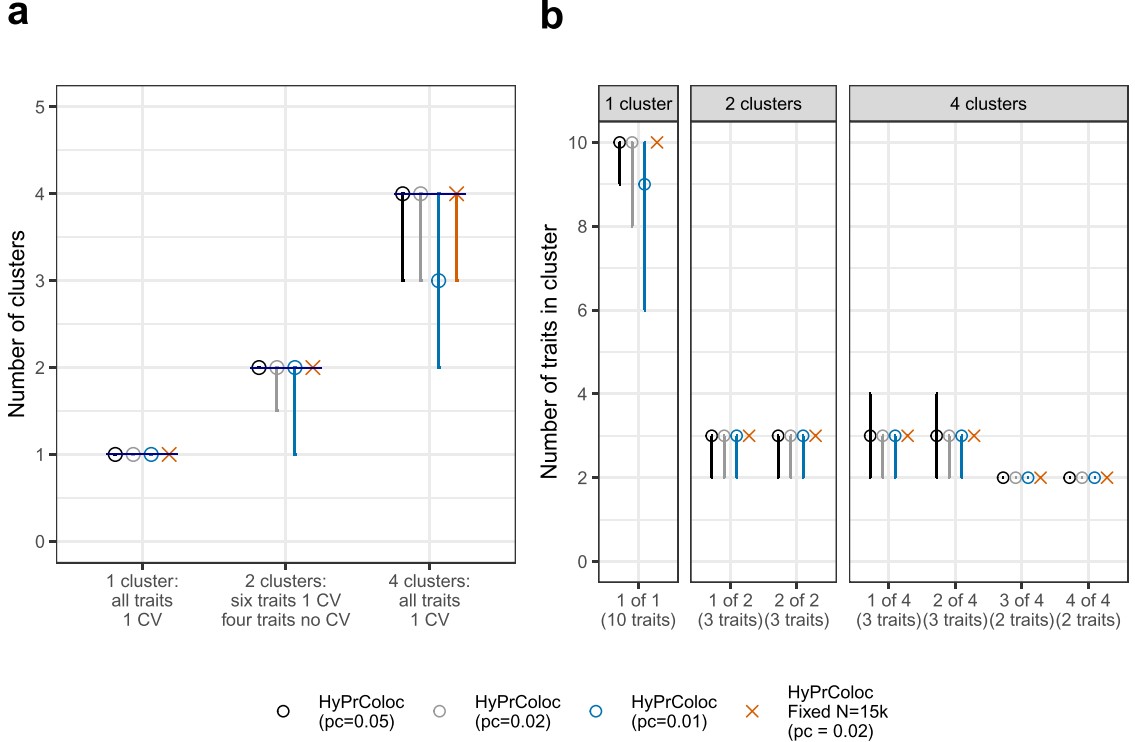

**Fig. 5 Number of clusters of colocalized traits and number of traits within a cluster.** Results from the single causal variant simulation study (c.f. Supplementary Fig. S2), presenting **a** the number of clusters of colocalized traits; and **b** the number of traits within each cluster identified by HyPrColoc. Error bars denote the 1st and 9th deciles and a point denotes the median value.

assign more than a pair of traits to a cluster, allowing information about the location of any shared causal variant to be borrowed across multiple traits, and therefore performed better at identifying the shared causal variant (Supplementary Fig. S2d). HyPrColoc significantly outperformed the pairwise approach when summarising results from the clusters of colocalized traits whose posterior probability satisfied $P_R P_A > 0.7$ (Fig. 6; Supplementary Table S7). This procedure reflects common practice, as colocalization results are generally only reported when the posterior probability of colocalization is greater than a threshold value, which we take here to be 0.7. Across all three scenarios, clusters of colocalized traits identified by HyPrColoc had a median accuracy and TPR of 1, with little sensitivity to the different choices of colocalization prior $p_c$. The FPR reduced also, for example in scenario (iii) when $p_c = 0.01$, the 1st, median and 9th deciles of the FPR were all zero. The FPR reduced for the pairwise approach after thresholding, but the TPR reduced as well. In pairwise approaches, a cluster of 3 or more colocalized traits is identified if and only if all pairs of traits colocalize (ideally at the same shared causal variant), the TPR of the pairwise method reduced after thresholding as only some of the pairs of traits passed the posterior threshold which increased the false negative rate. This is a drawback of methods which do not perform multi-trait colocalization. We repeated this simulation procedure for 20 traits and the results were similar (Supplementary Fig. S3B), highlighting the scalability of HyPrColoc to identify larger clusters of colocalized traits. Overall, across the range of scenarios considered the selection algorithm performed well in terms of sensitivity, specificity and accuracy. In many situations there will not be a strong prior belief in a single value for $p_c$. Based on our results and previous investigations[14], we recommend users set $p_c = 0.02$ and report results from the clusters of colocalized traits which satisfy $P_R P_A > 0.7$. Setting

$p_c = 0.02$ increased the number of datasets in which clusters satisfying $P_R P_A > 0.7$ were identified ('Methods') while maintaining a low FPR throughout. The HyPrColoc default $p_c = 0.02$ is equivalent to setting $p_{12} \approx 2 \times 10^{-6}$ which, for a pair of traits, is slightly more conservative than the recommended value of $p_{12} = 5 \times 10^{-6}$ by Wallace[14]. For more than a pair of traits, however, it can be much more conservative, e.g. setting $p_c = 0.05$ (i.e. $p_{12} \approx 5 \times 10^{-6}$) returns a prior probability of colocalization across 10 traits that is around 2000 times larger than when setting $p_c = 0.02$ (i.e. $p_{12} \approx 2 \times 10^{-6}$).

In scenarios (i), (ii) and (iii), HyPrColoc identified the clusters of colocalized traits on average 50, 30 and 25 times faster than the pairwise COLOC approach, indicating some sensitivity in computational performance to the type of colocalization structure present in the data. These figures improved to 200, 100 and 75 times faster when analysing 20 traits. The computational gains of HyPrColoc make it practical to perform multiple rounds of colocalization analyses, each with different values of the prior $p_c$ and the threshold parameters $P_R^*, P_A^*$, to assess any sensitivity in the clusters of colocalized traits identified to changes in parameter specifications. An example of this, taken from data generated under scenario (iii), is presented in Fig. 7a. The resulting heatmap highlights the presence of four clusters of colocalized traits in the data and these clusters persist across most of the prior and threshold parameter settings. We include this sensitivity analysis in the HyPrColoc software and recommend its use.

We further tested the algorithm using a variety of thresholds $\{P_R^*, P_A^*\}$ and two different prior frameworks (Supplementary Figs. S9–S10). We also assessed results in the presence of correlated traits and overlapping samples (Supplementary Information). We analysed these data in three ways: (a) ignoring all correlation, i.e. wrongly assuming non-overlapping participants between pairs of studies and ignoring known trait

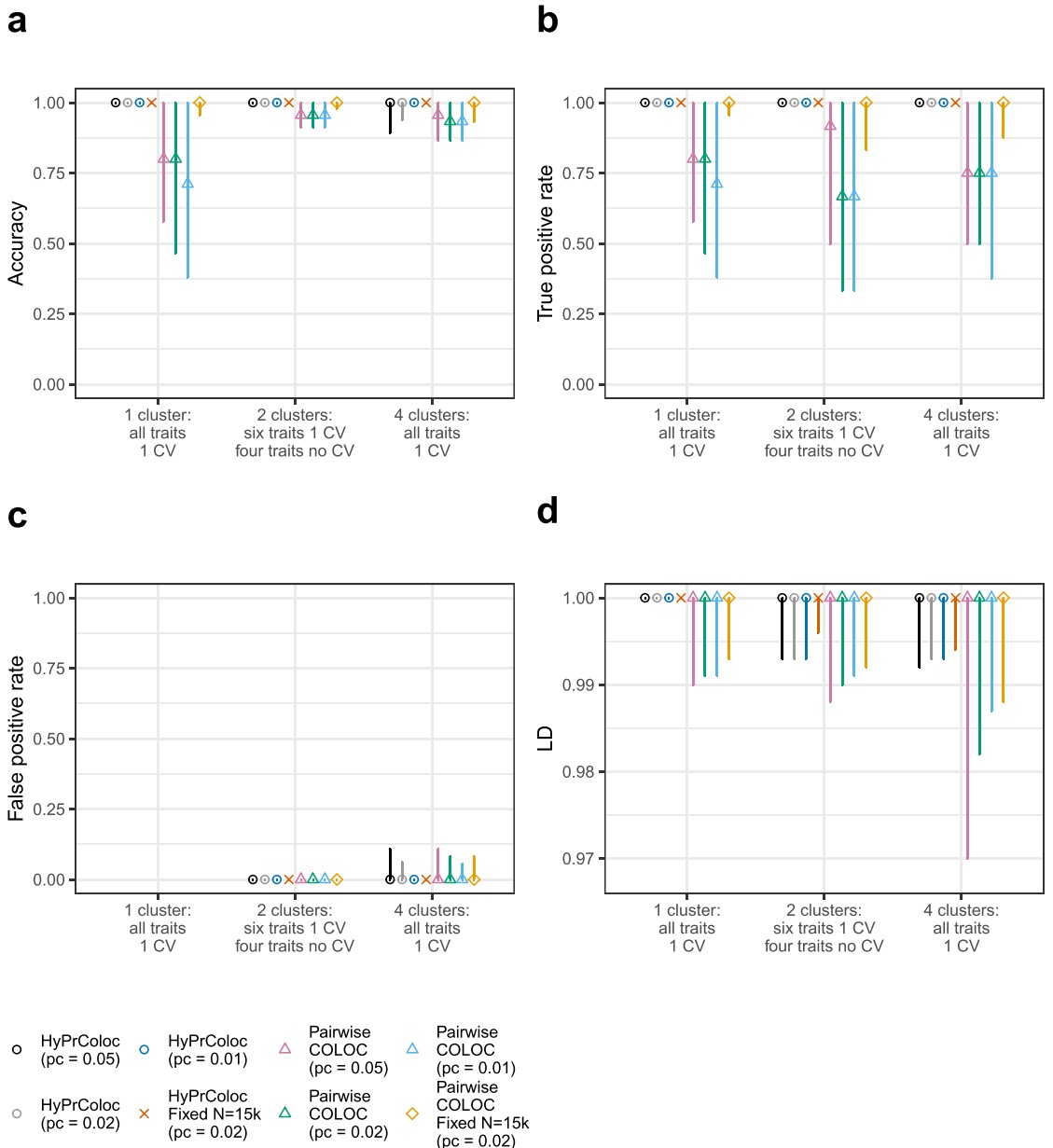

**Fig. 6 Performance of the BB clustering algorithm when excluding clusters of colocalized traits with lower posterior probability.** In each of the three scenarios presented, $m = 10$ traits with non-overlapping samples were generated, trait sample sizes were drawn randomly from the set $N = \{1,000, 5,000, 10,000, 15,000, 20,000\}$ and variant-level causal configuration priors were used with three choices of the colocalization prior $p_c \in \{0.05, 0.02, 0.01\}$. In scenario (i) there is one cluster of 10 colocalized traits; in scenario (ii) there are 2 clusters of colocalized traits, each comprising 3 traits, the remaining 4 traits do not have causal variants and; in scenario (iii) there are 4 clusters of colocalized traits, 2 clusters of 3 traits and 2 clusters of 2 traits sharing a causal variant. Traits within a cluster share a single causal variant and causal variants between clusters are distinct, however, a distinct variant can be in perfect LD, i.e. $r^2 = 1$, with another distinct variant. In all scenarios, we present results that passed the posterior probability of colocalization $P_R P_A \geq 0.7$. Presented are the classification measures: **a** accuracy; **b** true positive rate; and **c** the false positive rate. See 'Methods' for a description of how we define these in the context of clusters of colocalized traits. In **d** we present the LD between the identified causal variant for each cluster of colocalized traits and the true causal variant for each cluster. Error bars denote the 1st and 9th deciles and a point denotes the median value. The results highlight that on increasing the posterior threshold from 0.5 (c.f. Supplementary Fig. S2) to 0.7, HyPrColoc's ability to cluster multiple traits together demonstrably improves accuracy and the true positive rate relative to pairwise analyses.

correlation when setting the configuration prior probabilities; (b) adjusting for correlation between the summary data in the computation of the likelihood only; and (c) adjusting for correlation in the computation of the likelihood and accounting for known trait correlation when setting the configuration prior probabilities. Our findings suggest that analyses which account for correlation in the computation of the likelihood should also account for any known trait correlation in the configuration prior

probabilities: the posterior probability of colocalization between the truly colocalized traits in scenario (b), which ignored known correlation when setting the configuration prior, was significantly smaller than in scenario (c) - leading to a single large cluster of colocalized traits being split into smaller clusters (Supplementary Fig. S11 and Supplementary Table S2). Our results indicated that scenario (a), i.e. ignoring all correlation by treating studies as independent and traits as a-priori exchangeable, even when there

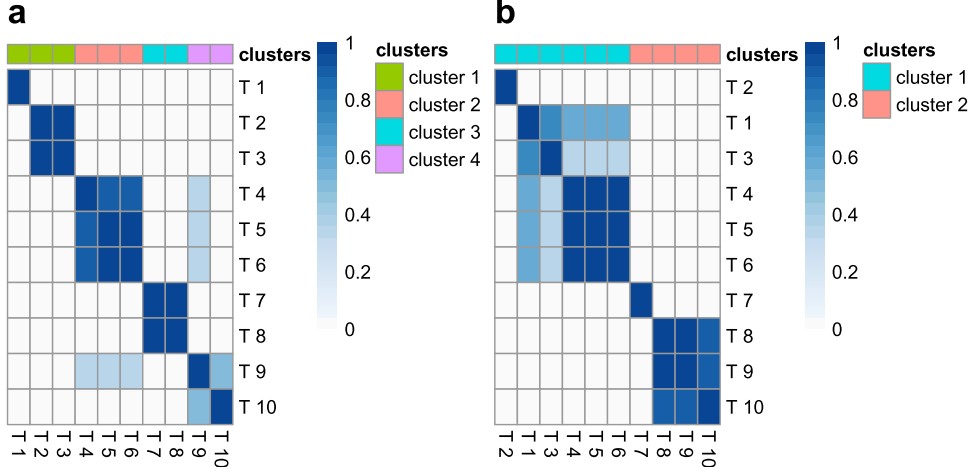

**Fig. 7 HyPrColoc's sensitivity analysis.** Heatmap visualizing changes in the clusters of colocalized traits identified by HyPrColoc when using different choices of the colocalization prior $p_c = \{0.05, 0.02, 0.01, 0.005\}$ and algorithm thresholds $P_R^* = P_A^* = \{0.5, 0.6, 0.7\}$. Cells appear darker when trait pairs cluster more often. Data were generated under scenario (iii) and when: **a** the single causal variant assumption is satisfied; or **b** the single causal variant assumption is violated.

is complete sample overlap (i.e. participants are the same in all studies), gives reasonable results and in our assessment was comparable to scenario (c) (Supplementary Fig. S10 and Tables S2–S3). We discuss the theoretical reasons for this in the Supplementary Information. We additionally provide an example analysis protocol in our online vignette, which accompanies our software (https://github.com/cnfoley/hyprcoloc), offering further guidance on the choice of prior configuration probabilities and assessing any sensitivity of the clusters of colocalized traits identified to the choice of prior parameters.

**Violations of the single causal variant assumption.** We assessed the performance of HyPrColoc when two or more traits have more than a single causal variant in the region. We simulated data for 10 traits and allowed up to 5 traits to have additional distinct causal variants in the region, so that the sample contains a mixture of traits which either satisfy or violate the single causal variant assumption. The data are generated under three scenarios, as previously, but now each cluster of colocalized traits share a single causal variant and half of the traits in a cluster have secondary distinct causal variants ('Methods'). In terms of marginal genetic associations, the additional variants were randomly selected to explain either slightly less trait variance than the shared causal variant ($\approx 0.75\%$) or the same amount of trait variance as the shared variant ($\approx 1\%$).

The median accuracy and TPR of HyPrColoc reduced by as much as 38% - in scenario (i) - and had greater variation between the 1st and 9th deciles when the single causal variant assumption was violated (Supplementary Figs. S4a,b); the reduction in performance was less pronounced when all studies had a large sample size. The FPR remained modest however, i.e. the 1st decile and median FPR were zero. A slight increase in the 9th decile of the FPR was noted when causal variants from distinct clusters were in strong LD, i.e. $r^2 > 0.95$, removing these reduced the FPR to zero (Supplementary Fig. S5c). For larger samples sizes, the 1st, median and 9th deciles of the FPR were approximately zero for each choice of prior (Supplementary Fig. S4c). When considering only the clusters of traits identified as colocalizing with $P_R P_A > 0.7$, HyPrColoc again provided very reliable results across all three classification measures (Supplementary Fig. S6a–c). Using the default settings $\{p = 10^{-4},$

$p_c = 0.02\}$, the algorithm generally performed well: in scenario (i) HyPrColoc regularly identified 8 of 10 traits as jointly colocalized; in scenario (ii) 5 out of 6 traits and; in scenario (iii) both clusters of colocalized traits, comprising 5 and 3 traits respectively (Supplementary Fig. S4f) - highlighting HyPrColoc is conservative when additional causal variants explain similar amounts of trait variation as the shared causal variant. We provide an illustration of HyPrColoc's sensitivity analysis tool under scenario (iii) (Fig. 7b) - correctly highlighting the presence of two clusters of colocalized traits. After applying more stringent prior and threshold values, one cluster reduced from 5 traits down to the 3 traits which have and share a single causal variant. This suggests strong evidence of 3 traits and weak evidence of 5 traits in the cluster. While the approach should be tailored to the problem at hand, if the analysis flags considerable sensitivity to the specification of the prior, we suggest: (a) reporting the clusters of colocalized traits identified as colocalizing with $P_R P_A > 0.7$ using the conservative prior setting $p_c = 0.02$; and (b) where computationally practical, running pairwise analyses using a multi causal variant method, e.g. eCAVIAR[5] or ENLOC[6], on the traits or clusters of traits which are reported in (a) but are not identified as colocalizing with $P_R P_A > 0.7$ using the more stringent prior $p_c = 0.01$ - this may help clarify if traits are being removed from clusters owing to the presence of additional non-shared causal variants, e.g. scenario (iii) (Fig. 7b), and should therefore be reported. We provide further guidance on the reliability of the BB algorithm when secondary causal variants are added to all traits in the region and when varying LD between causal variants (Supplementary Information; Supplementary Table S5).

We also compared results with those obtained using pairwise COLOC and eCAVIAR[5] (with a colocalization posterior probability, CLPP, cut-off of 1% and default prior choices), another software package for colocalization which allows each trait to have multiple causal variants but is limited to the analysis of pair of traits only. We note that the SNP level CLPP measure of eCAVIAR is computed in the presence of multiple causal variants and is distinct from the SNP level probabilities, computed under a single causal variant assumption, which comprise the posterior probability measure returned by HyPrColoc and COLOC - making comparisons between the methods challenging. We compare the methods as they are used in practice, summarizing HyPrColoc and COLOC using the posterior probability of the

hypothesis that a cluster or a pair of traits colocalize[2,8,14] and summarizing eCAVIAR using the SNP-level CLPP. Our choice of CLPP cut-off of 1% was shown to have a low FPR across a range of scenarios previously[5]. In our analyses we found that pairwise eCAVIAR had increased accuracy relative to HyPrColoc and pairwise COLOC, e.g. in scenario (i) median accuracy improved by as much as 0.15 (when sample sizes varied) and 0.2 (when sample sizes were large) (Supplementary Fig. S4a and Table S8). Broadly, this was a result of the single causal variant methods having a lower TPR (Supplementary Fig. S4a, b). However, by borrowing information between multiple traits HyPrColoc out-performed eCAVIAR when fine-mapping the shared causal variant (Supplementary Fig. S4d)—despite not incorporating LD information. After thresholding the posterior to $P_R P_A > 0.7$, HyPrColoc again outperformed pairwise COLOC (Supplementary Fig. S6a–c).

Despite violations of the single causal variant assumption, our analyses demonstrate that HyPrColoc can continue to identify clusters of colocalized traits, returning conservative results otherwise, with major computational advantages over competing software: in the analysis of 10 traits and in a region containing around 1000 SNPs, the single joint colocalization analysis of HyPrColoc was computed approximately 100,000 times faster than the 45 pairwise analyses of eCAVIAR. The HyPrColoc algorithm can additionally be used to rapidly identify genomic regions and clusters of traits to better prioritize the use of more computationally expensive multi-causal variant colocalization software for pairs of traits (Supplementary Information).

**Map of genetic risk shared across CHD and related traits**. We used HyPrColoc to investigate genetic associations shared across CHD[17] and 14 related traits: 12 CHD risk factors[18–22], a comorbidity[23] and a social factor[24] (Supplementary Table S1 for details). We performed colocalization analyses in pre-defined disjoint LD blocks spanning the entire genome[25]. To highlight that multi-trait colocalization analyses can aid discovery of new disease-associated loci, we used the CARDIoGRAMplusC4D 2015 data for CHD[17], which brought the total number of CHD-associated regions to 58, and contrasted our findings with the current total of ~160 CHD-associated regions[26]. For each region in which CHD and at least one related trait colocalized, we integrated whole blood gene expression[27] quantitative trait loci (eQTL) and protein expression[28] quantitative trait loci (pQTL) information into our analyses to prioritise candidate causal genes ('Methods').

**Multi-trait colocalization**. Our genome-wide analysis identified 43 regions in which CHD colocalized with one or more related traits (Fig. 8 and Tables 1–3). Twenty-three of the 43 colocalizations involved blood pressure, consistent with blood pressure being an important risk factor for CHD[29]. Other traits colocalizing with CHD across multiple genomic regions were cholesterol measures (16 regions); adiposity measures (9 regions); type 2 diabetes (T2D; 4 regions) and; rheumatoid arthritis (2 regions). Moreover, by colocalizing CHD and related traits, our analyses suggest these traits share some biological pathways.

In thirty-eight of the 43 (88%) colocalized regions, the candidate causal SNP proposed by HyPrColoc and/or its nearest gene, have been previously identified[16,17,26,30–35]. Importantly, 20 of these were reported after the CARDIoGRAMplusC4D study[17]. For example, *FGF5* was sub-genome-wide significant ($P > 5 \times 10^{-8}$) with CHD in the 2015 data, but through colocalization with blood pressure, we highlight it as a CHD locus and it is genome-wide significant in the most recent CHD GWAS[26]. The remaining 18 regions were reported previously, but one, *APOA1-C3-A4-A5*, was sub-genome-

wide significant in the CARDIoGRAMplusC4D study[17] despite having been reported previously[35]. However, we used HyPrColoc to show that the association of major lipids colocalize with a CHD signal, highlighting this as a CHD locus in these data (Table 1 and Supplementary Fig. S13). The locus has subsequently been replicated[26,31] and we show below that the signal also colocalizes with circulating apolipoprotein A-V protein levels (Table 1). This demonstrates that joint colocalization analyses of diseases and related traits can improve power to detect new associations (an approach which is advocated outside of colocalization studies[36]). Our results also illustrate that multi-trait colocalization analyses can provide further insights into well-known risk-loci of complex disease. For example, at the well-studied *SH2B3-ATXN2* region[26,35], HyPrColoc detected two cholesterol measures (LDL, HDL), two blood pressure measures (SBP, DBP) and rheumatoid arthritis (RA) colocalizing with CHD at the previously reported CHD-associated SNP[26] rs7137828 (PPFC = 0.909 of which 76.8% is explained by the variant rs7137828; Fig. 8). In addition, we implicated a candidate SNP and locus in a further 5 CHD regions not previously associated with CHD risk (Table 3). In one of the 5 regions, *CYP26A1*, CHD colocalized with tri-glycerides (TG) and HyPrColoc identified a single variant that explained over 75% of the posterior probability of colocalization, supporting this SNP as a candidate shared CHD/TG variant.

For each of the 43 regions that shared genetic associations across CHD and related traits, we further integrated whole blood gene[27] and protein[28] expression into the colocalization analyses. We tested *cis* eQTL for 1828 genes and *cis* pQTL from the 854 published proteins across the 43 loci for colocalization with CHD and the related traits. Of the 43 listed variants (Tables 1–3), 27 were associated with expression of at least one gene ($P < 5 \times 10^{-8}$) and a total of 125 such genes were identified. HyPrColoc refined this, identifying six regions colocalizing with eQTL for one expressed gene and one region, the *FHL3* locus, colocalizing with expression of three genes (*SF3A3*, *UTP11L*, *RNU6-510P*) (Table 2). The *GUCY1A3* locus has previously been associated with BP[37] and with CHD[16]. Here we show that these associations are likely to be due to the same variant, rs72689147 (PPFC = 0.93), with the G allele increasing DBP and risk of CHD. We furthermore show that the association colocalizes with expression of *GUCY1A1* in whole blood, with the G allele reducing *GUCY1A1* expression (PPFC = 0.77; Table 1). The *GUCY1A1* gene is ubiquitously expressed in heart tissues, including in the coronary and aortic arteries[38]. In the mouse, higher expression of *GUCY1A1* has been correlated with less atherosclerosis in the aorta[39]. *GUCY1A1* is a likely candidate gene in this locus[40], illustrating the utility of HyPrColoc to help prioritise candidate causal genes. The *CTRB2-BCAR1* locus was not known at the time of the release of the 2015 CARDIoGRAMplusC4D data, however we find the association at this locus is shared with T2D (PPFC = 0.83) and that *BCAR1* expression colocalized with the CHD association (PPFC = 0.86). Other studies have implicated the locus in CHD[34] and suggested *BCAR1* as the causal gene in carotid intimal thickening[41,42]. We note that two CHD loci also colocalize with circulating plasma proteins, *APOA1-C3-A4-A5*, with apolipoprotein A-V and the *APOE* locus with apolipoprotein E (Table 1).

Of the 38 known CHD loci that colocalized with a related trait, 8 are reported to have a single causal variant[26], of these we identified the same CHD-associated variant (or one in LD with either $r^2 > 0.8$ or $|D'| > 0.8$)[15] at seven loci (*SORT1*, *PHACTR1*, *ZC3HC1*, *CDKN2B-AS1*, *KCNE2*, *CDH13*, *APOE*). Despite the possible presence of multiple causal associations at other loci, HyPrColoc was still able to pick out single shared associations across traits: a result supported by our simulation study when additional distinct causal variants explain less or similar trait

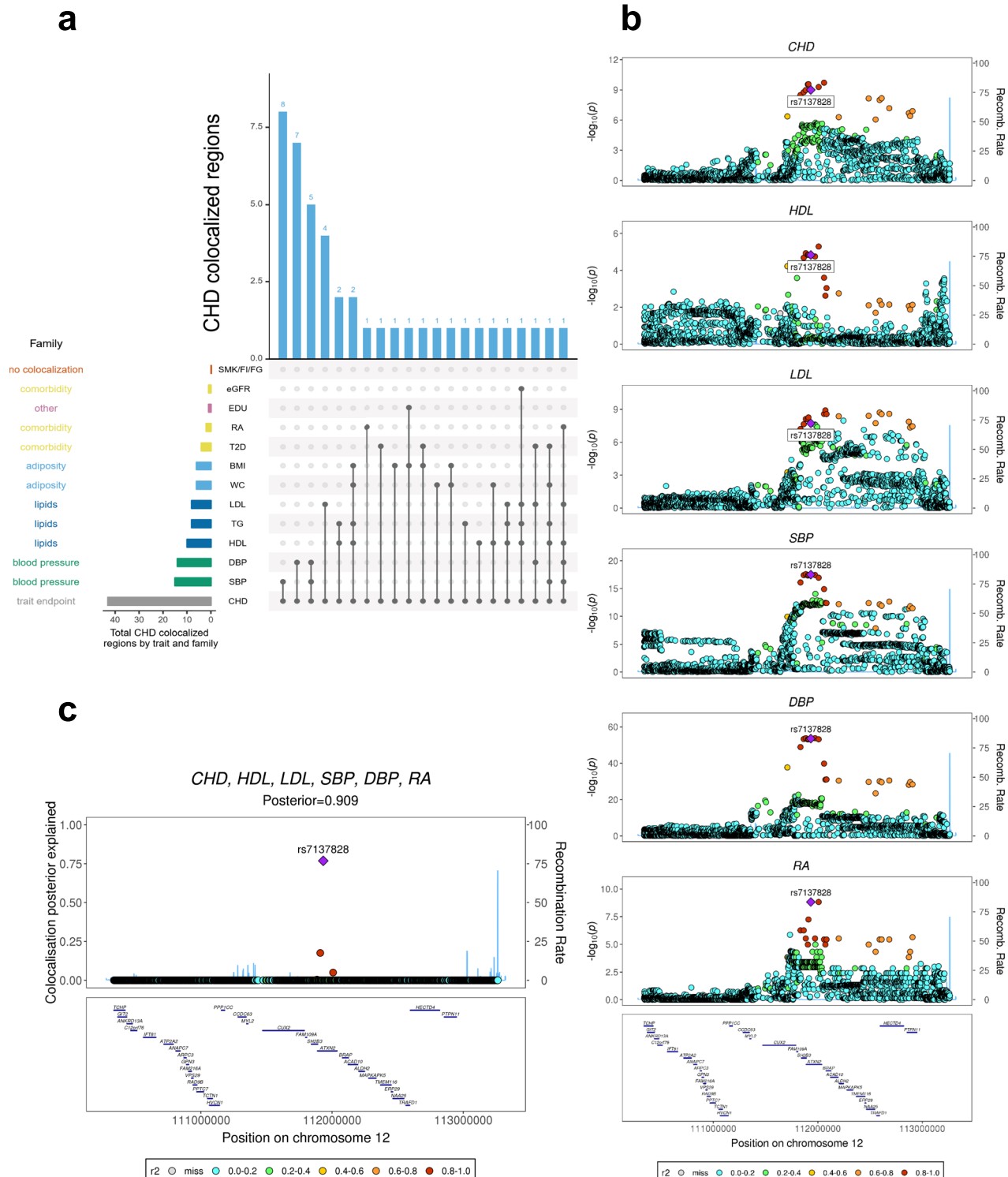

**Fig. 8 Genome-wide multi-trait colocalization analysis of CHD and fourteen related traits. a** Summary of the number of regions across the genome in which CHD colocalizes with at least one related trait. Results are aggregated by trait family, e.g. major lipids, and by each individual trait (see Supplementary Table S1 for a list of trait abbreviations). **b** Stacked association plots of CHD with high-density lipoprotein (HDL), low-density lipoprotein (LDL), systolic blood pressure (SBP), diastolic blood pressure (DBP) and rheumatoid arthritis (RA). HyPrColoc implicated both the *SH2B3-ATXN2* locus and risk variant rs713782, both of which have been previously reported as associated with CHD risk[26]. However, HyPrColoc extended this result by identifying that the risk loci and variant are shared with 5 conventional CHD risk factors[11]. SNPs in stronger LD with the putative causal SNP rs713782 appear darker in the plot. **c** HyPrColoc identified rs713782 as a candidate causal variant explaining the shared association signal between CHD and the 5 related traits. The posterior probability of colocalization between the traits was 0.909 and rs713782 explained over 76% of this, i.e. the posterior probability of rs713782 being the shared causal variant is $0.909 \times 0.76 = 0.69$. The next candidate variant explained <20%.

**Table 1 CHD loci that were known at the time of the CARDIoGRAMplusC4D data release (2015).**

| Chr | Locus | Traits | Colocalized SNP (consequence) | Gene | Known CHD locus (SNP) | PPFC (PPE) | Expressed gene (eQTL) | Protein (pQTL) |
|---|---|---|---|---|---|---|---|---|
| 2 | ABCG8, ABCG5 | CHD, LDL | rs4299376 (Intron) | ABCG8 | Yes[32] (Yes[32]) | 0.918 (0.949) | - | - |
| 4 | GUCY1A1 | CHD, DBP | rs72689147 (Intron) | GUCY1A1 | Yes[16] (Yes[17]) | 0.931 (0.241) | GUCY1A1 (rs12643599) | - |
| 6 | PHACTR1, EDN1 | CHD, SBP | rs9349379 (Intron) | PHACTR1 | Yes[33,35] (Yes[33]) | 0.999 (1) | - | - |
| 6 | LPA | CHD, LDL | rs10455872 (Intron) | LPA | Yes[32,35] (Yes[32,35]) | 0.998 (0.538) | - | - |
| 7 | HDAC9 | CHD, SBP | rs2107595 (Intergenic) | HDAC9 | Yes[16] (Yes[17]) | 0.996 (0.729) | - | - |
| 7 | ZC3HC1, KLHDC10 | CHD, DBP | rs11556924 (Missense) | ZC3HC1 | Yes[16,32,35] (Yes[16,32,35]) | 0.999 (0.994) | - | - |
| 8 | TRIB1 | CHD, HDL, LDL, TG, eGFR | rs2954029 (Intron) | RP11-136O12.2 | Yes[16] (Yes[16]) | 0.925 (0.872) | - | - |
| 9 | ANRIL, CDKN2B-AS1 | CHD, DBP | rs2891168 (Intron) | CDKN2B-AS1 | Yes[17] (Yes[17]) | 0.870 (0.755) | - | - |
| 9 | ABO | CHD, LDL, DBP, T2D | rs507666 (Intron) | ABO | Yes[16,35] (Yes[17]) | 0.984 (0.582) | - | - |
| 10 | KIAA1462 | CHD, DBP | rs1887318 (Intron) | KIAA1462 | Yes[16,33] (Yes[17]) | 0.937 (0.433) | - | - |
| 11 | APOA1-C3-A4-A5 | CHD, HDL, LDL, TG | rs964184 (3 prime UTR) | ZPR1, BUD13 | Yes[35] (Yes[35]) | 0.957 (1) | - | Apolipo-protein A-V (rs964184) |
| 12 | ATP2B1 | CHD, SBP | rs2681492 (Intron) | ATP2B1 | Yes[17] (Yes[17]) | 0.980 (0.303) | - | - |
| 12 | SH2B3 | CHD, HDL, LDL, SBP, DBP, RA | rs7137828 (Intron) | ATXN2 | Yes[35] (Yes[17]) | 0.909 (0.768) | TRAFD1 (rs7137828) | - |
| 15 | FES, FURIN | CHD, SBP, DBP | rs35346340 (Splice region) | FES | Yes[16] (Yes[17]) | 0.959 (0.579) | FES (rs8027450) | - |
| 18 | MC4R, PMAIP1 | CHD, HDL, TG, BMI, WC | rs12967135 (Intergenic) | - | Yes[17] (Yes[17]) | 0.859 (0.434) | - | - |
| 19 | LDLR, SMARCA4 | CHD, LDL | rs112374545 (Intergenic) | LDLR | Yes[16,35] (Yes[17]) | 0.937 (0.556) | - | - |
| 19 | APOC1, APOE, PVRL2, COTL1 | CHD, HDL, WC | rs4420638 (Downstream) | APOC1 | Yes[17] (Yes[17]) | 0.959 (0.999) | - | Apolipo-protein E (rs4420638) |
| 21 | KCNE2 | CHD, DBP | rs28451064 (Intron) | AP000318.2 | Yes[17] (Yes[17]) | 0.998 (0.974) | - | - |

HyPrColoc identified eighteen known CHD genetic risk loci (i.e. CHD loci reported before or at the time of the CARDIoGRAMplusC4D data release in 2015) with colocalized associations across CHD and one or more of 14 related traits. Chr: denotes chromosome; Locus: candidate causal gene(s) as listed by Erdmann et al.[40]. Traits: traits with colocalized association; Colocalized SNP(consequence): SNP marking association shared across the traits and its annotation in VEP[50] from PhenoScanner[51]. Gene: nearest gene to colocalized SNP; Known CHD locus: locus known at time of 2015 CHD data release[17] (i.e. published in ref. 17 or earlier) or subsequently identified[26]; PPFC: posterior probability of colocalization; PPE: proportion of PPFC explained by the listed SNP; eQTL: gene expression[27]; pQTL: protein expression[28]. See Supplementary Table S1 for a list of the trait abbreviations. Full results from these analyses are available at https://jrs95.shinyapps.io/hyprcoloc_chd.

**Table 2 CHD loci reported after the time of the CARDIoGRAMplusC4D data release (2015).**

| Chr | Locus | Traits | Colocalized SNP (consequence) | Gene | Known CHD locus (SNP) | PPFC (PPE) | Expressed gene (eQTL) | Protein (pQTL) |
|---|---|---|---|---|---|---|---|---|
| 1 | PRDM16 | CHD, SBP, DBP | rs2493288 (Intron) | PRDM16 | Yes[26] (Yes[26]) | 0.8009 (0.3471) | - | - |
| 1 | FHL3 | CHD, SBP | rs34655914 (Missense) | INPP5B | Yes[26] (Yes[26]) | 0.9468 (0.0832) | SF3A3 (rs28428561); UTP11L (rs4360494); RNU6-510P (rs61776719) | - |
| 1 | SORT1 | CHD, HDL | rs12740374 (3 prime UTR) | CELSR2 | Yes[26] (Yes[26]) | 0.9898 (0.9997) | - | - |
| 1 | LMOD1 | CHD, BMI, WC | rs2678204 (Intron) | IPO9 | Yes[30] (Yes[30]) | 0.8273 (0.1627) | IPO9 (rs2494115) | - |
| 2 | FIGN | CHD, SBP | rs268263 (Intron) | AC092684.1 | Yes[26] (Yes[26]) | 0.789 (0.995) | - | - |
| 2 | IRS1 | CHD, HDL, TG | rs62188784 (Intergenic) | AC068138.1 | Yes[26] (Yes[26]) | 0.8234 (0.4852) | - | - |
| 3 | RHOA | CHD, BMI, EDU | rs73078367 (Downstream) | NCKIPSD | Yes[26] (Yes[26]) | 0.9541 (0.5656) | - | - |
| 3 | RHOA | CHD, SBP | rs7623687 (Intron) | RHOA | Yes[34] (Yes[34]) | 0.9713 (0.2455) | - | - |
| 4 | FGF5, PRDM8 | CHD, SBP, DBP | rs13125101 (Intergenic) | FGF5 | Yes[26] (Yes[26]) | 0.9827 (0.4148) | - | - |
| 5 | MAP3K1 | CHD, HDL, TG, WC, SBP, T2D | rs9686661 (Intron) | C5orf67 | Yes[26] (Yes[26]) | 0.7755 (0.7115) | - | - |
| 6 | VEGFA | CHD, HDL, TG, BMI, WC | rs998584 (Downstream) | VEGFA | Yes[26] (Yes[26]) | 0.8376 (0.9746) | - | - |
| 10 | TSPAN14, FAM213A | CHD, RA | rs22343306 (Intron) | TSPAN14 | Yes[26] (No) | 0.9064 (0.7279) | - | - |
| 11 | ARNTL | CHD, DBP | rs10832013 (Upstream) | ARNTL | Yes[26] (Yes[26]) | 0.9403 (0.0823) | - | - |
| 11 | SIPA1 | CHD, HDL, TG | rs12801636 (Intron) | PCNX3 | Yes[30] (Yes[30]) | 0.8369 (0.8945) | - | - |
| 12 | HNF1A | CHD, LDL | rs1169288 (Missense) | HNF1A | Yes[30] (Yes[30]) | 0.9645 (0.5762) | - | - |
| 13 | N4BP2L2, PDS5B | CHD, BMI | rs35193668 (Intron) | N4BP2L2 | Yes[26] (Yes[26]) | 0.6785 (0.0911) | N4BP2L2 (rs9337) | - |
| 16 | CDHI3 | CHD, DBP | rs7500448 (Intron) | CDH13 | Yes[26] (Yes[26]) | 0.9947 (1) | - | - |
| 16 | CTRB2, BCAR1 | CHD, T2D | rs55993634 (Downstream) | CTRB2 | Yes[34] (Yes[26]) | 0.8296 (0.3868) | BCAR1 (rs28595463) | - |
| 17 | IGF2BP1 | CHD, BMI, T2D | rs11079849 (Intron) | IGF2BP1 | Yes[26] (Yes[26]) | 0.8389 (0.831) | - | - |
| 17 | PECAM1, DDX5, TEX2 | CHD, SBP, DBP | rs1867624 (Upstream) | RPL31P57 | Yes[30] (Yes[30]) | 0.7963 (0.4276) | - | - |

HyPrColoc identified twenty CHD genetic risk loci—reported after the time of the CARDIoGRAMplusC4D data release in 2015—with colocalized associations across CHD and one or more of 14 related traits. See Table 1 for a full description of the table items.

**Table 3 New CHD loci sharing colocalized associations with related traits.**

| Chr | Locus | Traits | Colocalized SNP (consequence) | Gene | Known CHD locus (SNP) | PPFC (PPE) | Expressed gene (eQTL) | Protein (pQTL) |
|---|---|---|---|---|---|---|---|---|
| 6 | *FHL5* | CHD, SBP | rs9486719 (Intron) | *FHL5* | – | 0.844 (0.1542) | – | – |
| 10 | *CYP26A1* | CHD, TG | rs2068888 (Downstream) | *CYP26A1* | – | 0.8454 (0.7669) | – | – |
| 16 | *ANKRD11* | CHD, WC | rs11643561 (Intron) | *ANKRD11* | – | 0.7827 (0.0795) | – | – |
| 19 | *RSPH6A* | CHD, SBP | rs8108474 (Intron) | *RSPH6A* | – | 0.7802 (0.1435) | – | – |
| 20 | *PREX1* | CHD, SBP, DBP | rs79044887 (Intron) | *PREX1* | – | 0.7237 (0.132) | – | – |

HyPrColoc identified five regions—not yet reported as CHD genetic risk loci—with colocalized associations across CHD and one or more related trait. See Table 1 for a full description of table items.

variation than that explained by a shared causal variant between colocalized traits (Supplementary Information). In our analyses, we set $p_c = 0.02$, i.e. $\gamma = 0.98$, and report only the clusters of traits whose posterior probability of colocalization was greater than 0.7. We assessed sensitivity to the choice of colocalization prior, repeating analyses with $p_c = 0.01$, and found no appreciable difference in the clusters identified (results not reported).

## Discussion

We have developed and applied a deterministic Bayesian colocalization algorithm, HyPrColoc, for multi-trait statistical colocalization analyses. HyPrColoc is based on the same underlying statistical model as COLOC[2], but enables colocalization analyses to be performed across massive numbers of traits, owing to the insight that the posterior probability of colocalization at a single causal variant can be accurately approximated by enumerating only a small number of putative causal configurations. HyPrColoc avoids repeated rounds of pairwise colocalization analyses which can inflate the false negative rate and have reduced performance in identifying a shared causal variant. The HyPrColoc algorithm was validated using simulations and used to assess genetic risk shared across CHD and related traits. Using CHD data from 2015[17], in which 46 regions were genome-wide significant ($P < 5 \times 10^{-8}$), our multi-trait colocalization analysis identified 43 regions in which CHD colocalized with ≥1 related trait. With this approach, we were able to identify CHD loci that were not known at the time of the data release (2015), demonstrating the benefit of synthesising data on related traits to uncover potential new disease-associated loci[8,36]. A further five regions, we postulate, may be identified as CHD loci in the future. Others have considered pleiotropic effects of CHD loci previously[43], but our formal colocalization analyses are more robust, e.g. in the *ABO* region we show colocalization of T2D and DBP in addition to the previously reported pleiotropic effect with LDL. We integrated eQTL and pQTL data to prioritise candidate genes at some loci, e.g. *GUCY1A1*, *BCAR1* and *APOE*.

The HyPrColoc algorithm identifies regions of the genome where there is evidence of a shared causal variant (by dissecting the genome into distinct regions) and also allows for a targeted analysis of a specific genomic locus of primary interest, e.g. when aiming to identify the perturbation of a biological pathway through the influence of a particular gene. Moreover, these region-specific analyses can highlight candidate causal genes, which will help improve biological understanding and may indicate potential drug targets to inform medicines development[44].

We have described HyPrColoc under the assumption of at most one causal variant per trait. Future work is required to extend this methodology and algorithm to multiple-causal variants. We note that the reliability of results under the single causal variant assumption only break down when secondary causal variants explain as much trait variation as the shared variant

(Supplementary Information). An example of which is the expression of *SH2B3*, where multiple causal variants for the expression of this gene masks colocalization with the CHD signal, we discuss an approach to building colocalization analyses which might help support the single causal variant assumption (Supplementary Information). We note that misspecification of LD between causal variants has a major impact on correct detection of multiple causal variants in a region[45], making a single causal variant assessment the most reliable when accurate study-level LD information is not available. To overcome challenges when specifying the prior probability of a causal configuration, we have suggested two different parsimonious configuration priors ('Methods'). The computational advantages of HyPrColoc make it practical to assess sensitivity of results to the specification of prior and threshold parameters as part of regular use. The HyPrColoc software includes a tool to do this, visualizing any changes in the clusters of colocalized traits identified as parameters are varied. Nevertheless, other priors may be more appropriate for particular applications.

In summary, we have developed a computationally efficient method that can perform multi-trait colocalization on a large scale. As the size and scale of available data on genetic associations with traits increase, computationally scalable methods such as HyPrColoc will be increasingly valuable in prioritizing causal genes and revealing causal pathways.

## Methods

**SNP association models.** Let $Y_i$ denote one of $i = 1,2,\ldots,m$, traits assessed in a maximum of $m$ studies, i.e. two or more traits can be measured in the same study, and $G_{ij}$ denote the genotype of the $j$th genetic variant. It is assumed that the outcome model for $Y_i$ is given by

$$\mathbb{E}\left[Y_i \middle| G_{ij}\right] = h_i^{-1}\left(\alpha_{ij} + \beta_{ij} G_{ij}\right), \tag{4}$$

where $\alpha_{ij}$ is the intercept term and $h_i$ is a function linking the $i^{th}$ outcome to the genotype $G_{ij}$, for all $j = 1,2,\ldots,Q$ genetic variants in the genomic region. The function $h_i$ is typically taken as the identity function for continuous traits and the logit function for binary traits. The aim of colocalization analyses is to identify genomic loci where there exists an $G_{ij}$ that is causally associated with at least two of the $m$ traits. For each of the $m$ traits and $Q$ genetic variants, we assume that GWAS summary statistics $\hat{\beta}_{ij}$ and $\text{var}\left(\hat{\beta}_{ij}\right)$ are available. We use these data to perform colocalization analyses in genomic loci.

**Colocalization posterior probability.** Using binary vectors to indicate whether a variant putatively causally influences a trait, we can define causal configurations ($S$) that can be grouped into sets ($\mathcal{S}_H$) which belong to a single data generating hypothesis ($H$). We use the notation $\mathcal{H}_{(i,j,\ldots)}$ to denote a *set* of hypotheses in which a collection of $i$ traits share a causal variant, a separate collection of $j$ traits share a distinct causal variant, and so on (Fig. 1). For, example, $\mathcal{H}_{(2,1)}$ denotes the set of hypotheses in which each hypothesis specifies uniquely 2 traits that share a causal variant, a single trait has a distinct causal variant and all remaining $m-3$ traits do not have a causal variant in the region. Assuming at most one causal variant for each trait these data generating hypotheses can be combined to generate a hypothesis space ($\Omega$). The posterior probability of hypothesis $H$, given the combined data $D$ from all $m$ studies, can therefore be computed using (Supplementary

Information),

$$P(H|D) = \frac{\sum_{S \in \mathcal{S}_H} BF(S)\frac{p(S)}{p(S_0)}}{\sum_{H_i \in \Omega} \sum_{S \in \mathcal{S}_{H_i}} BF(S)\frac{p(S)}{p(S_0)}}, \quad (5)$$

where $p(S)/p(S_0)$ is the prior-odds of configuration $S \in \mathcal{S}_H$ compared with the null-configuration $S_0$, i.e. no genetic association with any trait. See ref. [2] for a derivation with $m = 2$ traits. $BF(S)$ is a Bayes factor which is the likelihood of the data being generated under $S \in \mathcal{S}_H$ relative to the likelihood of the data being generated $S_0$.

We describe the space of multi-trait colocalization models using a set of mutually exclusive hypotheses and causal configurations as this approach extends the methodology and language used previously[2,8]. We note, however, that each causal configuration is equivalent to a model which, for each trait, details the location of the causal variant in the region. Hence, the problem of identifying a hypothesis and causal configuration with the greatest support given the data $D$, is equivalent to identifying the joint trait-variant model with greatest support[2,13].

**Computing Bayes factors: independent studies.** If the trait associations are calculated using independent studies (i.e. no overlapping samples in the GWAS datasets), the Bayes factors can be computed using Wakefield's Approximate Bayes Factors[13] (ABF) for each trait $i$ and genetic variant $j$, i.e.

$$ABF_{ij} = \sqrt{\frac{v_{ij}^2}{v_{ij}^2 + w_{ij}^2}} \exp\left(\frac{z_{ij}^2}{2} \times \frac{w_{ij}^2}{v_{ij}^2 + w_{ij}^2}\right), \quad (6)$$

where $z_{ij}$, $v_{ij}$ and $w_{ij}$ are the Z-statistic, standard error and the prior standard deviation for $\hat{\beta}_{ij}$, respectively. Following[2], for continuous variables $w_{ij}$ is set to 0.15 while for binary traits it is set to 0.2. As an example, the ABF for the hypothesis that all $m$ traits colocalize at genetic variant $j$ ($S_j \in \mathcal{S}_m$) is given by,

$$ABF(S_j) = \prod_i^m ABF_{ij}. \quad (7)$$

**Calculating Bayes factors: non-independent studies.** If the trait associations are not calculated using independent studies i.e. there are overlapping samples, the Bayes factor for each causal configuration can be computed using a Joint ABF (JABF) (Supplementary Information). The JABF for causal configuration $S$ is given by,

$$JABF(S) = \sqrt{\frac{\left|\Sigma_{\hat{\beta}}\right|}{\left|\Sigma_{\hat{\beta}} + \Sigma_{\beta}\right|}} \exp\left(\frac{1}{2}\hat{\beta}^T \left(\Sigma_{\hat{\beta}} + \Sigma_{\beta}\right)^{-1} \Sigma_{\beta} \Sigma_{\hat{\beta}}^{-1} \hat{\beta}\right), \quad (8)$$

where $\hat{\beta}$ is the vector of regression coefficients for all $m$ traits, $\Sigma_{\hat{\beta}}$ is an $m \times m$ variance-covariance matrix of the regression coefficients (i.e. $V\hat{\rho}V$, where $V^2$ is a diagonal matrix of variances for the regression coefficients, e.g. with $i$th diagonal element $v_i^2$, and $\hat{\rho}$ is the observed correlation matrix for the regression coefficients) and $\Sigma_{\beta}$ is the prior variance-covariance matrix (i.e. $W\rho W$, where $W^2$ is a diagonal matrix of prior variance, e.g. with $i$th diagonal element $w_i^2$, and $\rho$ is the prior correlation matrix between traits). The correlation matrix ($\hat{\rho}$) is computed using the tetrachoric correlation method[46] and we discuss our approach to setting $\rho$ in the Supplementary Information.

**Configuration prior probabilities.** We consider two different strategies for determining the priors for different hypotheses: variant-level priors and uniform priors.

**Variant-level prior probabilities.** The prior probability space for a single genetic variant can be fully partitioned into the prior probability that the genetic variant is not associated with any of the $m$ traits, $p_0$, the prior probability that the genetic variant is associated with only the first trait, $p_1$,..., the prior probability that the SNP is associated with a subset of $k$ traits $\{j_1, j_2, \ldots, j_k\}$, $p_{j_1 j_2 \ldots j_k}$, ..., the prior probability that the genetic variant is associated with all traits, $p_{12 \ldots m}$. Hence,

$$p_0 + \sum_{k=1}^m \left(\sum_{j_1=1}^m \sum_{j_2 > j_1} \cdots \sum_{j_k > j_{k-1}} p_{j_1 j_2 \ldots j_k}\right) = 1. \quad (9)$$

The space therefore requires the specification of $2^m$ prior parameters which, even for modest values of $m$, is computationally impractical. Following[2,8] we set that the prior probability to not vary by genetic variant, nor by the specific collection of colocalized traits of a given size, but by the number of colocalized traits, i.e. a SNP associated with a total of $k$ traits has a prior probability that depends on the number $k$ but not the specific collection of traits. To allow for the assessment of large numbers of traits we propose variant-level priors where the

prior probability that a genetic variant is associated with $k$ traits is given by,

$$p_{12 \ldots k} = p \prod_{i=2}^k \left(1 - \gamma^{i-1}\right), k = 2, \ldots m, \quad (10)$$

where $p$ is the probability of the genetic variant being associated with one trait and $\gamma$ is a parameter which controls the probability that a genetic variant is associated with an additional trait. Notably, $1 - \gamma$ is the probability of a variant being causal for a second trait given it is causal for one trait, i.e. it is the conditional colocalization prior $p_c$,

$$p_c = 1 - \gamma,$$

$1 - \gamma^2$ is the probability it is causal for a third trait given it is causal for two traits, and so on.

It follows that,

$$\frac{p(S)}{p(S_0)} = \frac{p_{12 \ldots k}}{p_0} = \frac{p}{p_0} \prod_{i=2}^k \left(1 - \gamma^{i-1}\right), k = 2, \ldots, m, \quad (11)$$

for configurations $S \in \mathcal{S}_{\mathcal{H}_k}$, where $k$ traits share a causal variant and the remaining $m - k$ traits do not have a casual variant, and

$$\frac{p(S)}{p(S_0)} = \frac{p_{12 \ldots (m-1)} p_1}{p_0^2} = \left(\frac{p}{p_0}\right)^2 \prod_{i=2}^{2m-1} \left(1 - \gamma^{i-1}\right), \quad (12)$$

for configurations $S \in \mathcal{S}_{\mathcal{H}_{(m-1,1)}}$, where $m - 1$ traits share a causal variant and the remaining trait has a distinct causal variant. This prior set-up allows evidence to grow in favour of $k$ traits colocalizing conditional on evidence supporting $k - 1$ traits colocalizing (Supplementary Information). For example, if the first $k$ traits are believed to share a causal variant a-priori, then the prior probability that the $(k+1)^{th}$ is also colocalized, conditional on the other $k$ traits, increases as the number of colocalized traits $k$ grows. The marginal prior probability of $k$ traits colocalizing is always very small, however, which controls the false positive rate (Fig. 6 and Supplementary Figs. S2–6; Supplementary Tables S2–3). Conditional growth limits the loss of power when assessing colocalization across a large number of traits. A loss in power necessarily occurs when analysing large numbers of colocalized traits, due to the rapid growth in the number of hypotheses in which a subset of traits can colocalize relative to all traits colocalizing. Evidence supporting these 'subset' hypotheses will eventually overwhelm evidence in favour of the maximum number of truly colocalized traits for a fixed sample size (top row Fig. 4). Based on our simulation results (Fig. 6 and Supplementary Figs. S2–6) and previous investigations[14], we recommend users set $p_c = 0.02$, i.e. $\gamma = 0.98$, and report results from the clusters of colocalized traits which satisfy $P_R P_A > 0.7$. Setting $p_c = 0.02$ increased the number of datasets in which clusters satisfying $P_R P_A > 0.7$ were identified (c.f. simulation study) while maintaining a low FPR throughout. Using the same posterior threshold of 0.7 and setting $p_c = 0.05$ returned reasonable results. However, we do not recommend users set $p_c = 0.05$ due to the slight increase in the 9th decile of the FPR in scenario (iii) (Fig. 6c). If two or more traits in a cluster are known to be related, this information would ideally be included in analyses and we outline an extension to our prior setup which allows for non-exchangeability of traits to be included (Supplementary Information).

**Conditionally uniform prior probabilities.** An alternative prior strategy is to assume uniform priors for each configuration within a hypothesis[47]. This strategy benefits from: (i) not setting variant-level information and (ii) implicitly accounting for large differences in the causal configuration space between hypotheses, which limits the loss in power of the PPFC for very large $m$. These priors take the form,

$$\frac{P(S|H)}{P(S_0|H_0)} = \frac{1/|\mathcal{S}_H|}{1/|\mathcal{S}_0|} = 1/|\mathcal{S}_H|, \quad (13)$$

where $\left|\mathcal{S}_{\mathcal{H}_k}\right| = Q$ and

$$\left|\mathcal{S}_{\mathcal{H}_{(m-1,1)}}\right| = \begin{cases} Q(Q-1) : m = 2, \\ mQ(Q-1) : m > 2. \end{cases} \quad (14)$$

Through simulations, we identified the conditionally uniform prior as less conservative than variant-level priors, having an increased false detection rate of colocalization. (Supplementary Information; Supplementary Figs. S10, 11). This could lead to an increased false positive detection rate in practice.

**HyPrColoc posterior approximation.** To compute the posterior probability of full colocalization across a large number of traits we propose the HyPrColoc posterior approximation. Let $P(H_m|D)$, $P_{scv}$, $P_{(m-1,1)}$ and $P_{all}$ denote: (i) the posterior probability of full colocalization; (ii) the sum of the posterior probabilities in which no traits have a causal variant, a subset of $m - 1$ traits share a causal variant (the remaining trait does not have a causal variant) and all $m$ traits colocalize ($P_{scv}$); (iii) the sum of posterior probabilities in which a subset of $m - 1$ traits share a causal variant and the remaining trait has a distinct causal variant ($P_{(m-1,1)}$) and; (iv) the sum of all posterior probabilities of at most one causal variant per

trait ($P_{all}$). That is,

$$P_{scv} = P(H_0|D) + P(\mathcal{H}_{m-1}|D) + P(H_m|D) \text{ and } P_{(m-1,1)} = P(\mathcal{H}_{(m-1,1)}|D). \quad (15)$$

The HyPrColoc posterior is computed in two steps. Step 1 computes the regional association probability $P_R$, defined as:

$$P_R = \frac{P(H_m|D)}{P_{scv}} \geq P(H_m|D). \quad (16)$$

Step 2 computes the alignment probability $P_A$, defined as:

$$P_A = \frac{P(H_m|D)}{P(H_m|D) + P_{(m-1,1)}} \geq P(H_m|D). \quad (17)$$

Note that $P_R$ is computed using $(m+1)Q$ causal configurations and $P_A$ is computed using an additional $mQ(Q-1)$ causal configurations. Hence, computation of $P_R$ and $P_A$ has $\mathcal{O}(mQ^2)$ computational cost. We let $P_{all}^c = P_{all} - P_{scv} - P_{(m-1,1)}$, then it follows that the posterior probability of all traits sharing a single causal variant is given by

$$P(H_m|D) = \frac{P(H_m|D)}{P_{all}} = \frac{P(H_m|D)}{P_{scv}} \frac{P_{scv}}{P_{all}} = \frac{P(H_m|D)}{P_{scv}} \times$$

$$\frac{\frac{P_{scv}}{P(H_m|D)} P(H_m|D)}{\frac{P_{scv}}{P(H_m|D)} \left( P(H_m|D) + P_{(m-1,1)} \right) - \frac{P_{scv}}{P(H_m|D)} \left( \left( 1 - \frac{P(H_m|D)}{P_{scv}} \right) P_{(m-1,1)} - \frac{P(H_m|D)}{P_{scv}} P_{all}^c \right)} \quad (18)$$

$$= \frac{P_R P_A}{1 - \left( (1-P_R)(1-P_A) - P_R(1-P_A) \frac{P_{all}^c}{P_{(m-1,1)}} \right)}$$

$$= P_R P_A + \mathcal{O}(\delta_A^2 + \delta_R \delta_A), \delta_R, \delta_A \to 0,$$

where $\delta_R = 1 - P_R$, $\delta_A = 1 - P_A$ and

$$\frac{P_{all}^c}{P_{(m-1,1)}} = \mathcal{O}(\delta_R + \delta_A),$$

(Supplementary Information). By definition, $P(H_m|D) \to 1 \Leftrightarrow P_R \to 1$ and $P_A \to 1$. Hence together the regional and alignment probabilities when multiplied form a statistic that is sufficient to accurately assess evidence of the full colocalization hypothesis. The objects $P_R$ and $P_A$ can be defined for various collections of hypotheses that partition $P_{all}$. However, the major insight is that the hypotheses contained in $P_R$ and $P_A$ are computed with minimal computation burden, i.e. computed using $\leq mQ^2$ causal configurations, amongst all alternatives, making the HyPrColoc approximation tractable for very large numbers of traits $m$.

Our software allows for the assessment of the HyPrColoc approximation by increasing the number of hypotheses used to approximate $P_R$, e.g. we can compute

$$P_R' = \frac{P(H_m|D)}{P(H_0|D) + P(\mathcal{H}_{m-2}|D) + P(\mathcal{H}_{m-1}|D) + P(H_m|D)}, \quad (19)$$

which is computed from $\mathcal{O}(m^2 Q)$ causal configurations and assess the relative difference between $P_R$ and $P_R'$. We show that $P_R' = P_R(1 + \delta_R)$ (Supplementary Information) and through simulations that there very close correspondence between $P_R'$ and $P_R$ (Supplementary Table S4).

**Branch and Bound divisive clustering algorithm.** To identify complex patterns of colocalization amongst all traits, we propose a branch and bound (BB) divisive clustering algorithm that utilizes the HyPrColoc approximation to identify a cluster of traits with the greatest evidence of colocalization at each iteration (Supplementary Fig. S1a and Supplementary Information). Starting with all of the traits in a single cluster, the algorithm explores evidence supporting any of 2$m$ branches - a branch represents a hypothesis whereby $m-1$ traits share a causal variant and either the remaining trait does not have a causal variant or has a causal variant elsewhere in the region - against the full colocalization hypothesis. These branches represent the hypotheses used in the computation of the regional and alignment probabilities $P_R$ and $P_A$. There are two bounds: (i) the minimum probability required to accept evidence that all $m$ traits are regionally associated $P_R^*$ and (ii) the minimum probability required to accept that the causal variant for all $m$ traits aligns at a single variant $P_A^*$. The BB algorithm accepts evidence supporting all $m$ traits sharing a single causal variant if $P_R P_A \geq P_R^* P_A^*$, after which the algorithm returns the HyPrColoc estimate of $PPFC$ and stops. If either $P_R < P_R^*$ or $P_A < P_A^*$ there is insufficient evidence supporting all traits sharing a causal variant and the BB algorithm moves to the branch with maximum evidence supporting $m-1$ traits sharing a causal variant. At this point the traits are partitioned into two clusters: one containing $m-1$ traits deemed most likely to share a causal variant and a second cluster containing the remaining trait. We repeat this process of branch selection and partitioning on the cluster of $m-1$ traits until we identify either: (A) a cluster of traits of size $k \geq 2$ whose regional and alignment statistics satisfy $P_R P_A \geq P_R^* P_A^*$, or (B) there is one trait left in the cluster. In scenario A, the HyPrColoc posterior probability that all $k$ traits colocalize is presented and the remaining $m-k$ traits are assessed for evidence of colocalization using the branch selection and partitioning scheme. In scenario B, the trait is deemed not colocalize with any other trait in the sample and the BB selection algorithm is repeated using $m-1$ traits. The entire process is repeated until all clusters of colocalized traits, whereby each cluster of traits colocalize at a distinct causal variant, have been

identified, all other traits are deemed not to share a causal variant with any other trait.

**Simulation study.** To create genomic loci with realistic patterns of LD, for each simulation scenario we simulated 1000 datasets and for each dataset we resampled phased haplotypes from the European samples in 1000 Genomes[15] and randomly chose one of the first 50 regions confirmed to be associated with CHD[16]. After removing variants with low MAF, i.e. MAF < 0.05, the number of SNPs analysed in these regions ranged from 228, in the APOE region, to a maximum of 1918 SNPs in the PDGFD region. The mean number of SNPs was 881.6. Unless stated otherwise, for traits that have a causal variant in the region, the variant explains 1% of trait variance. To go some way to mirroring real analyses, each trait was assumed to be measured in studies with different sample sizes, i.e. the sample size for the $i$-th study ($N_i$) was randomly chosen from the set $N_i \in \{1,000, 5,000, 10,000, 15,000, 20,000\}$. Variant-level priors were chosen for the simulation study: we set $p = 10^{-4}$ as in refs. [2,14] and, to assess sensitivity of results to the choice of conditional colocalization prior $p_c$, we ran each simulation three times for each of $p_c \in \{0.05, 0.02, 0.01\}$. Note that $p_c = 1 - \gamma$, so this is equivalent to $\gamma \in \{0.95, 0.98, 0.99\}$. For a pair of traits, colocalization between the traits is 5 times more likely a-priori when setting $p_c = 0.05$ relative to $p_c = 0.01$. In the analysis of ten traits, however, colocalization between all ten traits is around 1 million times more likely a-priori when setting $p_c = 0.05$ relative to $p_c = 0.01$. The prior probability of colocalization is still very small ~$10^{-11}$ when setting $p_c = 0.05$, however. Hence, the different values of $p_c$ we have chosen can result in substantial differences in the prior probability of colocalization.

**Violations of the single causal variant assumption.** These data were generated under three scenarios: (i) a single cluster of 10 colocalized traits, each trait shares a single causal variant and 5 traits have secondary distinct causal variants; (ii) a single cluster of 6 colocalized traits, each of the 6 traits share a single causal variant and 3 traits have secondary distinct causal variants, the remaining 4 traits do not have causal variants; and (iii) 2 clusters of colocalized traits, cluster 1 comprises 6 traits sharing a single causal variant with 3 of 6 traits having secondary distinct causal variants, cluster 2 comprises 4 traits sharing a single causal variant with 2 of 4 traits having secondary distinct causal variants. To maximize the number of traits with additional causal variants in a cluster (up to the maximum of 5), in scenarios (ii) and (iii) the total number of clusters of colocalized traits were reduced relative to the single causal variant assessment.

Measuring the accuracy, true positive and false positive rates of HyPrColoc

$$\text{Accuracy} = \frac{TP + TN}{TP + TN + FP + FN},$$

$$\text{True positive rate}(TPR) = \frac{TP}{TP + FN},$$

$$\text{False positive rate}(FPR) = \frac{FP}{FP + TN},$$

where TP and TN denote the true positive count and true negative count, and FP and FN denote the false positive count and false negative count. Hence, accuracy is the proportion of traits that are correctly identified as colocalizing or not colocalizing. To compare HyPrColoc with pairwise methods, we compute the TP, FP, TN and FN rates by aggregating information across all pairs of traits in the sample. A TP is measured when a pair of observations are correctly deemed to colocalize, a FP is measured when a pair of traits are incorrectly identified as colocalizing, a FN is recorded when a pair of traits are wrongly deemed not to colocalize and a TN is recorded when a pair of traits are correctly identified as not colocalizing.

When thresholding the posterior probability of colocalization, the TP, FP, TN and FN rates are computed after excluding traits which do not to colocalize with any other trait such that $P_A P_R > 0.7$. In the simulation study which allowed each trait a maximum of one causal variant in the region and with respect to scenarios (i), (ii) and (iii), when setting $p_c = 0.05$ HyPrColoc identified clusters of colocalized traits with $P_R P_A > 0.7$ in approximately 70%, 93% and 99% of simulated datasets, when $p_c = 0.02$ in approximately 65%, 91% and 98% datasets, reducing to around 60%, 86% and 97% of datasets when $p_c = 0.01$. Pairwise COLOC identified pairs of colocalized traits with $P_R P_A > 0.7$ in over 96% of simulated datasets, across all three scenarios and specifications of $p_c$. In the simulation study which allowed a maximum of two causal variants per trait, these figures reduced: when setting $p_c = 0.05$ HyPrColoc identified clusters of colocalized traits with $P_R P_A > 0.7$ in approximately 65%, 80% and 93% of simulated datasets, when $p_c = 0.02$ in ~60%, 72% and 92% datasets, reducing to around 55%, 65% and 85% of datasets when $p_c = 0.01$. Pairwise COLOC identified pairs of colocalized traits with $P_R P_A > 0.7$ in over 94% of simulated datasets, across all three scenarios and specifications of $p_c$

**Application to CHD and cardiovascular risk factors.** The GWAS results used in the assessment of colocalization of CHD with related traits were taken from large-scale analyses of CHD[17], blood pressure (http://www.nealelab.is/uk-biobank), adiposity measures (http://www.nealelab.is/uk-biobank), glycaemic traits[18], renal function[19], type II diabetes[20], lipid measurements[21], smoking[22], rheumatoid arthritis[23] and educational attainment[24] (Table S1). All datasets had either been imputed to 1000 Genomes[15] prior to GWAS analyses or were imputed up to 1000

Genomes from the summary results using DIST[48] (INFO > 0.8). We performed colocalization analyses in two steps. In step one, we assessed colocalization of CHD with the 14 risk-factors in pre-specified LD blocks from across the genome[25]. We used a conservative variant-level prior structure with $p = 1 \times 10^{-4}$ and $\gamma = 0.98$, i.e. 1 in 500,000 variants are expected to be causal for two traits, and set strong bounds for the regional and alignment probabilities, i.e. $P_R^* = P_A^* = 0.8$ so that the algorithm identified a cluster of colocalized traits only if $P_R P_A > 0.64$. The full results from this analysis are available at https://jrs95.shinyapps.io/hyprcoloc_chd.

To prioritise candidate causal genes in regions where CHD and at least one related trait colocalized, we re-ran the colocalization analysis and included whole blood cis eQTL[27] (31,684 samples) and cis pQTL[28] (3301 samples) data in addition to the primary traits in a second step, using the same LD blocks as before. A colocalization analysis was performed for every transcript with data within each region. cis eQTL were defined 1MB upstream and downstream of the centre of the gene probe (1828 genes were analysed across the 43 regions). cis pQTL were defined 5MB upstream and downstream of the transcript start site (854 proteins were analysed across the 43 regions). We integrated gene expression information taken from whole blood tissue as: (i) the eQTLGen dataset[27] has a large sample size relative to other publicly available gene expression data resources and; (ii) the pQTL data were also measured in whole blood tissues, so there was consistency in the tissue analysed.

**Reporting summary**. Further information on research design is available in the Nature Research Reporting Summary linked to this article.

## Data availability

The genome-wide association summary data that support the findings of this study are available from: CARDIoGRAMplusC4D (http://www.cardiogramplusc4d.org) for coronary heart disease; MAGIC (www.magicinvestigators.org) for glycaemic traits; GLGC (www.lipidgenetics.org) for lipid measures; TAG (https://www.med.unc.edu/pgc/download-results/tag/) for smoking; SSAGC (www.thessgac.org) for years in education; DIAGRAM (https://www.diagram-consortium.org) for type 2 diabetes; CKDGen (http://ckdgen.imbi.uni-freiburg.de/) for renal function measure eGFR; Okada et al. (http://plaza.umin.ac.jp/~yokada/datasource/software.htm) for rheumatoid arthritis; and the first release of the Neale Lab's GWAS analysis of UK-Biobank (http://www.nealelab.is/uk-biobank) for the adiposity measures and blood pressure traits. The summary data on gene expression and protein expression in whole blood are available from eQTLGen (http://www.eqtlgen.org/cis-eqtls.html) and Sun et al. (https://www.phpc.cam.ac.uk/ceu/proteins/), respectively. The LD information was computed using the phased haplotypes from the 1000 Genomes study (http://www.internationalgenome.org/). Full results from the genome-wide colocalization analysis of CHD and 14 related traits using HyPrColoc are available at https://jrs95.shinyapps.io/hyprcoloc_chd.

## Code availability

We developed an R package for performing the HyPrColoc[49] analyses (https://github.com/cnfoley/hyprcoloc). Please visit the HyPrColoc Zenodo page (https://doi.org/10.5281/zenodo.4293559) for information on how to cite the software. The regional association plots (as seen in Fig. 8) were created using gassocplot (https://github.com/jrs95/gassocplot) and LD information from 1000 Genomes[15]. We compared the performance of HyPrColoc with the publicly available software packages: COLOC (Version: 3.2-1; https://cran.r-project.org/web/packages/coloc/); eCAVIAR (Version: 2.0.0; https://github.com/fhormoz/caviar); and MOLOC (Version: 0.1.0; https://github.com/clagiamba/moloc).

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

## Acknowledgements

The authors would like to thank Professor Frank Dudbridge, University of Leicester, for valuable comments and suggestions, which greatly improved the manuscripts, and Dr. Robin Young, Robertson Centre for Biostatistics, University of Glasgow, for help with the simulation study. This work was funded by the UK Medical Research Council (MR/L003120/1, MC_UU_00002/13, MC UU 00002/7), British Heart Foundation (RG/13/13/30194), and the UK National Institute for Health Research Cambridge Biomedical Research Centre. JMMH was funded by the National Institute for Health Research [Cambridge Biomedical Research Centre at the Cambridge University Hospitals NHS Foundation Trust].

## Author contributions

C.N.F. developed the mathematical and statistical methodology, developed the statistical software and applied the methods to the analysis of CHD and related risk factors. J.R.S. advised on the statistical methodology and software, developed the bioinformatical software and command-line tool, designed and applied the methods to the analysis of CHD and related risk factors. P.G.B. contributed to the statistical methodology. B.B.S. designed the analysis of CHD and related risk-factors. P.D.W.K. and S.B. reviewed the statistical methodology and scientific content. J.M.M.H. conceived the project, contributed to the overall scientific content and goals of the project. All authors contributed to the writing of the manuscript.

## Competing interests

J.M.M.H. became a full-time employee of Novo Nordisk Ltd while this manuscript was under review. All other authors declare no competing interests.
