## [Peer Review File · Nature Communications]

Reviewers' comments:

Reviewer #1 (Remarks to the Author):

Foley et al. have proposed a method, HyPrColoc, to perform colocalization of multiple traits to detect causal variants. The main intuition behind HyPrColoc is that the causal variants shared between multiple traits are more likely to be true causal variants. Unfortunately, the method has fundamental problems as described below that lead to wrong results.

Main concerns.

1) HyPrColoc makes the assumption of one causal variant. This assumption is a very strong assumption and it is not true in real datasets (Wen et al. Plos Gene 2017, Hormozdiari et al. AJHG 2016, Jansen et al. Human Mol. Genetics 2017). It is shown that not modeling this phenomenon of more than one causal variant in a locus can confound the fine-mapping and colocalizing results.

2) Regarding point 1, how does HyPrColoc perform when we have more than causal variants. It is known that conditional method can detect the wrong set of causal variants.

3) I think the authors need to compare their method with a baseline model. In the baseline model, the authors will perform a fixed effect meta-analysis (absolute value of effect size) of all the traits and then perform a fine-mapping method to detect the causal variants. This baseline method should be extremely fast and produce better results than HyPrColoc.

4) The main drawback of coloc and HyPrColoc is that the LD information is never utilized.

5) How does the method perform compared to RTC (Nica et al. Plos Gene 2010)? I would guess that RTC to be extremely fast method.

Reviewer #2 (Remarks to the Author):

Foley et al. propose an analytic approach to perform colocalization analysis across multiple traits. I applaud the authors taking on a difficult yet crucial methodological challenge in statistical genetics. Unfortunately, I have a few serious concerns about the work presented in this manuscript. I hope the following comments help the authors to improve the manuscript.

1. The paper focuses on an approximation solution to compute the likelihood/Bayes factor from the data when the number of traits is large and possible scenarios grows exponentially. However, the other integral component of the Bayesian computation, i.e., the prior specification, does not receive sufficient attention and, in my opinion, oversimplified.

The authors describe two strategies in specifying the necessary priors for possible configurations. They are reasonable mathematical assumptions, but the practical implications on genetic applications do not seem to be sensible. For both strategies, the prior does not depend on the actual traits of interest and all the traits are assumed exchangeable. -- this can be dangerous. For example, consider an analysis dealing with both highly polygenic traits (e.g., molecular traits, heights) and the traits like type II diabetes. It is not reasonable to assume an equal prior of causal variants for all traits, because it implies that all traits have the same prior expected association signals. Unfortunately, both prior strategies proposed by the authors have such undesired property. In comparison, existing works like coloc define the prior property for each trait (i.e., p_1 , p_2) individually.

2. Because the proposed method is sensitive to prior specification and the approximation of the likelihood function/Bayes factor, the authors should provide a procedure/guideline to perform model checking and validation. The sensitivity of the colocalization results is well-documented in the literature when analyzing two traits. I can only imagine that the problem gets worse when extending the application to many more traits. I suggest that the authors i) examine and show the

sensitivity of the proposed approach; ii) propose a reasonable procedure to guide users in checking model/parameter assumptions.

3. In my opinion, the underlying statistical problem of colocalization is best described as a model selection problem not a hypothesis testing problem. Although using the hypothesis testing language is not wrong in describing the proposed procedure, in many places of the paper, the descriptions are heuristic and lack of rigor. Fundamentally, the underlying problem is about classification, that is, classify the alignment of the causal genetic variants into many possible classes. Thus, we should care about the overall measure of misclassification. I fail to see the proposed approach discussing this issue, or provide a general measurement of misclassification in the simulation studies. Hypothesis testing, on the other hand, focuses on controlling one specific type of classification errors and does not seem appropriate in the context of dealing with multiple traits.

4. Like coloc, the proposed approach assumes at most one causal variant per trait in a genomic locus of interest. I don't disagree with such an assumption, but feel that the authors should provide details on defining proper regions for analysis. This is because if the region defined contains more than one causal variant for a specific trait, the math shown by the authors will break down and give erroneous answers. It seems that some fine-mapping analysis should be carried out to ensure the assumption indeed holds.

5. The simulation studies need to be better designed and reported. The calibration of the resulting colocalization probabilities needs to be examined. The general classification error rates should be reported. The robustness of the proposed method should also be addressed.

Reviewer #1 (Remarks to the Author):

A0. Foley et al. have proposed a method, HyPrColoc, to perform colocalization of multiple traits to detect causal variants. The main intuition behind HyPrColoc is that the causal variants shared between multiple traits are more likely to be true causal variants. Unfortunately, the method has fundamental problems as described below that lead to wrong results.

Reply > The reviewer has misunderstood the methodology which has led them to incorrectly assert that **HyPrColoc** returns “wrong results”. We robustly address below the specific points that led to this erroneous conclusion, and moreover provide additional analyses (immediately after our responses to this reviewer) in order to further demonstrate the benefits of **HyPrColoc**. <

Main

concerns.

A1. HyPrColoc makes the assumption of one causal variant. This assumption is a very strong assumption and it is not true in real datasets (Wen et al. Plos Gene 2017, Hormozdiari et al. AJHG 2016, Jansen et al. Human Mol. Genetics 2017). It is shown that not modeling this phenomenon of more than one causal variant in a locus can confound the fine-mapping and colocalizing results.

Reply >

Our proposed approach, **HyPrColoc**, extends the widely-used and widely-cited **coloc** method of Giambartolomei et al. (2013), which tests for colocalisation between pairs of genetic association studies, to the multiple-trait setting. It identifies clusters of traits that share a causal variant, and goes well beyond the current state of the art approach for multiple trait colocalization (**moloc**; see Giambartolomei et al.; 2018) by allowing vast numbers of traits to be considered (e.g. 100 traits can be jointly analysed in around 1 second using **HyPrColoc**, whereas **moloc** is computationally impractical beyond 4 traits). **HyPrColoc**'s ability to simultaneously assess colocalization across multiple traits contrasts the listed approaches. In common with the existing **coloc** and **moloc** approaches, which have been broadly accepted by the genetics community and have been found in numerous studies to provide insightful and valid discoveries, **HyPrColoc** makes the simplifying assumption of one causal variant. While additionally extending multiple trait colocalization to allow multiple causal variants would be a desirable target for future work, it was not the focus of our manuscript. As we stated in our discussion:

Future work is required to extend this methodology and algorithm to multiple-causal variants. However, we note that the reliability of results under the single causal variant assumption only break down when secondary causal variants explain as much trait variation as the shared variant (Supplementary Material). An example of which is the expression of SH2B3, where multiple causal variants for the expression of this gene masks colocalization with the CHD signal. We note that misspecification of LD between causal variants has a major impact on correct detection of multiple causal variants in a region, making a single causal variant assessment the most reliable when accurate study-level LD information is not available.

Given our own results, and the widespread success of the existing **coloc** algorithm, the reviewer's suggestion that this simplifying assumption necessarily leads to "wrong results" is clearly incorrect.

<

A2. Regarding point 1, how does HyPrColoc perform when we have more than causal variants. It is known that conditional method can detect the wrong set of causal variants.

Reply >

As noted in our previous comment (A1), we explored the reliability of results when the single causal variant assumption does not hold in our Supplementary Material. The reviewer's comment about the "conditional method" is not relevant to our work: **HyPrColoc** does not make use of conditional methods, and nowhere in our manuscript do we advocate their use.

A3. I think the authors need to compare their method with a baseline model. In the baseline model, the authors will perform a fixed effect meta-analysis (absolute value of effect size) of all the traits and then perform a fine-mapping method to detect the causal variants. This baseline method should be extremely fast and produce better results than HyPrColoc.

Reply >

Given a (potentially vast) number of diverse traits, **HyPrColoc** identifies clusters of traits that share a causal variant within a genomic region, as well as simultaneously identifying the shared variant. The reviewer's suggested "baseline model" does not do anything like this, so it is unclear how this would be a useful comparison.

Furthermore, performing a meta-analysis of potentially unrelated traits measured on different scales with potentially different normalisation approaches (using a fixed effects meta-analysis approach) is inappropriate. Consequently, the reviewer's proposed heuristic approach will lead to spurious and uninterpretable results. Please also refer to our response to A5.

<

A4. The main drawback of coloc and HyPrColoc is that the LD information is never utilized.

Reply > The lack of necessity of LD information is a strength of the single causal variant assumption as noted in the original **coloc** paper of Giambartolmei et al 2013. Moreover, as noted in Benner et al 2018, misspecification of the LD structure when aiming to fine-map more than a single causal variant can lead to major biases in results. <

A5. How does the method perform compared to RTC (Nica et al. Plos Gene 2010)? I would guess that RTC to be extremely fast method.

Reply > The RTC method of Nica et al is an empirical colocalization method which does not identify clusters of traits sharing a causal variant, and so is not directly comparable to **HyPrColoc**. Moreover, RTC has a number of fundamental issues: (i) as previously noted in the original **coloc** paper of Giambartolomei et al., 2013, the approach does not provide a formal test of a null hypothesis for, or against, colocalization at the locus of interest and; (ii) RTC has the drawback of having to specify a subset of SNPs on which to base the test, which Wallace (Genet Epidemiol 37: 802–813, 2013) shows can generate significant biases. Given these details, we conclude that a comparison with the RTC method is inappropriate. <

Additional analyses:

To further illustrate the advantages and benefits of **HyPrColoc**, here we compare its performance with **eCAVIAR** (Hormozdiari et al., 2016), which Reviewer 1 mentioned in their first comment (see Figures 1 and 2 below). In the analysis of 10 traits we show that **HyPrColoc** can outperform **eCAVIAR** in terms of (i) correctly fine-mapping the shared causal variant and (ii) computational efficiency - the single joint analysis of **HyPrColoc** was computed >1000 times faster than the 45 pair-wise analyses of **eCAVIAR**. Furthermore, we note that in these simulated analyses we made available the exact LD structure to **eCAVIAR**, something that is unlikely to be available in practice. Hence, in real multiple trait analyses, we expect that the results from **HyPrColoc** are likely to outperform **eCAVIAR**.

Unless stated otherwise, references to figures are to the figures 1-4 displayed at the end of this document.

Reviewer #2 (Remarks to the Author):

B0. Foley et al. propose an analytic approach to perform colocalization analysis across multiple traits. I applaud the authors taking on a difficult yet crucial methodological challenge in statistical genetics. Unfortunately, I have a few serious concerns about the work presented in this manuscript. I hope the following comments help the authors to improve the manuscript.

Reply > We thank the reviewer for their comments and have taken the suggestions into account in revising the manuscript. <

B1. The paper focuses on an approximation solution to compute the likelihood/Bayes factor from the data when the number of traits is large and possible scenarios grows exponentially. However, the other integral component of the Bayesian computation, i.e., the prior specification, does not receive sufficient attention and, in my opinion, oversimplified.

Reply > We have now expanded the description and rationale behind the priors we have chosen. Our choice was in part based on necessity as the prior configuration space, like the hypothesis space, grows super-exponentially in the number of traits. Considering just 20 traits, the number of prior parameters that a user would need to specify would be several hundred trillion (using a **coloc**-type prior specification). Our aim was therefore to provide a compromise between a flexible prior specification and limiting the number of

parameters required to run analyses. Our two proposals for the distribution of priors aim to tackle these issues simultaneously. Nevertheless, considering the reviewer's valuable comments, we have made adjustments to the manuscript and specification of the prior parameters within the **HyPrColoc** software. Our focused responses are found below comments B2 and B3, which tackle specific concerns about our choice and discussion of prior information as well as excerpts from our updated manuscript.

B2. The authors describe two strategies in specifying the necessary priors for possible configurations. They are reasonable mathematical assumptions, but the practical implications on genetic applications do not seem to be sensible. For both strategies, the prior does not depend on the actual traits of interest and all the traits are assumed exchangeable. -- this can be dangerous. For example, consider an analysis dealing with both highly polygenic traits (e.g., molecular traits, heights) and the traits like type II diabetes. It is not reasonable to assume an equal prior of causal variants for all traits, because it implies that all traits have the same prior expected association signals. Unfortunately, both prior strategies proposed by the authors have such undesired property. In comparison, existing works like coloc define the prior property for each trait (i.e., p_1 , p_2) individually.

Reply > We would like to thank the reviewer for highlighting the very important issue of trait exchangeability (or non-exchangeability) which we have now incorporated into **HyPrColoc** to help improve specification of the prior configuration probabilities.

The reviewer is correct to state that not allowing for trait specific information to be incorporated into the causal configuration priors may lead to poorer results. We have incorporated the reviewer's idea of trait exchangeability to improve our "variant specific" prior strategy to allow users to now specify: (i) the prior probability of a trait colocalizing with a cluster of co-localized traits containing at least one related trait and; (ii) the prior probability of a trait colocalizing with a cluster of co-localized traits which does not contain a related trait. Note that any cluster can contain one or more traits. This requires the additional specification of two parameters only and therefore meets the challenge of avoiding setting an impractical number of prior parameters whilst allowing for some information about the (non)exchangeability of traits to be included a-priori:

*"Our variant-level prior extends that of COLOC² and MOLOC⁸ to a framework that is suitable for the analysis of large numbers of traits while also allowing prior information about the relatedness of traits to be incorporated, i.e. two or more traits that are a-priori believed to be related may be more likely to colocalize than unrelated traits. We adopt an approach which requires the specification of a partition of the traits into clusters of related traits, together with three interpretable parameters: p , the probability that a variant is causal for one trait; γ_r , where $1 - \gamma_r$ is the conditional probability that a variant is causal for a second trait given it is causal for a related trait and; γ_u , where $1 - \gamma_u$ is the conditional probability that a variant is causal for a second trait given it is causal for an unrelated trait (**Methods**). Related traits are considered a-priori more likely to colocalize, i.e. $\gamma_r \leq \gamma_u$, hence by default our software assumes that all traits are unrelated." (Page 6 main text)*

<

B3. Because the proposed method is sensitive to prior specification and the approximation of the likelihood function/Bayes factor, the authors should provide a

procedure/guideline to perform model checking and validation. The sensitivity of the colocalization results is well-documented in the literature when analyzing two traits. I can only imagine that the problem gets worse when extending the application to many more traits. I suggest that the authors i) examine and show the sensitivity of the proposed approach; ii) propose a reasonable procedure to guide users in checking model/parameter assumptions.

Reply > We agree with the reviewer, prior specification and sensitivity of results to that specification is an issue in applied Bayesian analyses - which includes colocalization analyses. We have therefore written an analysis protocol as part of our online software tutorial. Our protocol assesses: (i) the sensitivity of results to the choice of prior parameters, across a variety of (recommended) values for the prior parameters and; (ii) provides a discussion on the interpretation of such a sensitivity analyses when analysing colocalization across more multiple traits.

“We additionally provide an example analysis protocol in our online vignette, which accompanies our software (<https://github.com/jrs95/hyprcoloc>), offering guidance on the choice of prior configuration probabilities and assessing any sensitivity of the clusters of colocalized traits identified to the choice of prior parameters.”

(Page 12 main text)

<

B4. In my opinion, the underlying statistical problem of colocalization is best described as a model selection problem not a hypothesis testing problem. Although using the hypothesis testing language is not wrong in describing the proposed procedure, in many places of the paper, the descriptions are heuristic and lack of rigor. Fundamentally, the underlying problem is about classification, that is, classify the alignment of the causal genetic variants into many possible classes. Thus, we should care about the overall measure of misclassification. I fail to see the proposed approach discussing this issue, or provide a general measurement of misclassification in the simulation studies. Hypothesis testing, on the other hand, focuses on controlling one specific type of classification errors and does not seem appropriate in the context of dealing with multiple traits.

Reply > We describe the number of ways in which one or more traits can share (or not share) a single causal variant using a set of (mutually exclusive and exhaustive) hypotheses as this is same language used in the colocalization methodologies proposed by both **coloc** (Giambrolomei et al 2013) and **moloc** (Giambrolomei et al 2018). **HyPrColoc** extends the methodology of these approaches. We therefore feel that adopting this terminology will be appreciated by readers already familiar with these approaches. We purposefully avoid formally describing the space of all hypotheses as our **HyPrColoc** approximation does not require it. Instead we partition the space of all hypotheses into those we enumerate (to approximate a colocalization posterior probability) and those we do not: this avoids introducing extraneous mathematical details into the main text. We do, however, provide a formal mathematical description of the **HyPrColoc** approximation, in a uniform asymptotic analysis framework, in the supplementary details (see e.g. “Properties of the **HyPrColoc** approximation”).

We acknowledge the need for model selection language and therefore make note in the manuscript (Methods) that each causal configuration, related to a given hypothesis, is equivalent to a model which locates a causal variant (if any) for each trait. Hence,

identifying a causal configuration and related hypothesis with the most support given the data is equivalent to selecting a best model given the data. The connection between the underlying model and the hypothesis space is described in both Wakefield, 2009, and Giambartolomei, 2013, references [2] and [13] respectively:

“We describe the space of multi-trait colocalization models using a set of mutually exclusive hypotheses and causal configurations as this approach extends the methodology and language used previously^{2,8}. We note, however, that each causal configuration is equivalent to a model which, for each trait, details the location of the causal variant in the region. Hence, the problem of identifying a hypothesis and causal configuration with the greatest support given the data D , is equivalent to identifying the joint trait-variant model with greatest support^{2,13}.”
(page 26, Methods, main text)

We accept the reviewer’s point about a lack of clarity on classification probabilities. This question overlaps with the main points in B6. Hence, our focused responses to classification probabilities can be found in our reply to B6.

<

B5. Like coloc, the proposed approach assumes at most one causal variant per trait in a genomic locus of interest. I don't disagree with such an assumption, but feel that the authors should provide details on defining proper regions for analysis. This is because if the region defined contains more than one causal variant for a specific trait, the math shown by the authors will break down and give erroneous answers. It seems that some fine-mapping analysis should be carried out to ensure the assumption indeed holds.

Reply > A number of approaches are in common practice to define a genomic locus around a sentinel SNP in the literature, e.g. (i) define a +/- 500Mb window around the sentinel SNP or; (ii) define a region based on a pairwise r^2 measure cut-off between SNPs within a +/-XXMb window (for a specified value XX). However, in our manuscript

“We performed colocalization analyses in pre-defined disjoint LD blocks spanning the entire genome²⁴” (Page 13 main text)

This approach places no restriction on the number of causal variants in the region. Restricting analyses to regions containing a single causal variant could lead to incorrect inference of colocalization due to missing (SNP association) data. Moreover, we note that if we were to reduce the size of the genomic region of interest, attempting to meet the single causal variant assumption for each trait, we would necessarily reduce the proportion of non-colocalization causal configurations relative to the number of colocalization causal configurations which has the potential to increase the false positive rate. To see this we note that, the number of non-colocalization causal configurations grows as Q^m , where Q denotes the number of variants in the region and m the number of traits. The colocalization causal configurations, however, grow as Q^{m-1} , i.e. when two traits colocalize and $m - 2$ do not. Hence, the number of non-colocalization causal configurations grows Q times quicker than the colocalization configurations. Reducing the number of variants Q in a region necessarily increases the proportion of

colocalization to non-colocalization configurations. This would (mathematically) bias analyses toward a colocalization hypothesis as evidence supporting a colocalization configuration would now be weighted relative to a smaller (aggregated) number of non-colocalization models/configurations.

Given the reviewers concerns, we now provide more information as to why we define regions via approximate LD blocks: through new simulations and some mathematical reasoning discussed below.

We assessed the performance of **HyPrColoc** against **eCAVIAR** - a method which allows for multi-causal variants in the region per trait but is limited to the analysis of two traits only – when two or more traits have more than a single causal variant in the region (**Figures 1-3**). We do this to show that either **HyPrColoc**: (i) maintains reasonable performance or; (ii) returns conservative colocalization results (which is the preferred behaviour). This helps us address some of the reviewers concerns about “defining...regions for analyses” and what happens when “the math breaks down” owing to violations of the single causal variant assumption. In summary, we show that when two or more traits have an additional distinct causal variant:

- **HyPrColoc** maintains good accuracy (median value ≈ 0.85) to detect colocalization across multiple traits when an additional causal variant explains less than, but near to, the trait variance explained by a shared causal variant (**Figure 3**).
- **HyPrColoc** can recover better fine-mapping results than **eCAVIAR**, owing to jointly fine-mapping a shared variant across more than a pair of traits. (**Figure 1**).
- the power and accuracy of **HyPrColoc** drops when an additional causal variant explains the same or more trait variation than the shared causal variant per trait (**Figures 2 and 3**). This is owing to an increased false negative rate, i.e. power for a pair of traits is halved and the median Rand index for 10 traits drops from ≈ 0.75 to ≈ 0.5 , see **Figure 2**. We discuss reasons for this in the paragraph below.
- **HyPrColoc** computed results over 1000 times faster (median time) than the 45 pairwise analyses of **eCAVIAR** for 10 traits. The time difference increased in larger genomic regions and would necessarily increase further as the number of traits under consideration increases beyond 10 traits.

The single causal variant assumption, therefore, remains reasonable when any additional causal variants explains less trait variation than a shared variant, otherwise our results indicate that **HyPrColoc** returns conservative results (which supports our theoretical results made in the Supplementary Materials p44). This, together with the speed gains of **HyPrColoc**, motivate us to not place a restriction on defining regions based on the increased likelihood of meeting the single causal variant assumption.

While the mathematical details are correct for up to one causal variant per trait only, violations of the single causal variant assumption (owing to multiple causal variants per trait) allow evidence supporting non-colocalization hypotheses to grow more quickly than evidence supporting colocalization hypotheses. This is a consequence of the relative sizes of the causal configuration space between non-colocalization and colocalization hypotheses (models). The number of non-colocalization configurations is in general much larger than colocalization configurations and, consequently, when two causal

variants are present a non-colocalization configuration benefits from this more, as evidence is computed at more than a single variant in the region for non-colocalization configurations. In comparison, a colocalization configuration computes evidence at a single variant in the region. This explains why we return conservative colocalization results when the single causal variant assumption is violated (**Figure 2**). While the reviewer is correct to state that “the maths breaks down”, this breakdown will lead to more conservative results (particularly if additional causal variants explain as much or more trait variation than a shared variant). We provide a form uniform asymptotic investigation of the HyPrColoc approximation, when there are multiple causal variants in the region, in the Supplementary Material (page 44). <

B6. The simulation studies need to be better designed and reported. The calibration of the resulting colocalization probabilities needs to be examined. The general classification error rates should be reported. The robustness of the proposed method should also be addressed.

Reply > We accept the reviewer’s point about a lack of clarity on classification probabilities. **HyPrColoc** is simultaneously a principled probabilistic method for sharing information across traits to improve the identification of causal variants and a clustering algorithm which identifies clusters of traits that share a causal variant. Previously, we focused on assessing performance just in terms of our ability to identify the causal variant; we have now extended our assessment to assess performance in terms of our ability to correctly partition traits into clusters. There are numerous ways in which clustering performance can be assessed and we have added more details on this. In particular, we have included the following clarification in the text to highlight the two approaches we now consider:

“Here we assess the performance of the branch and bound (BB) divisive clustering algorithm to identify clusters of colocalized traits over a range of scenarios and using two classification criteria: firstly, we calculate a separate classification probability for each of the possible partitions of traits into clusters, and; secondly, we assess the overall measure of classification by computing the accuracy (proportion of correctly specified traits) of HyPrColoc.” (Page 10 main text).

In light of the reviewers point, we now include the general classification measure ‘accuracy’ (**Figures 3 and 4**). We define this in the manuscript (Methods):

“Accuracy is defined as:

$$Accuracy = \frac{TP + TN}{TP + TN + FP + FN}$$

where TP and TN denote the true positive count and true negative count, and FP and FN denote the false positive count and false negative count. Hence, accuracy is the proportion of traits that are correctly identified as colocalizing or not colocalizing. For each cluster of traits identified, a colocalizing trait is correctly identified if it is included in the largest group of colocalized traits; otherwise it is incorrectly identified. A non-colocalizing trait is correctly identified if it is not judged as colocalizing with any other trait; otherwise it is incorrectly identified.” (Page 34, Methods, main text)

We retain Figure 6 in the main manuscript as this provides some clarity on the probability of individual trait classifications and provides more granular information on classification than the aggregated measure of accuracy. For example, suppose two traits have been wrongly classified as colocalized. There are three ways this can occur: (i) the two traits jointly colocalize with a cluster of truly colocalized traits; (ii) the traits separately colocalize with two distinct clusters of colocalized traits or; (iii) the traits colocalize with one another only. The value of the ‘accuracy’, given above, is the same in all these scenarios. To go some way toward avoiding this ambiguity, our box plot (**Figure 6** main text) presents separate classification probabilities for each of the partitions (of traits into clusters) with non-zero probability.

We note that an exhaustive quantification of all partitions of traits into clusters is impossible (as there are far too many partitions of the traits to enumerate in a figure or table). Hence, by now presenting both accuracy and relevant partition probabilities, readers should now get an idea of the “general classification error rate”, as requested, as well as receive some helpful guidance on the types of partitions of the sample that **HyPrColoc** makes.

In summary, we have clarified the reporting of the simulations by adding further details on (i) “the calibration of the resulting colocalization probabilities needs to be examined”; (ii) “the general classification error rates should be reported” and; (iii) “the robustness of the proposed method should also be addressed.”

- **“Robustness of the proposed method”**
The performance of **HyPrColoc** is assessed when the assumption of a single causal variant is violated for all or some of the traits in the sample, and we benchmark our results against **eCAVIAR** which allows multiple causal variants in the region. Page 13, a note in the Supplementary Material and supplementary figure **Figure S6**.
- **“The general classification error rate”**
The ‘accuracy’ of **HyPrColoc**, which provides a general measure of the classification of traits into clusters of colocalized and non-colocalized traits, is now computed. Pages 11, 12 and 35 (Methods) and **Figure 6d** in manuscript;
- **“The calibration of the resulting colocalization probabilities needs to be examined”**
We now provide more detail on the cluster-level classification probabilities, which relate to distinct partitions of the traits, by contrasting these with the overall classification measure ‘accuracy’. Pages 11, 12 and **Figures 6a-d** in manuscript.

<

Figure 1. Fine-mapping a shared causal variant between traits which have up to two causal variants in the region.

Presented are results from HyPrColoc (using software default settings) and eCAVIAR. For eCAVIAR, a variant is deemed shared between traits when either: (i) the colocalization posterior probability (CLPP) is above 0.05 (eCAVIAR_CLPP_5%) or, under a less stringent threshold; (ii) the CLPP is above 0.01 (eCAVIAR_CLPP_1%). In scenarios A and B, two traits share a causal variant and both traits have an additional distinct causal variant in the region explaining either 0.75% of trait variance (scenario A) or 1% of trait variance (scenario B). In scenarios C and D, ten traits share a causal variant and five of these have an additional distinct causal variant in the region explaining 0.75% of trait variance (scenario C) or 1% of trait variance (scenario D). In all scenarios the shared variant explains 1% of trait variance and each study had a sample size of $N=20000$. The data were generated using the simulation protocol outlined in the main text (page 8).

Figure 2. (**Left of panel**) Power to detect colocalization between a pair of traits and (**right of panel**) the Rand index. The Rand index denotes the similarity between the HyPrColoc identified cluster of colocalized traits and the true cluster of colocalized traits, and is used when assessing more than a pair of traits. eCAVIAR does not allow for joint analyses of more than a pair of traits and so we do not compute the Rand index for eCAVIAR. All results are deduced following the same simulation protocol as in Figure 1.

Figure 1. Accuracy of HyPrColoc when two or more traits have multiple causal variants in the region. All results are deduced following the same simulation protocol as in Figure 1..

Figure 2. Accuracy of HyPrColoc when identifying one or more clusters of colocalized traits from a collection of 100 traits which have at most one causal variant in the region. In scenario A, there is a single cluster of 10 traits which share a causal variant, the remaining traits have no causal variants in the region. In scenario B, there is a single cluster of 10 traits sharing a causal variant, a collection of 10 traits which have distinct causal variants and the remaining 80 traits do not have a causal variant in the region. In scenario C, there are two clusters of 10 traits which colocalize at distinct causal variants, the remaining 80 traits do not have a causal variant in the region. The causal variant, when present, explains 1% of trait variation per trait. A full description of the simulation protocol is given in the main text (page 10).

Reviewers' comments:

Reviewer #1 (Remarks to the Author):

I appreciate the authors' effort to respond to the raised concerns. However, the main concern is still not answered.

1) HyPrColoc makes the assumption of one causal variant. This assumption is very strong assumption and it is not true in real datasets (Wen et al. Plos Gene 2017, Hormozdiari et al. AJHG 2016, Jansen et al. Human Mol. Genetics 2017). It is shown that not modeling this phenomenon of more than one causal variant in a locus can confound the fine-mapping and colocalizing results.

2) It is worth mentioning that COLOC currently adds the conditional method to solve the problem of multiple causal variants in a sub-optimal way.

3) LD information is never utilized and LD provides great information to avoid false positive. COLOC utilize conditional method to solve this problem (COLOC solution is not great).

Reviewer #2 (Remarks to the Author):

I appreciate the authors' efforts in addressing the issues raised in the previous review. Unfortunately, I don't feel that the current revision substantially addresses my concerns on this manuscript. The detailed comments are provided below.

1. On prior specification. The authors added additional text to address the issue of prior specifications in "Methods" and the online vignette of the software page. But I think they should take this issue more seriously. Are any of the simulation results sensitive to prior specifications? In the real data analysis, how the priors are selected, and are the results sensitive to these choices? If the authors' own analysis exhibits such a lack of rigor (or even a lack of effort), it can be very dangerous for readers to follow the suite. I agree with the authors' assessment that "our aim was therefore to provide a compromise between a flexible prior specification and limiting the number of parameters required to run analyses." But the compromise needs to be reasonable and ensures correct conclusions. I honestly can not say this to the current solution provided.

2. On one causal SNP assumption. I take some issues on the authors' mathematical reasoning on the potential consequence of the assumption violation. It is probably true that "The number of non-colocalization configurations is in general much larger than colocalization configurations." However, I am not sure that those configurations are correctly counted in the coloc-type of calculation. Specifically, I worry the normalizing constant for the colocalization probability can be systematically *under-estimated*. Although I don't doubt the results from the new simulations, I am not entirely convinced by the authors' reasoning. The authors might very well be correct on this issue, but a more detailed mathematical argument is needed to justify the claim (some formulas can be helpful).

3. On evaluation of classification errors. Based on the authors' argument that the proposed method is robust to model misspecification. It seems necessary to separately evaluate false positives and false negatives. The current measure of accuracy makes it extremely difficult to validate the authors' own point on robustness. Presenting standard measures like realized false-discovery rate *and* power should suffice.

Reviewer #4 (Remarks to the Author):

Foley and colleagues present the first method that is capable of computing colocalization using a large number of traits. This is a novelty because other existing methods cannot practically perform with more than 4 traits.

The method is statistically sound and the derivations are very well described. The authors expose mathematical derivations which are very useful to direct the community for further development on this challenging yet important task.

The authors propose approximating the posterior probability of full colocalization, and the first step includes a novel Bayesian divisive clustering algorithm. I find this the most novel part of the approach. Personally, if the authors find it suitable for this goal, I am wondering whether the method would be better suited to identify regions of the genome important for disease and to prioritize traits, rather than focusing on assessing full colocalization of all traits. However I have some concerns on the behavior of the statistic and the priors, and on the real data application. I describe below in detail.

I hope addressing these could help the practical applications of this method and its use in real data analysis.

1.

The first step of the algorithm aims to define the cluster of traits with strongest evidence of regional association, and HyPrColoc uses the regional selection criterion to partition a cluster.

(i) It seems as there is a threshold for increasing power with additional number of traits - above which it is not convenient to add another trait (e.g. in Supplement, from reducing the number of traits from 100 -> 20, there is a ~5% increase in true detection).

(ii) It is also not clear whether using this method with $m=2$ traits recovers all (or most) of the signals using pairwise colocalization. It seems as there is no advantage over eCAVIAR in performing HyPrColoc with two traits.

Therefore, I am wondering whether this method would be better suited to prioritize traits and regions to later fine-map. In particular, I am wondering if this model could be used primarily to find the location of the causal variant in the region, and prioritize traits that are more informative, to then use these in a fine-mapping exercise. This could maximize the strength of this model, since it can test many traits at a time for relevance to disease.

2.

Realistically, not all traits used in the HyPrColoc analysis will have the same sample size, and in fact will have folds differences one with another (regular sample sizes for GWAS > 20K, but eQTLs <1000).

I have a concern on the behavior of the statistics when using unbalanced sample sizes.

In particular:

(i) when defining the cluster of traits, how different sample sizes will influence the statistics is not clear (e.g. 20 traits of which 10 traits colocalize, all with different sample sizes, but say it so happens that the traits that do not colocalize have higher sample sizes - and since the clustering algorithm is a sequential decision tree (Figure 3), the combinatorial evidence could be completely mis-calculated due to combination of sample sizes supporting the different configurations).

(ii) possibly a prior that controls for N may be more appropriate here?

3.

I share Reviewer 2's concerns on priors, polygenicity, and regional definitions.

The authors provide a nice theoretical explanation of priors, and clarify the explanation of different approaches for priors. In fact, making available a sensitivity analysis pipeline for the prior misspecification is a great contribution from the authors.

However, I do not understand the link between the priors and the number of causal configuration that the authors describe (i.e., "when two causal variants are present, a non-coloc configuration benefits from this more").

The algorithm (priors) should account for differences in the number of colocalization vs not colocalization scenarios. If it doesn't, there is a problem with the algorithm.

Since we are assessing evidence at the region-level, the combination of different priors will influence the lower bound needed for the evidence to declare presence of a CV and colocalization. We usually do not have information on this, although authors give a good way to account for that in case we do.

If we use a method where the variant-level prior is set: say we have two traits, the prior probability of colocalization of independent traits ($p_1 * p_2$) versus related traits (p_{12}), should control for the difference in number of configurations in the coloc versus non-coloc situation. From what I understood, this should be the function of p and γ in the variant-level prior approach, where γ is the conditional prior probability. Please clarify in the text, along with the practical values recommended for p and γ (possibly with reference to p_1 , p_2 , p_{12}) when the two traits are independent.

On this topic, I found the author's reply to Reviewer 2 on region definition unsatisfactory. Reviewer 2's question seemed to be about the window size used for a region, possibly because LD blocks are very large and very likely to overlap many causal variants. This is also related to Reviewer 2's concern on polygenicity.

If the priors are set at a region-level, and the region is as big as an LD block, it will most likely violate the assumption of one causal variant. Although the algorithm is robust to this violation, the argument seems to state that the test can be biased if we were attempting to meet the single causal variant assumption, because of an unbalanced proportion of causal and non-causal configurations.

The authors reply using the relationship of Q (number of SNPs in the region) and number of causal configurations, which is confusing. There needs to be a distinction between Q , and region definition. The denser the Q for a set region (i.e., better imputed), the better estimates of colocalization we should have because we have a better chance to capture the causal variant in the region.

4.

I am very skeptical on the use of LD blocks in real data analysis.

A whole LD block is almost insured to include > 1 causal variant.

I am wondering in fact if the divisive clustering algorithm can be used first to select signals (e.g. ± 500 kb around the most causal SNP) around which to build colocalization analysis, assuming potentially only 1 signal in the region.

Additionally, it looks like there is inconsistency in real data (using LD blocks) and simulations. In simulations, regions are defined around a CHD signal. What is the mean Q in simulations? Region size?

Moreover, the region definition was changed when analyzing eQTL and pQTL (page 15). This is confusing. Indeed if the aim is to find a potential gene to build a drug target to disease, it makes sense that the region definition is centered around a gene. On the other hand, setting the region definition to around a GWAS signal also makes sense since we are interested in testing colocalization of the GWAS signal with other molecular traits. If both steps are included (centered on the LD blocks or around cis-eQTLs/pQTLs), please describe how many regions are discordant using the two approaches of region definition.

Reviewers' comments:

Comments to all reviewers:

We would like to thank all reviewers for their comments. We have addressed them which has resulted in improvements to our manuscript and the usability of our software. All changes in the manuscript and supplementary material are marked in yellow and we provide the locations of any edits made to the manuscript (**section**, page number, line number) and supplementary material (**Supplementary Material**, page number, line number) herein.

In particular, we have substantially improved and clarified our simulation design as follows: (a) we provide two new simulation studies which better assess performance of HyPrColoc across a range of scenarios under (i) the single causal variant assumption and (ii) violations of the single causal variant assumption - in both studies we now assess performance relative to the alternative of performing repeated rounds of pairwise colocalization analyses, using COLOC in (i) and (ii) and eCAVIAR in (ii); (b) we summarise the performance of each method in terms of three classification measures (accuracy, true positive rate and false positive rate) and over a range of values of the colocalization prior probabilities (to assess sensitivity of results to the specification of priors); (c) we provide guidance for users on the specification of colocalization priors and the reporting of results and; (d) provide an example of HyPrColoc's sensitivity analyses pipeline tool.

In order to make some comparisons between the results from HyPrColoc and pairwise methods in the two simulation studies, we reduced the number of traits under consideration relative to our previous simulation study, i.e. previously we assessed 100 traits and now we assess 10. Both pairwise COLOC and the multi-causal variant software eCAVIAR cannot be used to analyse 100 traits. eCAVIAR can take up to 1 hour to perform 1 round of the 45 pairwise comparisons necessary to analyse 10 traits in regions with around 1000 SNPs. In our new simulation studies, we additionally assess performance of each method over 3 different scenarios in which one or more clusters of traits colocalize in a region.

With respect to (a) and (b) above, the manuscript has been edited and now reads:

(Branch and bound divisive clustering algorithm, page 11, line 2):

“we assess the performance of the branch and bound (BB) divisive clustering algorithm to identify clusters of colocalized traits over a range of scenarios, several specifications of the colocalization prior p_c and using three classification criteria: the accuracy, which is an overall measure of the classification of traits into clusters; the true positive rate (TPR) and; the false positive rate (FPR), see Methods for more details. We simulated 10 traits from non-overlapping datasets under three scenarios: (i) a single cluster of 10 colocalized traits; (ii) 2 clusters of 3 colocalized traits, the remaining 4 traits do not have a causal variant (reflecting “hypothesis free” colocalization searches) and; (iii) 4 clusters of colocalized traits, comprising 2 clusters of 3 traits and 2 clusters of 2 traits. Each cluster of colocalized traits share a single causal variant and causal variants between clusters are distinct, but can be in perfect LD, i.e. $r^2 = 1$, with one another. To mirror scenarios in which data are taken from studies with different sample sizes, we take the number of individuals in each study (N_i) as a random draw from the set $N_i \in \{1k, 5k, 10k, 15k, 20k\}$. We

compare this with results when each study has a large sample size by additionally performing an analysis in which $N_i = 15\text{k}$ for all traits.”

(Branch and bound divisive clustering algorithm, page 11, line 17):

“We assess sensitivity to the choice of colocalization prior p_c , i.e. $(1 - \gamma_u)$, by performing three separate analyses for each dataset using $p_c \in \{0.05, 0.02, 0.01\}$. These values can result in substantial differences in the prior probability of colocalization as the number of traits in a cluster increases (**Methods**). For comparison, we compare HyPrColoc against the alternative of performing pairwise colocalization analyses using COLOC², which restricts clusters sizes to two traits only. Results are presented in **Figures 6(a-g)**.”

(Violations of the single causal variant assumption, page 14, line 13):

“We assessed the performance of HyPrColoc when two or more traits have more than a single causal variant in the region and compare these results with those obtained using pairwise COLOC and pairwise eCAVIAR⁵ (with a colocalization posterior probability, CLPP, cut-off of 1%), another software package for colocalization which allows each trait to have multiple causal variants but is limited to the analysis of pair of traits only. We simulated data for 10 traits and allowed up to 5 traits to have additional distinct causal variants in the region, so that the sample contains a mixture of traits which either satisfy or violate the single causal variant assumption. The data are generated under three scenarios, as previously, but now each cluster of colocalized traits share a single causal variant and half of the traits in a cluster have secondary distinct causal variants (**Methods**). In terms of marginal genetic associations, the additional variants were randomly selected to explain either slightly less trait variance than the shared causal variant ($\approx 0.75\%$) or the same amount of trait variance as the shared variant ($\approx 1\%$). We varied each study sample size and followed the simulation protocol as above.”

(Methods, Violations of the single causal variant assumption, page 39, line 7):

“These data were generated under three scenarios: (i) a single cluster of 10 colocalized traits, each trait shares a single causal variant and 5 traits have secondary distinct causal variants; (ii) a single cluster of 6 colocalized traits, each of the 6 traits share a single causal variant and 3 traits have secondary distinct causal variants, the remaining 4 traits do not have causal variants and; (iii) 2 clusters of colocalized traits, cluster 1 comprises 6 traits sharing a single causal variant with 3 of 6 traits having secondary distinct causal variants, cluster 2 comprises 4 traits sharing a single causal variant with 2 of 4 traits having secondary distinct causal variants. To maximize the number of traits with additional causal variants in a cluster (up to the maximum of 5), in scenarios (ii) and (iii) the total number of clusters of colocalized traits were reduced relative to the single causal variant assessment.”

With respect to improvement (c), i.e. providing guidance for users on the specification of colocalization priors and the reporting of results, we now make this clear in the manuscript:

(Branch and bound divisive clustering algorithm, page 13, line 13):

“Based on our results, we recommend users set $p_c = 0.02$, i.e. $\gamma_u = 0.98$, and report results from the clusters of colocalized traits which satisfy $P_R P_A > 0.7$. Setting $p_c = 0.02$ increased the number of datasets in which clusters satisfying $P_R P_A > 0.7$ were identified **(Methods)** while maintaining a low FPR throughout.”

With respect to improvement (d), i.e. providing an example of HyPrColoc's sensitivity analysis pipeline, the manuscript now reads:

(Branch and bound divisive clustering algorithm, page 13, line 20):

“The computational gains of HyPrColoc make it practical to perform multiple rounds of colocalization analyses, each with different values of the prior p_c and the threshold parameters P_R^*, P_A^* , to assess any sensitivity in the clusters of colocalized traits identified to changes in parameter specifications. An example of this, taken from data generated under scenario (iii), is presented in **Figure 6g**. The resulting heatmap highlights the presence of four clusters of colocalized traits in the data and these clusters persist across most of the prior and threshold parameter settings. We include this sensitivity analysis in the HyPrColoc software and recommend its use.”

Reviewer #1 (Remarks to the Author):

A1. HyPrColoc makes the assumption of one causal variant. This assumption is very strong assumption and it is not true in real datasets (Wen et al. Plos Gene 2017, Hormozdiari et al. AJHG 2016, Jansen et al. Human Mol. Genetics 2017). It is shown that not modelling this phenomenon of more than one causal variant in a locus can confound the fine-mapping and colocalizing results.

Reply > We have extended our assessment of violations of the single causal variant assumption in multi-trait colocalization analyses. HyPrColoc is vastly more computationally efficient than alternative methods, (e.g. it is 100,000 times faster than the multi-causal variant method eCAVIAR for 10 traits in a region of 1000 SNPs) which makes it practical to perform robust sensitivity analyses as part of regular use of the software (and we provide tools to do this).

We demonstrate that by considering only clusters of colocalized traits whose posterior probability of colocalization is greater than 0.7, we markedly improve the classification probabilities of HyPrColoc (colocalization results are generally only reported when the posterior probability of colocalization is greater than a threshold value which we take to be 0.7). In this situation, the false positive rate (FPR) of HyPrColoc matches that of eCAVIAR, i.e. is approximately zero. This was true for each choice of colocalization prior probability used when running HyPrColoc (see **Supplementary Figures S4** and **S6** to compare results without thresholding and after thresholding). Our results provide good evidence that issues

of confounding of fine-mapping results (when the single causal variant assumption is violated) can be attenuated by reporting only the clusters of colocalized traits with strong evidence of colocalization. We have edited the manuscript to reflect this.

(Branch and bound divisive clustering algorithm, page 13, line 13):

“we recommend users set $p_c = 0.02$, i.e. $\gamma_u = 0.98$, and report results from the clusters of colocalized traits which satisfy $P_R P_A > 0.7$. Setting $p_c = 0.02$ increased the number of datasets in which clusters satisfying $P_R P_A > 0.7$ were identified (**Methods**) while maintaining a low FPR throughout.”

Our suggested protocol for reporting results does impact on power to detect clusters of colocalized traits, and we note this in the manuscript

(Methods, page 40, line 8):

“In the simulation study which allowed each trait a maximum of one causal variant in the region and with respect to scenarios (i), (ii) and (iii), when setting $p_c = 0.05$ HyPrColoc identified clusters of colocalized traits with $P_R P_A > 0.7$ in approximately 70%, 93% and 99% of simulated datasets, when $p_c = 0.02$ in approximately 65%, 91% and 98% datasets, reducing to around 60%, 86% and 97% of datasets when $p_c = 0.01$. Pairwise COLOC identified pairs of colocalized traits with $P_R P_A > 0.7$ in over 96% of simulated datasets, across all three scenarios and specifications of p_c . In the simulation study which allowed a maximum of two causal variants per trait, these figures reduced: when setting $p_c = 0.05$ HyPrColoc identified clusters of colocalized traits with $P_R P_A > 0.7$ in approximately 65%, 80% and 93% of simulated datasets, when $p_c = 0.02$ in approximately 60%, 72% and 92% datasets, reducing to around 55%, 65% and 85% of datasets when $p_c = 0.01$. Pairwise COLOC identified pairs of colocalized traits with $P_R P_A > 0.7$ in over 94% of simulated datasets, across all three scenarios and specifications of p_c .”

The massive computational gains of HyPrColoc, which allow for rapid assessments of large numbers of traits, makes up for this loss in power. Moreover, the quality of results, i.e. the accuracy, true positive rate and false positive rate, of HyPrColoc in these scenarios is much improved relative to the pairwise methods (**Figures 7 and S6**). The pairwise approaches are fixed to perform 45 rounds of pairwise analyses – around a 10-fold increase in the number of colocalization assessments that HyPrColoc made across the three scenarios. Having to perform a large number of pairwise assessments can be problematic, not just in terms of computation, but also in terms of interpretation.

Specifically, a cluster of 3 or more colocalized traits is identified using a pairwise approach if and only if all pairs of traits colocalize at the same shared causal variant(s) (or one in perfect LD). We found that, after thresholding results based on a posterior probability of colocalization >0.7 , many pairs of traits which are part of a large cluster of colocalized traits were not identified as being colocalized. For example, traits 1 and 2 and traits 2 and 3 might be identified as colocalizing above 0.7, but traits 1 and 3 were not. Even when pairs of traits from a large cluster of colocalized traits were identified, all pairs often did not colocalize at the same variant or at variants in perfect LD with one another. The multi-trait approach of HyPrColoc does not return these types of ‘fuzzy’ or ‘partial’ colocalization results. This is a much needed improvement on pairwise approaches and it also helps to explain why the multi-trait colocalization approach (HyPrColoc) can perform better at identifying a shared

causal variant even when the single causal variant assumption is violated as information about a causal variant is shared across ≥ 2 traits in a cluster, see **Figures S4c and S6c**.

We modify the manuscript to acknowledge the above points as follows:

(Branch and bound divisive clustering algorithm, page 13, line 5)

“In pairwise approaches, a cluster of 3 or more colocalized traits is identified if and only if all pairs of traits colocalize (ideally at the same shared causal variant), the TPR of the pairwise method reduced after thresholding as only some of the pairs of traits passed the posterior threshold which increased the false negative rate. This is a drawback of methods which do not perform multi-trait colocalization.”

(Violations of the single causal variant assumption, page 15, line 18)

“When considering only the clusters of traits identified as colocalizing with $P_R P_A > 0.7$, HyPrColoc again outperformed the pairwise COLOC approach, providing very reliable results across all three classification measures (**Figure S6a-c**). The results indicate that HyPrColoc might outperform pairwise eCAVIAR also. However, this could not be confirmed as the CLPP of eCAVIAR is not equivalent to the posterior probability of HyPrColoc and COLOC, and so a like-for-like comparison was not possible.”

We note that (Wen et al. Plos Gene 2017, Hormozdiari et al. AJHG 2016, Jansen et al. Human Mol. Genetics 2017) did not apply a similar assessment of COLOC, i.e. thresholding the posterior probability and varying the prior probability, to identify situations in which fine-mapping results are not confounded under the single causal variant assumption.

In response to the comment **C1(iii)** by reviewer 4, we now include advice on how HyPrColoc can be used to identify genomic regions and clusters of traits with which to better prioritize the use of more computationally expensive pairwise multi-causal variant methods – should HyPrColoc fail to identify colocalization at a single causal variant using our advised protocol for reporting results. We note this in the manuscript:

(Violations of the single causal variant assumption, page 16, line 8)

“The HyPrColoc algorithm can additionally be used to rapidly identify genomic regions and clusters of traits to better prioritize the use of more computationally expensive multi-causal variant colocalization software (**Supplementary Material**).”

Please see our reply to comment **C1(iii)** for specific details of this approach. The upshot is that even if HyPrColoc fails to identify colocalization between the traits, the HyPrColoc algorithm still has utility as it can rapidly identify regions and traits with which to better employ the approaches of e.g. (Wen et al. Plos Gene 2017, Hormozdiari et al. AJHG 2016, Jansen et al. Human Mol. Genetics 2017).

The results from HyPrColoc demonstrate that intuition and evidence from two-trait colocalization analyses does not carry over to multi-trait analyses. HyPrColoc borrows information across multiple traits to fine-map a causal variant, two trait multi-causal variant methods cannot do this, they instead rely on LD information, which can be problematic if reference datasets rather than study specific LD measures are used. We demonstrate that when multiple traits colocalize, a multi-trait colocalization method (HyPrColoc) outperforms a

pairwise multi-causal variant method, even in the presence of multiple causal variants, and across a range of scenarios. Going forward, the ideal is to have a multi-trait multi-causal variant method, leveraging power from multiple traits and LD information to increase detection rate.

<

A2. It is worth mentioning that COLOC currently adds the conditional method to solve the problem of multiple causal variants in a sub-optimal way.

Reply > We appreciate the reviewer highlighting this, but HyPrColoc does not offer a stepwise conditional method as part of the software.

<

A3. LD information is never utilized and LD provides great information to avoid false positive. COLOC utilize conditional method to solve this problem (COLOC solution is not great).

Reply > We agree, LD information can help. However, LD information, particularly if based on reference genomes rather study specific, can result in mis-leading colocalization (and fine mapping) results. We have shown that borrowing strength across multiple traits can offer improved information (**Supplementary Figures S4 and S6**). In future, we will combine these two sources of information.

We would like to flag that we provided eCAVIAR with the study specific LD information using all samples – this is the best-case scenario for a multi-causal variant fine-mapping method and this information is not readily available in most real applications. Despite having this information, the approach of borrowing strength across multiple traits to identify a shared causal variant performed better (**Figure S4c**). <

Reviewer #2 (Remarks to the Author):

B1. On prior specification. The authors added additional text to address the issue of prior specifications in "Methods" and the online vignette of the software page. But I think they should take this issue more seriously. Are any of the simulation results sensitive to prior specifications? In the real data analysis, how the priors are selected, and are the results sensitive to these choices? If the authors' own analysis exhibits such a lack of rigor (or even a lack of effort), it can be very dangerous for readers to follow the suite. I agree with the authors' assessment that "our aim was therefore to provide a compromise between a flexible prior specification and limiting the number of parameters required to run analyses." But the compromise needs to be reasonable and ensures correct conclusions. I honestly can not say this to the current solution provided.

Reply > We thank the reviewer for highlighting the importance of including more guidance on (a) the specification of prior information and (b) the sensitivity of results to changes in the specification of priors. We now include a thorough review of this and provide the data to support our conclusions.

We link the HyPrColoc prior to the COLOC prior (as recommended by reviewer 4 in comment **C3 (iv)**) by introducing the prior probability parameter p_c – which, conditional on a trait having a causal variant, is the prior probability that a second trait shares the same causal variant. We have edited the manuscript as follows:

(Description of the HyPrColoc method, page 6, line 15)

“We adopt an approach which requires the specification of a partition of the traits into clusters of related traits, together with three interpretable parameters: p , the probability that a variant is causal for one trait (equivalent to parameter p_1 in COLOC²); γ_r , where $1 - \gamma_r$ is the conditional probability that a variant is causal for a second trait given it is causal for a related trait and; γ_u , where $1 - \gamma_u$ is the conditional probability that a variant is causal for a second trait given it is causal for an unrelated trait (**Methods**). Note that, $1 - \gamma_u$ is equivalent to $\left(\frac{p_{12}}{p_{12}+p_1}\right)$ in COLOC². As it will be helpful later, we introduce the conditional colocalization prior p_c , where $p_c = \left(\frac{p_{12}}{p_{12}+p_1}\right) = 1 - \gamma_u$.”

The parameter p_c is composed of two parameters, one of which is the prior probability p , i.e. p_1 in COLOC. By default, HyPrColoc assumes traits are unrelated and in this situation the prior set-up is explained by two parameters p and p_c . Consequently, changing the value/specification of the parameter p_c in analyses automatically alters the influence of the parameter p . We therefore varied the parameter p_c only in our assessment of prior sensitivity and fixed $p = 10^{-4}$ as per Giambartolomei et al 2014.

As noted in our response to all reviewers at the beginning of this document, we now assess sensitivity of results to the specification of p_c by performing three separate analyses for each simulated dataset using $p_c \in \{0.05, 0.02, 0.01\}$. To help understand how the different choices of p_c impact on the prior probability of colocalization between all of the traits, we have modified the manuscript to provide some examples for readers:

(Methods, page 38, line 23)

“For a pair of traits, colocalization between the traits is 5 times more likely a-priori when setting $p_c = 0.05$ relative to $p_c = 0.01$. In the analysis of ten traits, however, colocalization between all ten traits is around 1 million times more likely a-priori when setting $p_c = 0.05$ relative to $p_c = 0.01$ (the prior probability of colocalization is still very small $\sim 10^{-11}$ when setting $p_c = 0.05$). Hence, the different values of p_c we have chosen can result in substantial differences in the prior probability of colocalization.”

As the reviewer points out, some of the results in the simulation are sensitive to the specification of this probability. Results showed more sensitivity when studies with small sample sizes are included and some classification measures were more sensitive than others. For example, in scenarios in which all traits do not colocalize into a single cluster the accuracy of HyPrColoc showed little sensitivity to the choice of p_c – this was because any reduction in the true positive rate was met with a proportionate reduction in the false positive rate when lowering the value of p_c (**Figures 6a and S4a**). There is also some evidence of prior sensitivity when traits which do not have causal variants in the region are included in analyses, e.g. a 30% drop in the median TPR using $p_c = 0.01$ relative to results when setting $p_c = 0.05$ or $p_c = 0.02$ (**Figure 6b**). We additionally assessed results from the clusters of colocalized traits whose posterior probability was greater than 0.7 (attempting to reflect

standard practice in real analyses, i.e. reporting only the results above a posterior threshold value). In this situation, results from HyPrColoc were less sensitive to the choice of prior probability across all three classification measures (**Figures 7a-c and S6a-c**). Although the number of datasets in which we identify clusters with a posterior >0.7 did vary by choice of prior (c.f. our reply **A1** and the **Methods**, page 40, line 8). In contrast, the results from pairwise COLOC were still sensitive to the choice of prior probability. As noted in our reply **A1**, this was due to an increase in the false negative rate of the pairwise approach as some pairs of traits in a cluster were not identified as colocalizing with a posterior probability >0.7 .

We now report on the sensitivity of results to the specification of p_c . The manuscript has been edited in several areas to reflect this:

(Branch and bound divisive clustering algorithm, page 11, line 24)

“We observed that both HyPrColoc and pairwise COLOC perform reasonably well across all three scenarios. The median accuracy and TPR is generally ≥ 0.75 , for all three choices of p_c , improving to around 1 when the sample size of each study is large (**Figures 6a-b; Table S6**) - indicating that including studies with smaller sample sizes decreases the TPR. Accuracy was more sensitive to the choice of p_c when all traits colocalized into a single cluster, i.e. scenario (i), relative to scenarios (ii) and (iii) where we observe little sensitivity to p_c (**Figure 6a**). We noted increased variability in the TPR when traits that do not have a causal variant were included in analyses, i.e. scenario (ii), particularly using the more stringent colocalization prior $p_c = 0.01$ (**Figure 6b**). The FPR was generally low across all scenarios and prior choices: the 1st decile and median values were all zero. However, in scenario (iii), when there are 4 clusters of traits and 4 causal variants in the region, the 9th decile of the FPR increased for both methods, from around zero in scenario (ii) up to 0.16, 0.1 and 0.08 when p_c was 0.05, 0.02, and 0.01, respectively (**Figure 6c**). The increase in FPR in scenario (iii) was a consequence of HyPrColoc occasionally wrongly including an extra trait in one of the clusters (**Figure 6f**), and the pairwise approach overestimating the number of clusters (**Figure 6e**). This was because the causal variants from distinct clusters were in strong LD, i.e. $r^2 > 0.95$, the FPR of both methods reduced when excluding causal variants in strong LD (**Figure S2c**).”

(Branch and bound divisive clustering algorithm, page 12, line 21)

“HyPrColoc significantly outperformed the pairwise approach when summarising results from the clusters of colocalized traits whose posterior probability satisfied $P_R P_A > 0.7$ (**Figures 7a-e; Table S7**). This procedure reflects common practice, as colocalization results are generally only reported when the posterior probability of colocalization is greater than a threshold value, which we take here to be 0.7. Across all three scenarios, clusters of colocalized traits identified by HyPrColoc had a median accuracy and TPR of 1, with little sensitivity to the different choices of colocalization prior p_c . The FPR reduced also, for example in scenario (iii) when $p_c = 0.01$, the 1st, median and 9th deciles of the FPR were all zero. The FPR reduced for the pairwise approach after thresholding, but the TPR reduced as well. In pairwise approaches, a cluster of 3 or more colocalized traits is identified if and only if all pairs of traits colocalize (ideally at the same

shared causal variant), the TPR of the pairwise method reduced after thresholding as only some of the pairs of traits passed the posterior threshold which increased the false negative rate. This is a drawback of methods which do not perform multi-trait colocalization.”

We highlight the importance of assessing sensitivity of results to the specification of priors and that this can/should be performed as part of routine analyses using HyPrColoc. We moreover give examples of HyPrColoc’s inbuilt prior sensitivity analysis tool which can do this automatically (**Figures 6g and S4g**). The manuscript has been edited in several places as follows:

(Branch and bound divisive clustering algorithm, page 13, line 20)

“The computational gains of HyPrColoc make it practical to perform multiple rounds of colocalization analyses, each with different values of the prior p_c and the threshold parameters P_R^*, P_A^* , to assess any sensitivity in the clusters of colocalized traits identified to changes in parameter specifications. An example of this, taken from data generated under scenario (iii), is presented in **Figure 6g**. The resulting heatmap highlights the presence of four clusters of colocalized traits in the data and these clusters persist across most of the prior and threshold parameter settings. We include this sensitivity analysis in the HyPrColoc software and recommend its use.”

(Discussion, page 20, line 24)

“To overcome challenges when specifying the prior probability of a causal configuration, we have suggested two different parsimonious configuration priors (**Methods**). The computational advantages of HyPrColoc make it practical to assess sensitivity of results to the specification of prior and posterior threshold parameters as part of regular use. The HyPrColoc software includes a tool to do this, visualizing any changes to the clusters of colocalized traits identified as parameters are varied.”

For the real-world data example, we selected priors guided by the outcome of our simulation results. This information is now provided in the manuscript as follows:

(Map of genetic risk shared across CHD and related traits page 19, line 8):

“In our analyses we set $p_c = 0.02$, i.e. $\gamma_u = 0.98$, and report only the clusters of traits whose posterior probability of colocalization was greater than 0.7. We assessed sensitivity to our choice of colocalization prior p_c , repeating analyses with $p_c = 0.01$, and found no appreciable difference in the clusters identified (results not reported).”

<

B2. On one causal SNP assumption. I take some issues on the authors' mathematical reasoning on the potential consequence of the assumption violation. It is probably true that "The number of non-colocalization configurations is in general much larger than colocalization configurations." However, I am not sure that those configurations are correctly counted in the coloc-type of calculation. Specifically, I worry the normalizing constant for the

colocalization probability can be systematically *under-estimated*. Although I don't doubt the results from the new simulations, I am not entirely convinced by the authors' reasoning. The authors might very well be correct on this issue, but a more detailed mathematical argument is needed to justify the claim (some formulas can be helpful).

Reply > Our apologies for any confusion. In our new simulations we (i) vary sample size between studies (ii) assess different patterns of colocalization and; (iii) vary the number of causal variants per trait (and include traits with no causal variants in the region). The results from these assessments, provided in the manuscript, offer convincing evidence that HyPrColoc controls the false positive rate (FPR) and therefore does not systematically underestimate the normalising constant for the clusters of colocalized traits it reports. This is particularly true when considering only the clusters of colocalized traits whose posterior probability of colocalization is > 0.7, c.f. **Figures 7c and S6c**. In this situation, when more than one causal variant is present, the 1st median and 9th decile of the FPR of HyPrColoc are approximately zero **Figure S6c**.

We hope that the results from the new simulation studies help convince the reviewer that HyPrColoc does not systematically under-estimate the normalizing constant. As requested, we provide some additional mathematical details at the end of this document (**Appendix**), detailing some of our reasoning as to why in multi-trait colocalization assessments are more likely to return conservative results in the presence of violations of the single causal variant assumption. We include the information presented in the Appendix in the Supplementary Material as well (**Supplementary Material**, Page 44, line 21).

<

B3. On evaluation of classification errors. Based on the authors' argument that the proposed method is robust to model misspecification. It seems necessary to separately evaluate false positives and false negatives. The current measure of accuracy makes it extremely difficult to validate the authors' own point on robustness. Presenting standard measures like realized false-discovery rate *and* power should suffice.

Reply > We agree with the reviewer. We now present the results in terms of three classification measures: (i) accuracy; (ii) true positive rate (power) and; (iii) false positive rate.

(**Branch and bound divisive clustering algorithm**, page 11, line 2)

“Here we assess the performance of the branch and bound (BB) divisive clustering algorithm to identify clusters of colocalized traits over a range of scenarios, several specifications of the conditional colocalization prior p_c and using three classification criteria: the accuracy, which is an overall measure of the classification of traits into clusters; the true positive rate (TPR) and; the false positive rate (FPR), see Methods for more details.”

(**Methods**, page 39, line 17)

“Measuring the accuracy, true positive and false positive rates of HyPrColoc

$$Accuracy = \frac{TP + TN}{TP + TN + FP + FN}$$
$$True\ positive\ rate\ (TPR) = \frac{TP}{TP + FN}$$

$$\text{False positive rate (FPR)} = \frac{FP}{FP + TN}$$

where TP and TN denote the true positive count and true negative count, and FP and FN denote the false positive count and false negative count. Hence, accuracy is the proportion of traits that are correctly identified as colocalizing or not colocalizing. To compare HyPrColoc with pairwise methods, we compute the TP, FP, TN and FN rates by aggregating information across all pairs of traits in the sample. A TP is measured when a pair of observations are correctly deemed to colocalize, a FP is measured when a pair of traits are incorrectly identified as colocalizing, a FN is recorded when a pair of traits are wrongly deemed not to colocalize and a TN is recorded when a pair of traits are correctly identified as not colocalizing. When thresholding the posterior probability of colocalization, the TP, FP, TN and FN rates are computed after excluding traits which do not to colocalize with any other trait $P_{APR} > 0.7$."

<

Reviewer #4 (Remarks to the Author):

C0. Foley and colleagues present the first method that is capable of computing colocalization using a large number of traits. This is a novelty because other existing methods cannot practically perform with more than 4 traits.

The method is statistically sound and the derivations are very well described. The authors expose mathematical derivations which are very useful to direct the community for further development on this challenging yet important task.

The authors propose approximating the posterior probability of full colocalization, and the first step includes a novel Bayesian divisive clustering algorithm. I find this the most novel part of the approach. Personally, if the authors find it suitable for this goal, I am wondering whether the method would be better suited to identify regions of the genome important for disease and to prioritize traits, rather than focusing on assessing full colocalization of all traits. However I have some concerns on the behavior of the statistic and the priors, and on the real data application. I describe below in detail.

I hope addressing these could help the practical applications of this method and its use in real data analysis.

Reply We thank the reviewer for their positive and helpful comments.

C1. The first step of the algorithm aims to define the cluster of traits with strongest evidence of regional association, and HyPrColoc uses the regional selection criterion to partition a cluster.

C1 (i). It seems as there is a threshold for increasing power with additional number of traits - above which it is not convenient to add another trait (e.g. in Supplement, from reducing the number of traits from 100 -> 20, there is a ~5% increase in true detection).

Reply > Yes. The limiting factor appears to be the number of alternative models, i.e. models in which all traits do not colocalize. As the number of traits under consideration increases, the alternative model space grows rapidly. Hence, evidence supporting colocalization across all traits must increase at a rate similar to the growth in the number of alternative models.

Despite the drops in true detection, there is still an increase in performance when identifying the shared causal variant, e.g. from 20 -> 100 a ~9% increase in LD between the identified variant and the true causal variant for small studies (N~5k) and a ~3% increase for larger studies (N~20k).

<

C1 (ii). It is also not clear whether using this method with m=2 traits recovers all (or most) of the signals using pairwise colocalization. It seems as there is no advantage over eCAVIAR in performing HyPrColoc with two traits.

Reply > Thank you for flagging this - we now include comparisons between HyPrColoc and pairwise versions of COLOC and eCAVIAR to address this. We note several differences between HyPrColoc and these approaches and have edited the manuscript to reflect these as follows:

(Branch and bound divisive clustering algorithm page 12 line 15)

“Over all scenarios, HyPrColoc regularly identified both the correct number of clusters of colocalized traits in the data (**Figure 6e**) as well as the correct number of colocalized traits within each cluster (**Figure 6f**). The pairwise approach resulted in more variation in the number of clusters identified (**Figure 6e**). HyPrColoc can assign more than a pair of traits to a cluster, allowing information about the location of any shared causal variant to be borrowed across multiple traits, and therefore performed better at identifying the shared causal variant (**Figure 6d**). HyPrColoc significantly outperformed the pairwise approach when summarising results from the clusters of colocalized traits whose posterior probability satisfied $P_R P_A > 0.7$ (**Figures 7a-e; Table S7**). This procedure reflects common practice, as colocalization results are generally only reported when the posterior probability of colocalization is greater than a threshold value, which we take here to be 0.7. Across all three scenarios, clusters of colocalized traits identified by HyPrColoc had a median accuracy and TPR of 1, with little sensitivity to the different choices of colocalization prior p_c . The FPR reduced also, for example in scenario (iii) when $p_c = 0.01$, the 1st, median and 9th deciles of the FPR were all zero. The FPR reduced for the pairwise approach after thresholding, but the TPR reduced as well. In pairwise approaches, a cluster of 3 or more colocalized traits is identified if and only if all pairs of traits colocalize (ideally at the same shared causal variant), the TPR of the pairwise method reduced after thresholding as only some of the pairs of traits passed the posterior threshold which increased the false negative rate. This is a drawback of methods which do not perform multi-trait colocalization.”

(Branch and bound divisive clustering algorithm page 13, line 17)

“In scenarios (i), (ii) and (iii), HyPrColoc identified the clusters of colocalized traits on average 50, 30 and 25 times faster than the pairwise COLOC approach, indicating some sensitivity in computational performance to the type of colocalization structure present in the data. These figures improved to 200, 100 and 75 times faster when analysing 20 traits. The computational gains of HyPrColoc make it practical to perform multiple rounds of colocalization analyses, each with different values of the prior p_c and the threshold parameters P_R^* , P_A^* , to assess any sensitivity in the clusters of colocalized traits identified to changes in parameter specifications. An example of this, taken from data generated under scenario (iii), is presented in **Figure 6g**. The resulting heatmap highlights the presence of four clusters of colocalized traits in the data and these clusters persist across most of the prior and threshold parameter settings. We include this sensitivity analysis in the HyPrColoc software and recommend its use.”

And under violations of the single causal variant assumption we note that

(Violations of the single causal variant assumption page 14, line 25)

“Pairwise eCAVIAR had increased accuracy relative to HyPrColoc and pairwise COLOC”

However, when identifying the shared causal variant HyPrColoc once again outperformed the pairwise colocalization approaches:

(Violations of the single causal variant assumption page 15, line 8)

“By borrowing information between multiple traits, HyPrColoc outperformed eCAVIAR when fine-mapping the shared causal variant (**Figure S4d**) – despite not incorporating LD information. Moreover, in scenario (i) HyPrColoc regularly identified 8 of 10 traits as jointly colocalized; in scenario (ii) 5 out of 6 traits and; in scenario (iii) both clusters of colocalized traits, comprising 5 and 3 traits respectively (**Figure S4f**)”

In addition, HyPrColoc was vastly more computationally efficient than the pairwise multi-causal variant method:

(Violations of the single causal variant assumption page 16, line 6)

“in a region containing around 1,000 SNPs, the single joint colocalization analysis of HyPrColoc was computed approximately 100,000 times faster than the 45 pair-wise analyses of eCAVIAR. The HyPrColoc algorithm can additionally be used to rapidly identify genomic regions and clusters of traits to better prioritize the use of more computationally expensive multi-causal variant colocalization software for pairs of traits (**Supplementary Material**).”

<

C1 (iii). Therefore, I am wondering whether this method would be better suited to prioritize traits and regions to later fine-map. In particular, I am wondering if this model could be used

primarily to find the location of the causal variant in the region, and prioritize traits that are more informative, to then use these in a fine-mapping exercise. This could maximize the strength of this model, since it can test many traits at a time for relevance to disease.

Reply > Given the computational efficiency of HyPrColoc, we agree, a useful application of the algorithm could be to prioritise causal variants and regions for fine mapping in addition to multi-trait colocalization. Based on the reviewer's suggestion, we highlight one such strength and provide details – including example R code - as to how to employ HyPrColoc for these alternative uses (see also our reply to the related question **C4** for more details):

(Violations of the single causal variant assumption page 16, line 8)

“The HyPrColoc algorithm can additionally be used to rapidly identify genomic regions and clusters of traits to better prioritize the use of more computationally expensive multi-causal variant colocalization software for pairs of traits (**Supplementary Material**).”

(Supplementary Material page 58, line 10)

“Step 1 described in the previous section identifies clusters of traits which have overlapping association signals in a genomic region. This procedure is not a formal colocalization analysis. However, it is not constrained by the single causal variant assumption: overlap in association signals can occur because there are one or more causal variants in the region. For example, a pair of traits might have overlapping association signals because: (i) the traits colocalize at a single causal variant; (ii) the traits each have one distinct causal variant and these variants are in strong LD with one another or; (iii) at least one trait has more than one causal variant, some of these causal variants might be shared between traits and/or they are distinct and in strong LD. The approach outlined in step 1 can quickly identify evidence supporting one of these scenarios across multiple traits. For regions and clusters of traits in which (a) an overlap in association signal is detected and (b) HyPrColoc computes that colocalization at a single causal variant is unrealistic, these regions can be re-evaluated using more computationally expensive multi-causal variant software, e.g. eCAVIAR⁸, to assess whether the overlap in association signals can be explained by multiple causal variants per trait.”

Nevertheless, we believe that the primary use of HyPrColoc is in the context of multi-trait colocalization because, as we demonstrate, it can still rapidly identify regions and clusters of traits in which a single causal variant is shared across multiple traits, despite many of these traits having secondary causal variants in the region. Future developments will include extension to multiple causal variants.

<

C2 (i). Realistically, not all traits used in the HyPrColoc analysis will have the same sample size, and in fact will have folds differences one with another (regular sample sizes for GWAS > 20K, but eQTLs <1000). I have a concern on the behavior of the statistics when using unbalanced sample sizes. In particular: when defining the cluster of traits, how different sample sizes will influence the statistics is not clear (e.g. 20 traits of which 10 traits

colocalize, all with different sample sizes, but say it so happens that the traits that do not colocalize have higher sample sizes - and since the clustering algorithm is a sequential decision tree (Figure 3), the combinatorial evidence could be completely mis-calculated due to combination of sample sizes supporting the different configurations).

Reply > We appreciate the reviewers point and have modified the simulation study to accommodate differences in sample sizes between studies. We now allow up to a 20-fold difference in sample size between the largest and smallest studies and edited the manuscript as follows:

(Branch and bound divisive clustering algorithm page 11, line 12):

“To mirror real scenarios in which data are taken from studies with different sample sizes, we take the number of individuals in each study (N_i) as a random draw from the set $N_i \in \{1k, 5k, 10k, 15k, 20k\}$. We compare this with results when all studies have a large sample size by additionally performing an analysis in which $N_i = 15k$ for all traits.”

(Branch and bound divisive clustering algorithm page 11, line 25):

“The median accuracy and TPR is generally ≥ 0.75 , for all three choices of p_c , improving to around 1 when the sample size of each study is large (**Figures 6a-b; Table S6**) - indicating that including studies with smaller sample sizes decreases the TPR.”

<

C2 (ii). possibly a prior that controls for N may be more appropriate here?

Reply > This is a great point. We are currently finalising a theoretical research paper which introduces the divisive algorithm in a general statistical setting. Dirichlet Process (DP) priors have been used in that work and these automatically control for N (the number of traits). However, in the context of a colocalization analyses, we have found that tuning the DP ‘alpha’ parameter, which controls the number and size of clusters, is both difficult and computationally cumbersome. So far, the results indicate that the HyPrColoc extension of the COLOC prior performs much better. Possibly in the future we will have a DP algorithm that is helpful in a colocalization context. <

C3 (i). I share Reviewer 2's concerns on priors, polygenicity, and regional definitions.

The authors provide a nice theoretical explanation of priors, and clarify the explanation of different approaches for priors. In fact, making available a sensitivity analysis pipeline for the prior misspecification is a great contribution from the authors.

Reply > We thank the reviewer for their positive comment. We now include an example of our sensitivity analysis in the main text and highlight its importance:

(Branch and bound divisive clustering algorithm page 13, line 20)

“The computational gains of HyPrColoc make it practical to perform multiple rounds of colocalization analyses, each with different values of the prior p_c and the threshold parameters P_R^* , P_A^* , to assess any sensitivity in the clusters of colocalized traits identified to changes in

parameter specifications. An example of this, taken from data generated under scenario (iii), is presented in **Figure 6g**. The resulting heatmap highlights the presence of four clusters of colocalized traits in the data and these clusters persist across most of the prior and threshold parameter settings. We include this sensitivity analysis in the HyPrColoc software and recommend its use.”

<

C3 (ii). However, I do not understand the link between the priors and the number of causal configuration that the authors describe (i.e., "when two causal variants are present, a non-coloc configuration benefits from this more").

The algorithm (priors) should account for differences in the number of colocalization vs not colocalization scenarios. If it doesn't, there is a problem with the algorithm.

Reply > We apologise for the confusion. The priors do indeed account for differences in the number of colocalization vs non-colocalization scenarios. We now summarise our results in terms of three classification measures (accuracy, TPR and FPR) as well as three choices of the colocalization prior, to highlight that the algorithm performs well across a broad range of measures and multiple choices of causal configuration prior parameters. Please also refer to reply to reviewer 2 comment **B2**.

<

C3 (iv). If we use a method where the variant-level prior is set: say we have two traits, the prior probability of colocalization of independent traits ($p_1 * p_2$) versus related traits (p_{12}), should control for the difference in number of configurations in the coloc versus non-coloc situation.

From what I understood, this should be the function of p and γ in the variant-level prior approach, where γ is the conditional prior probability. Please clarify in the text, along with the practical values recommended for p and γ (possibly with reference to p_1 , p_2 , p_{12}) when the two traits are independent.

Reply > We appreciate this point and have modified the manuscript to make clear that the COLOC prior p_1 is equal to our prior p and that the conditional colocalization prior in COLOC, i.e. $\left(\frac{p_{12}}{p_{12}+p_1}\right)$ is equal to $1-\gamma$. We refer to the conditional colocalization prior as p_c in the main text and use this when comparing pairwise COLOC with HyPrColoc

(Description of the HyPrColoc method, page 6, line 17):

“ p , the probability that a variant is causal for one trait (equivalent to parameter p_1 in COLOC²); γ_r , where $1 - \gamma_r$ is the conditional probability that a variant is causal for a second trait given it is causal for a related trait and; γ_u , where $1 - \gamma_u$ is the conditional probability that a variant is causal for a second trait given it is causal for an unrelated trait

(Methods). Note that, $1 - \gamma_u$ is equivalent to $\left(\frac{p_{12}}{p_{12}+p_1}\right)$ in COLOC². As it will be helpful later, we introduce the conditional colocalization prior p_c , where $p_c = \left(\frac{p_{12}}{p_{12}+p_1}\right) = 1 - \gamma_u$.”

As noted in our reply to comment **B1** from reviewer 2, we now provide guidance on practical values for the priors as well as examples of HyPrColoc's sensitivity analysis tool – which automatically assesses sensitivity of results to different choices of the priors and threshold parameters.

<

C3 (v). On this topic, I found the author's reply to Reviewer 2 on region definition unsatisfactory.

Reviewer 2's question seemed to be about the window size used for a region, possibly because LD blocks are very large and very likely to overlap many causal variants. This is also related to Reviewer 2's concern on polygenicity.

C3 (v). Reply > To further assess issues associated with polygenicity, we now include more simulation scenarios in which multiple traits have more than a single causal variant in the region. As requested by reviewer 2, we provide some additional mathematical arguments at the end of our replies (see **Appendix**). More importantly, however, is that the false positive rate in these new scenarios was modest and zero (for HyPrColoc) when hard-thresholding results to keep only those in which the posterior probability of colocalization was greater than 0.7 (**Figure S6c**). Please also refer to reply to comment **B2** from reviewer 2. <

C3 (vi). If the priors are set at a region-level, and the region is as big as an LD block, it will most likely violate the assumption of one causal variant.

Although the algorithm is robust to this violation, the argument seems to state that the test can be biased if we were attempting to meet the single causal variant assumption, because of an unbalanced proportion of causal and non-causal configurations. The authors reply using the relationship of Q (number of SNPs in the region) and number of causal configurations, which is confusing. There needs to be a distinction between Q , and region definition. The denser the Q for a set region (i.e., better imputed), the better estimates of colocalization we should have because we have a better chance to capture the causal variant in the region.

C3 (vi). Reply > We agree that shrinking a genomic region down to a window in which the single causal variant assumption might be reasonable. We are somewhat cautious about this, however (see our reply to the related comment in **C4**)

We agree with the reviewer, that the denser Q the better the estimate of colocalization will be and so we now include this point in the manuscript:

(**Description of HyPrColoc method** page 5, line 12):

“To increase the probability of identifying any underlying causal variant(s) in the region, the number of SNPs Q included in analyses should be maximised, i.e. the region should be well imputed.”

<

C4. I am very skeptical on the use of LD blocks in real data analysis.

A whole LD block is almost insured to include > 1 causal variant.

I am wondering in fact if the divisive clustering algorithm can be used first to select signals (e.g. +/- 500kb around the most causal SNP) around which to build colocalization analysis, assuming potentially only 1 signal in the region.

Reply > We wanted to give a more general set-up in the applied example that does not necessarily follow a primary trait by using the LD block approach suggested by Berisa and Pickrell (2015) - often users will not have a primary trait. The LD block approach also allows systematic colocalization analyses to be performed genome-wide.

Another reason for defining a region based on LD is to ensure that we include all possible causal variants that are tagged by the sentinel (top) SNP, we appreciate that this could have a knock-on effect of including multiple causal variants. However, we recovered reasonable results in our real data analysis – reproducing many known results. Our software allows users to define their own regions for analyses, so alternative region definitions including smaller sliding windows could be employed.

The reviewer's strategy of first selecting signals can indeed be employed by users and we now provide example R code should users wish to do this (**Supplementary Material** page 56, line 16). We have modified the manuscript as follows:

(**Discussion** page 20, 16)

“We note that the reliability of results under the single causal variant assumption only break down when secondary causal variants explain as much trait variation as the shared variant (**Supplementary Material**). An example of which is the expression of *SH2B3*, where multiple causal variants for the expression of this gene masks colocalization with the CHD signal, we discuss an approach to building colocalization analyses which might help support the single causal variant assumption (**Supplementary Material**).”

(**Supplementary Material** page 56, line 16)

“Here we discuss a two-step procedure which might help increase the possibility of having one causal variant per trait. Step 1; instead of running a full colocalization assessment for a given genomic region, we first run the divisive algorithm using the “reg.only” assessment - which identifies clusters of traits with overlapping association signals, via the regional probability P_R (**Methods**), and identifies candidate SNPs which explain the overlap. Step 2; we separate the sample of traits into the clusters of traits identified in step 1 and run a full colocalization analysis across all traits within each cluster, we do this in a region of size $\pm X$ kb (where X is user defined) around the candidate SNP identified in step 1. Step 1 has $\mathcal{O}(mQ)$ computational complexity and hence can rapidly scan a region for clusters of traits which show potential in colocalizing. We provide example code below, using the test data available with the HyPrColoc software. In this example the same clusters of traits and candidate causal SNPs are identified using both the two-step approach and the single full colocalization analysis. We recommend exerting some caution with the two-step procedure – users should necessarily assess sensitivity of results to the number of SNPs (i.e. choice of X) included in analyses for each of the clusters. HyPrColoc's sensitivity analysis pipeline can be modified to accommodate this.”

C4 (ii). Additionally, it looks like there is inconsistency in real data (using LD blocks) and simulations. In simulations, regions are defined around a CHD signal. What is the mean Q in simulations? Region size?

Reply > Yes, in our simulation studies we did not use LD blocks. Instead, we generated realistic LD patterns by resampling phased haplotypes from the European samples in 1000 Genomes and randomly choosing one of the first 50 regions confirmed to be associated with CHD. We now include some summaries of these regions and the number of SNPs in the manuscript:

(**Methods** page 38, line 14):

“After removing variants with low MAF, i.e. $MAF < 0.05$, the number of SNPs analysed in these regions ranged from 228, in the APOE region, to a maximum of 1918 SNPs in the PDGFD region. The mean number of SNPs was 881.6.”

<

C4 (iii). Moreover, the region definition was changed when analyzing eQTL and pQTL (page 15). This is confusing. Indeed if the aim is to find a potential gene to build a drug target to disease, it makes sense that the region definition is centered around a gene. On the other hand, setting the region definition to around a GWAS signal also makes sense since we are interested in testing colocalization of the GWAS signal with other molecular traits. If both steps are included (centered on the LD blocks or around cis-eQTLs/pQTLs), please describe how many regions are discordant using the two approaches of region definition.

Reply > The analysis was performed using the same LD blocks as before. We additionally provide the information on the eQTL and pQTL data available in the region. For eQTL data we had 1Mb either side of the gene, which usually covers the LD block, for pQTL we used 5Mb which always covers the LD block. We clarify this in the manuscript as follows:

(**Methods** page 41 line 15)

“To prioritise candidate causal genes in regions where CHD and at least one related trait colocalized, we re-ran the colocalization analysis and included whole blood cis eQTL (31,684 samples) and cis pQTL (3,301 samples) data in addition to the primary traits in a second step, using the same LD blocks as before. A colocalization analysis was performed for every transcript with data within each region. *cis* eQTL were defined 1MB upstream and downstream of the centre of the gene probe (1,828 genes were analysed across the 43 regions). *cis* pQTL were defined 5MB upstream and downstream of the transcript start site (854 proteins were analysed across the 43 regions).”

<

Appendix

Reply B2: Some further mathematical details:

Let m denote the number of traits and Q be the number of SNPs. Under the single causal variant assumption, the non-colocalization configurations that are important to HyPrColoc are: (i) the

$$\left(\binom{m}{m-1}Q\right) * (Q - 1) = mQ(Q - 1) = \mathcal{O}(mQ^2)$$

causal configurations which make up the number of ways that all traits except one share a causal variant, at one of the Q locations, the remaining trait has a causal variant elsewhere, i.e. at one of the $Q - 1$ remaining SNPs, and; (ii) the

$$\binom{m}{m-1}Q = mQ$$

causal configurations which make up the number of ways that all traits except one share a causal variant, the remaining trait does not have a causal variant.

Suppose there are 2 causal variants per trait, and further that these are shared across all traits. To help initially, we ignore LD. Combining the non-colocalization configurations above there are

$$2m + 2m = 4m$$

non-colocalization configurations which visit the two causal variants. This notably grows as a function of the number of traits. Compare this with the Q possible configurations in which all traits colocalise at a single causal variant. Only 2 of these configurations will visit the two causal variants (which does not vary in the number of traits m). Thus, there are

$$\frac{4m}{2} = 2m$$

fold more non-colocalization configurations - which visit the two causal variants - relative to colocalization configurations. If we introduce some LD, say two additional variants are in near perfect LD with the two causal variants, then the relative difference is now

$$\frac{(4 + 12)m}{4} = 4m.$$

That is, the relative difference has doubled in favour of the non-colocalization configurations. Adding more traits and/or causal variants (and accounting for LD between variants) increases the relative difference in favour of non-colocalization configurations owing to their mQ^2 growth. This helps to explain why multi-trait colocalization (under a single causal variant assumption) can provide some safeguards against misspecification. Notably, if we additionally report only the results of clusters of colocalized traits whose posterior probability is above a threshold, e.g. $P_R P_A > 0.7$, then this provides further safeguards against false positives (**Figure S6c**).

As a sensitivity analysis, we allow users to increase the number of non-colocalization configurations computed so that there are $m^x Q + mQ^2$ folds more non-colocalization configurations (users can choose x and we give an example of this in the vignette). We found no difference in our results when increasing x beyond the default $x = 1$.

Reviewer #1 (Remarks to the Author):

The authors have answered all my raised concerns.

Reviewer #2 (Remarks to the Author):

Although the authors generally acknowledged the issues raised by other reviewers and me, I am not sure if their responses sufficiently resolve these issues. It should be noted that fast computational/processing time is not equivalent to computational efficiency, for which the necessary pre-requisite is accuracy and correctness.

1. On prior specification. I appreciate that the authors made an effort to re-write and explain their prior parametrization. However, by setting a single set of p and p_c values for all traits analyzed, the authors implicitly assume that all traits are exchangeable. This seems to be problematic. Considering a colocalization analysis of standing heights and type II diabetes (for example), the values of p for the two traits should be very different in magnitude based on the marginal association analysis. The same arguments apply to p_c . I agree with the authors that applying a sensitivity analysis for a range of p_c values. But it is also important to address the scenario in which large discrepancies of colocalization results emerge from the sensitivity analysis. Because the sensitivity analysis may be helpful to highlight the problem, it does not provide a useful solution. The fundamental question is what a user should do if the colocalization result is indeed sensitive to the prior? I wonder if the authors can provide some natural and interpretable "conservative" prior settings akin to what is used in eCAVIAR? I am also puzzled by the following statement in the authors' response: "For the real-world data example, we selected priors guided by the outcome of our simulation results. " Why? Does any grand truth from practice inform the simulations? The setup used in the simulations seems rather arbitrary to me. Overall, I don't think the authors provide a principled way to help practitioners set up the critical priors for the proposed analysis. I am especially worried about the "default" parameters without context-specific consideration may lead to severe misuses of the proposed method.

2. On one causal SNP assumption. Although I am not *convinced* that the assumption does not lead to systematic underestimation of the normalizing constant, I am willing to be persuaded that such an assumption may not lead to severe inflation of false-positive findings based on the added simulations. That said, the authors' solution to evaluating FP and FN findings is limited to the simulation setting. I wonder that in practice if there is a way to *estimate* the false discovery rate/number? Is there a pre-established alpha level that the user can examine the potential false discovery rate of the identified colocalization sites. I think this feature is more critical for practitioners.

3. On added simulation studies. I take some issues on the added simulations to compare the performance of HyPrColoc and eCAVIAR. The comparison does not seem to be fair. First, as the authors noted, CLPP from eCAVIAR is for SNP-level only, whereas the colocalization probability reported by HyPrColoc refers to the whole genomic region; Second, there is a quite significant discrepancy in prior specifications between the methods. It is unclear that the comparison is at the equal footing. So I am not sure if the claims based on these simulations are completely valid.

Reviewer #4 (Remarks to the Author):

Foley and colleagues have thoroughly addressed all concerns about the proposed method with extensive simulations under additional scenarios. The method is well supported under many situations, and results are consistently compared with other methods.

My only comment is to clean up the main Figures.

Below you can find my suggestions. Apart from these minor cleaning suggestions, I recommend this article for publication.

- Figure 3 could go to Supplementary.
- Figure 4 panel B Line 946: maybe include the 5th trait also in the second panel for consistency with the first panel Just a suggestion. Is "Posterior Probability" what is called "PPFC" in HyperCOLOC? I would clarify in Figure or legend.
- Figure 5 only choose one sample size or one variance explained across sample sizes and place the rest in Supplementary.
- Figure 6 is way too cluttered and hard to follow.

I would split this into two Figures at least.

One Figure to include the first and second rows, with Accuracy, FP, TP and LD with comparison with pairwise COLOC.

Other Figure to include the "Number of clusters" (current Figure 6e without the COLOC pairwise comparison, since this is outside the scope of COLOC anyway), and "Number of Traits" (current Figure 6f, adding a greater gap across the clusters to reflect these are subgroups of Figure 6e), computed versus theoretical.

Possibly move Figure 6g to Supplementary.

- Figures 6a-e and Figures 7a-e convey very similar information. I would choose one only and move the other to Supplementary.

- Figure 8: what is "Colocalization posterior explained"? Is "PPFC" in HyperCOLOC? I would clarify in Figure or legend.

Line 247-248: "We compare this with results when all studies have a large sample size by additionally performing an analysis in which $N = 15k$ for all traits." -> reference to Supp Figure missing.

 EDITOR'S NOTE: This reviewer was also asked to comment on Reviewer 2's comments, please see Reviewer 4's additional comments below dashed line.

Regarding point 1 (On prior specification): I agree with this reviewer that the text about the prior specifications in relation to the correlation across traits (in the first section "Description of the HyPrColoc method") needs to be clarified: it is not clear if the traits are exchangeable, since the authors describe their method as "allowing prior information about the relatedness of traits to be incorporated", then describe three types of priors:

p -> "probability that a variant is causal for one trait (equivalent to parameter p_1 in COLOC)";

γ r -> related traits: when is this actually used?

γ u -> why not just have the user specify p_{12} as in COLOC and compute the $p_{12}/(p_{12}+p_1)$?

If the correlations across traits is actually never used in the main software, this formulation is confusing.

In Supplementary, it is clear that the authors have explored behavior of the method in the presence of correlation of traits. Also, the authors have tested their method under realistic context to help choose priors (and it is well described in Supplementary). I do not believe there are further analyses to be done. However the formulation of the prior settings could be simplified. Also, the text should describe how the correlation across traits can influence the prior and posterior, and whether this is accounted for (are the traits actually "exchangeable"?). The text needs to be clarified to give the user a better intuition of what the default parameters actually mean.

Regarding point 2 (On one causal SNP assumption): In cases where there are multiple variants, the posteriors will vary depending on LD between the causal variants and trait variance explained. Unfortunately, with only summary statistics (no LD information or raw data), I do not think this issue can be solved easily with a pre-established value any further than what the authors have done already using FP/FN under realistic simulation scenarios.

Regarding point 3 (On added simulation studies): I agree with the reviewer that the goal of this section needs to be clarified and possibly split into two sections (see below).

It is unclear whether the goal here is to compare methods or compare settings under multiple causal variants, and the author's statement "the CLPP of eCAVIAR is not equivalent to the posterior probability of HyPrColoc and COLOC" can be easily misinterpreted. I don't think the authors were referring to per-SNP versus regional probability (and actually the regional colocalization can be translated easily in SNP-level probability), rather, the way that the CLPP is computed in the presence of multiple causal variants and by taking account of the LD structure.

I suggest to split the paragraph in two:

One paragraph, "Violations of the single causal variant assumption", describing results from simulations of multiple causal variants;

a separate paragraph could describe results of the comparison of HyPrColoc with existing methods, COLOC and eCAVIAR (and now eCAVIAR can be directly compared to COLOC when assuming at most one shared causal SNP in each region).

REVIEWER COMMENTS

Comments to all reviewers:

We would again like to express our thanks to all reviewers for their comments. Changes to the main text manuscript and supplementary material are marked in yellow and we provide the locations of any edits made to the manuscript (**section**, page number, line number) and supplementary material (**Supplementary Material**, page number, line number) herein. Our replies to the reviewer's comments are placed below the comment and within angled brackets > <. We have additionally itemized the reviewer's comments, e.g. **B0-B3**, for clarity.

Based on feedback from both reviewers, we have removed from the main text the statements about setting a configuration prior based on the relatedness of traits. Specifically, we have removed all references to the parameters $\{\gamma_w, \gamma_r\}$ from the main text and methods. To avoid confusion, we instead express our prior framework in terms of the prior parameters $\{p, p_c\}$ as these are the parameters that are referenced and/or tested throughout the text. We have further clarified the connection between these parameters and those used in COLOC, we highlight a benefit of our parsimonious prior framework and flag a potential limitation owing to the assumption that all traits are exchangeable. We have updated the manuscript accordingly:

(**Overview**, page 6, line 13)

"We adopt an approach which requires the specification of a partition of the traits into clusters, together with two interpretable parameters: p , the probability that a variant is causal for one trait; and p_c , the conditional probability that a variant is causal for a second trait given it is causal for one trait (**Methods**). As it will be helpful later, we refer to p_c as the conditional colocalization prior. COLOC² requires specification of three prior parameters $\{p_1, p_2, p_{12}\}$ and, while the scope of the configuration priors in HyPrColoc is different for more than a pair of traits, it is instructive to note that $p \equiv p_i$, for $i \in \{1,2\}$, and $p_c \equiv \left(\frac{p_{12}}{p_{12}+p_1}\right)$ when $m = 2$. To help users of the COLOC² software, our software allows users to specify the parameter p and one of either (i) p_c ; or (ii) p_{12} , from which p_c is computed. For simplicity and as a conservative measure, we assume a priori that the genetic association probability p and the conditional colocalization probability p_c are equal for all traits. This approach allows sensitivity analyses assessing robustness of posterior inference to be routinely performed. However, it implicitly assumes traits are a priori exchangeable, e.g. assumes $p_1 = p_2$; this is supported across a range of designs (case/control or quantitative trait) but may lead to poorer performance in specific datasets⁵⁰."

Reviewer #2 (Remarks to the Author):

We thank the reviewer for their additional comments. We have acknowledged these and now provide:

- A clarification of our use of ‘computational efficiency’.
- Information on the relevant colocalization literature used to guide the design of our prior parameter sensitivity analyses.
- Additional guidance on what to do when results are sensitive to changes in the specification of priors.
- Additional information regarding the specification of the default prior information.

We have edited the manuscript to make clearer our principled approach to both constructing our simulation protocol and assessing robustness of results to changes in the specification of the prior parameters.

B0. It should be noted that fast computational/processing time is not equivalent to computational efficiency, for which the necessary pre-requisite is accuracy and correctness.

Reply > While we appreciate the reviewer’s point, we note that our use of computational efficiency in the manuscript is in keeping with how other authors employ the term. For example, the authors of the widely cited and popular software Stan state that “computational efficiency generally relates to measuring the amount of time or memory required for a given step [or process] in a calculation”. We also note that the HyPrColoc software is calculating the posteriors in line with the description in the manuscript and therefore is an accurate representation of the method (**Figure 3** illustrates that the difference between the posterior probabilities computed by HyPrColoc and MOLOC for an increasing number of traits is $\lesssim .005$). We now clarify our use of the term computational efficiency in the manuscript:

(**Efficient computation of PPFC**, page 7, line 11)

“HyPrColoc overcomes this challenge by approximating $p(D)$ in a way that is both computationally efficient, **i.e. has fast computational time,** and tightly bounds the approximation error.”

<

B1(a). On prior specification. I appreciate that the authors made an effort to re-write and explain their prior parametrization. However, by setting a single set of p and p_c values for all traits analyzed, the authors implicitly assume that all traits are exchangeable. This seems to be problematic. Considering a colocalization analysis of standing heights and type II diabetes (for example), the values of p for the two traits should be very different in magnitude based on the marginal association analysis. The same arguments apply to p_c . I agree with the authors that applying a sensitivity analysis for a range of p_c values. But it is

also important to address the scenario in which large discrepancies of colocalization results emerge from the sensitivity analysis. Because the sensitivity analysis may be helpful to highlight the problem, it does not provide a useful solution. The fundamental question is what a user should do if the colocalization result is indeed sensitive to the prior? I wonder if the authors can provide some natural and interpretable “conservative” prior settings akin to what is used in eCAVIAR?

Reply. > We have split our reply to this comment into three sections: (i) the exchangeability assumption; (ii) the specification of default conservative priors; and (iii) guidance if there is sensitivity to prior specification.

(i) On the exchangeability assumption:

We agree with the reviewer that by “setting a single set of p and p_c values for all traits analyzed,...[we] implicitly assume that all traits are exchangeable.” We moreover agree that in certain analyses results may improve by: (i) introducing non-exchangeable prior parameters; and (ii) making good and robust choices when specifying these additional parameters, e.g. p_1 and p_2 in COLOC. As the reviewer notes, the number of GWAS hits can vary by the particular trait or traits under consideration as well as sample size (e.g. Visscher et al, Am J Hum Genet, 2017). However, using data from over 5000 GWAS studies Wallace, Plos Genetics 2020, deduces that the COLOC (and therefore the HyPrColoc) default prior of $p_1 = p_2 = p = 10^{-4}$ (which assumes exchangeability) is well supported across a range of designs (case/control or quantitative trait, with varying MAF and sample size):

(Wallace, Plos Genetics 2020, Marginal priors, Page 7)

“The default coloc marginal prior of $p_1 = p_2 = 10^{-4}$ is thus supported by the convergence of these three approaches to values of the order of 10^{-4} .”

In the context of multi-trait colocalization, expanding the prior parameter space to account for non-exchangeability introduces some difficulties due to the large number of parameters required, e.g. an extension of the COLOC prior introduces up to 2^m parameters (where m denotes the number of traits, see **Methods**). A large prior parameter space would make any essential assessment into prior sensitivity very difficult. We do acknowledge the reviewer’s point, however, and have now edited the manuscript to highlight HyPrColoc’s exchangeability assumption.

(Description of the HyPrColoc method, page 6, line 22)

“For simplicity and as a conservative measure, we assume a priori that the genetic association probability p and the conditional colocalization probability p_c are equal for all traits. This approach allows sensitivity analyses assessing robustness of posterior inference to be routinely performed. However, it implicitly assumes traits are a priori exchangeable, e.g. assumes $p_1 = p_2$; this is supported across a range of designs (case/control or quantitative trait) but may lead to poorer performance in specific datasets⁵⁰.”

(ii) On the specification of default conservative priors:

Our variant specific configuration prior relies on the specification of two parameters $\{p, p_c\}$ which, as we have shown, are related to prior parameters $\{p_1, p_2, p_{12}\}$ already known in the colocalization literature (Giambartolomei et al, 2014 and 2018; Wallace, 2020). Sensitivity of results to $\{p_1, p_2, p_{12}\}$ has been thoroughly investigated in the literature by Wallace, Plos Genetics 2020, and we make use of these results in our investigation. Our search for generally robust choices of $\{p, p_c\}$ starts from the robust defaults for $\{p_1, p_2, p_{12}\}$ identified by Wallace. We have edited the manuscript to make clearer our principled approach to the specification of the parameters $\{p, p_c\}$:

(Branch and bound divisive clustering algorithm, page 11, line 19)

“Following the approach of Wallace⁵⁰, we assess sensitivity to the choice of colocalization prior p_c , i.e. $(1 - \gamma)$. Across a wide range of simulated data, Wallace⁵⁰ demonstrated that setting $p_{12} = 5 \times 10^{-6}$ in COLOC (approximately $p_c = 0.05$ in HyPrColoc) was generally a robust choice. Starting from this value, we evaluated results with more conservative choices of p_c by performing three separate analyses for each dataset using $p_c \in \{0.05, 0.02, 0.01\}$, equivalent to $p_{12} \approx \{5 \times 10^{-6}, 2 \times 10^{-6}, 1 \times 10^{-6}\}$ with $p = 10^{-4}$ fixed⁵⁰, in order to identify a robust choice of p_c . These values can result in substantial differences in the prior probability of colocalization as the number of traits in a cluster increases (**Methods**).”

In the expected case an investigator does not have a strong belief about values for the configuration priors, Wallace’s new investigation concludes that it is reasonable, in general, to treat the traits as a-priori exchangeable. Wallace then focuses their prior sensitivity analysis to the identification of a robust choice for the prior probability that a variant is causal for both traits p_{12} . Wallace’s approach to analysing the sensitivity of results to the specification of p_{12} motivated our sensitivity analysis of p_c . Wallace identifies $p_{12} = 5 \times 10^{-6}$ as a “generally robust choice”, which equates to $p_c \approx 0.05$ in HyPrColoc. Setting $p_c \approx 0.05$ would therefore seem like a reasonable choice for HyPrColoc, however, further investigation – via our prior sensitivity analyses – led us to identify a “conservative” robust choice of prior $p_c = 0.02$, which is overly conservative for a pair of traits, becoming more conservative as the number of traits grows. We have edited the manuscript to note this:

(Branch and bound divisive clustering algorithm, page 13, line 24)

“The HyPrColoc default $p_c = 0.02$ is equivalent to setting $p_{12} \approx 2 \times 10^{-6}$ which, for a pair of traits, is slightly more conservative than the recommended value of $p_{12} = 5 \times 10^{-6}$ by Wallace⁵⁰. For more than a pair of traits, however, it can be much more conservative, e.g. setting $p_c = 0.05$ (i.e. $p_{12} \approx 5 \times 10^{-6}$) in the variant-level prior returns a prior probability of colocalization across 10 traits that is around 2000 times larger than when setting $p_c = 0.02$ (i.e. $p_{12} \approx 2 \times 10^{-6}$).”

(iii) On sensitivity to prior choice:

We now provide additional guidance for users of HyPrColoc when inference appears to change significantly based on the choice of prior values. In these situations, we encourage investigators to report results from analyses with strong evidence of colocalization, i.e. a posterior >0.7 , and using the conservative default prior setting $p_c = 0.02$. Both Wallace and the HyPrColoc authors found this prior value had a very low false positive rate with diminished power to detect colocalized traits, hence, we are confident that this is generally a conservative choice.

We additionally encourage investigators, if computationally feasible, to investigate the reasons for any observed differences in results between runs which use (i) the default conservative prior of $p_c = 0.02$ and (ii) the more stringent prior choice $p_c = 0.01$. If the differences in the number of traits/clusters-of-traits is modest, between the runs, this might present a computationally practical opportunity to prioritise the use of more expensive multi-causal variant methods like eCAVIAR or ENLOC. For example, in our simulation study in which the single causal variant assumption was violated, traits with two causal variants were dropped from a cluster of colocalized traits when using the more stringent prior despite these traits all sharing a single causal variant (**Figure 7b**). Hence, if the results of the multi-causal variant methods match HyPrColoc's results – implicating the same shared causal SNP or one in near perfect LD – we can reasonably conclude that these traits were dropped using the more stringent prior as a result of HyPrColoc's single causal variant assumption. We note this in the manuscript:

(Violations of the single causal variant assumption, page 16, line 12)

“We provide an illustration of HyPrColoc's sensitivity analysis tool under scenario (iii) (**Figure 7b**) – correctly highlighting the presence of two clusters of colocalized traits. After applying more stringent prior and threshold values, one cluster reduced from 5 traits down to the 3 traits which have and share a single causal variant. This suggests strong evidence of 3 traits and weak evidence of 5 traits in the cluster. While the approach should be tailored to the problem at hand, if the analysis flags considerable sensitivity to the specification of the prior, we suggest: (a) reporting the clusters of colocalized traits identified as colocalizing with $P_R P_A > 0.7$ using the conservative prior setting $p_c = 0.02$; and (b) where computationally practical, running pairwise analyses using a multi causal variant method, e.g. eCAVIAR⁵ or ENLOC⁶, on the traits or clusters of traits which are reported in (a) but are *not* identified as colocalizing with $P_R P_A > 0.7$ using the more stringent prior $p_c = 0.01$ - this may help clarify if traits are being removed from clusters owing to the presence of additional non-shared causal variants, e.g. scenario (iii) (**Figure 7b**), and should therefore be reported.”

B1(b). I am also puzzled by the following statement in the authors' response: “For the real-world data example, we selected priors guided by the outcome of our simulation results.” Why? Does any grand truth from practice inform the simulations? The setup used in the simulations seems rather arbitrary to me. Overall, I don't think the authors provide a

principled way to help practitioners set up the critical priors for the proposed analysis. I am especially worried about the “default” parameters without context-specific consideration may lead to severe misuses of the proposed method.

Reply. > We now clarify that the specification of the prior parameters in our analyses was a consequence of guidance from the colocalization literature. In their analysis, Wallace estimates values for p_1, p_2 , i.e. p , and p_{12} , i.e. p_c , using GteX and GWAS data, concluding that p_1, p_2 tend to around 10^{-4} for common SNPs and genomic windows of ~ 1 mb. The value of $p = 10^{-4}$ was informed from practice and used in our simulations and CHD analysis. Following Wallace’s approach, we fixed the parameter p , i.e. $p_1 = p_2 = p = 10^{-4}$, to identify a generally robust choice of the parameter p_c via our simulation study – identifying an overly conservative setting relative to Wallace’s findings. We note that our simulation protocol is consistent with the simulation protocols used in two recent contributions to the colocalization literature: (i) Hukku et al; arXiv 2020; and (ii) Wallace; Plos Genetics, 2020 – simulating colocalization and non-colocalization scenarios which are designed to simultaneously investigate false and true positive findings. We have edited the manuscript to note this:

(**Branch and bound divisive clustering algorithm**, page 11, line 10)

“Scenarios (ii) and (iii) are designed to simultaneously investigate potential false and true positive findings.”

On the use of priors, we have clarified this in the manuscript (see response to reviewer 2’s comment **B1(a)**).

<

B2. On one causal SNP assumption. Although I am not *convinced* that the assumption does not lead to systematic underestimation of the normalizing constant, I am willing to be persuaded that such an assumption may not lead to severe inflation of false-positive findings based on the added simulations. That said, the authors' solution to evaluating FP and FN findings is limited to the simulation setting. I wonder that in practice if there is a way to *estimate* the false discovery rate/number? Is there a pre-established alpha level that the user can examine the potential false discovery rate of the identified colocalization sites. I think this feature is more critical for practitioners.

Reply. >

The reviewer raises a very good point, estimating an expected FDR for analyses would be useful. However, an approach to do this would require very careful quantification/investigation of the relationship between an (a-priori) alpha-level and, e.g., the number of traits being analysed; the size of the genomic region; the expected number of clusters of colocalized traits; and the expected number of traits within each cluster – reviewer 4 also notes the role of LD between the causal variants and trait variance explained.

Such an investigation is deserving of a research paper in its own right. We are confident that our default prior choices are generally robust and – should our prior sensitivity analysis tool highlight issues – we have provided investigators with reasonable guidance on what to do: (i) be conservative and (ii) if computationally feasible, investigate why traits are removed from clusters (between different specifications of p_c) using alternative methodologies. Moreover, as reviewer 4 notes we “do not think this issue can be solved easily with a pre-established value any further than what the authors have done already using FP/FN under realistic simulation scenarios.”

<

B3. On added simulation studies. I take some issues on the added simulations to compare the performance of HyPrColoc and eCAVIAR. The comparison does not seem to be fair. First, as the authors noted, CLPP from eCAVIAR is for SNP-level only, whereas the colocalization probability reported by HyPrColoc refers to the whole genomic region; Second, there is a quite significant discrepancy in prior specifications between the methods. It is unclear that the comparison is at the equal footing. So I am not sure if the claims based on these simulations are completely valid.

Reply. > As suggested by reviewer 4, we have split the section *violations of the single causal variant assumption* into two paragraphs: the first paragraph (**page 15, line 23**) reviews HyPrColoc’s performance when the single causal variant assumption is violated; the second paragraph (**page 17, line 5**) summarises HyPrColoc’s performance relative to the pairwise methods of COLOC and eCAVIAR. In paragraph two, we have edited the manuscript to make clear that HyPrColoc’s/COLOC’s posterior probability measure and eCAVIAR’s CLPP measure are computationally distinct:

(Violations of the single causal variant assumption, page 17, line 6)

“We note that the per-SNP CLPP measure of eCAVIAR is computed in the presence of multiple causal variants and is distinct from the per-SNP probabilities, computed under a single causal variant assumption, which make up the posterior probability measure used to summarise HyPrColoc and COLOC – making comparisons between the methods challenging. We compare the methods as they are used in practice, summarizing HyPrColoc and COLOC using the posterior probability of the hypothesis that a cluster or a pair of traits colocalize^{2,8,50} and summarizing eCAVIAR using the SNP-level CLPP. Our choice of CLPP cut-off of 1% was shown to have a low FPR across a range of scenarios previously⁵.”

We chose to compare HyPrColoc, COLOC and eCAVIAR, by summarizing performance of the methods as they are used in practice e.g. by using the default priors as suggested by the authors – which would be the typical implementation. Evidence for each of the 5 interpretable hypotheses (which exhaustively describe region-wide association patterns in terms of either a colocalization mechanism or multiple non-colocalization mechanisms) is what are overwhelmingly reported by users of COLOC (Wallace; Plos Genetics, 2020) - users of HyPrColoc are doing the same (in a computationally scalable way). As the reviewer

notes, eCAVIAR computes SNP level probabilities (i.e. the CLPP) and this is what is reported by users of eCAVIAR.

To make a comparison between the methods, we specify reasonable thresholds for the respective probability measures of each method – beyond which users might conclude there is evidence of colocalization between the traits. In the case of eCAVIAR we use a CLPP cut-off of 0.01 – this cut-off was shown by Hormozdiari et al (2016) to have a low false positive rate and reasonable true positive rate (TPR) across a variety of simulations. We replicate these results in our simulations, giving some confidence to our choice of CLPP cut-off. As expected, in the presence more than one causal variant per trait, our results reveal that eCAVIAR outperforms the single causal variant methods in terms of accuracy and TPR:

(Violations of the single causal variant assumption, page 17, line 13)

“Our choice of CLPP cut-off of 1% was shown to have a low FPR across a range of scenarios previously⁵. In our analyses we found that pairwise eCAVIAR had increased accuracy relative to HyPrColoc and pairwise COLOC, e.g. in scenario (i) median accuracy improved by as much as 0.15 (when sample sizes varied) and 0.2 (when sample sizes were large) (Figure S4a; Table S8). Broadly, this was a result of the single causal variant methods having a lower TPR (Figures S4a-b).”

From our perspective, the important result is that, unlike COLOC, HyPrColoc can return better ‘fine-mapping’ results than a pairwise multi-causal variant method. This result was not unexpected: borrowing strength across multiple traits through a single joint analysis will – if done correctly - necessarily lead to better fine-mapping results than methods which restrict analysis to pairwise assessments. Our result is interesting because we have demonstrated that a multi-trait method, constrained to assume a single causal variant, can – in certain situations – outperform a pairwise multi-causal variant method, despite violations of the single causal variant assumption and not accounting for LD information. We have robustly demonstrated the existence of such scenarios:

(Violations of the single causal variant assumption, page 17, line 18)

“However, by borrowing information between multiple traits HyPrColoc outperformed eCAVIAR when fine-mapping the shared causal variant (Figure S4d) – despite not incorporating LD information. After thresholding the posterior to $P_R P_A > 0.7$, HyPrColoc again outperformed pairwise COLOC (Figure S6a-c).”

On reviewer 2’s comment that there is “quite significant discrepancy in prior specifications between the methods”, eCAVIAR is computationally too expensive to perform routine prior sensitivity analyses when analysing large numbers of traits via pairwise analyses. We therefore limited our assessment to the software defaults.

<

Reviewer #4 (Remarks to the Author):

Foley and colleagues have thoroughly addressed all concerns about the proposed method with extensive simulations under additional scenarios. The method is well supported under many situations, and results are consistently compared with other methods.

My only comment is to clean up the main Figures.

Below you can find my suggestions. Apart from these minor cleaning suggestions, I recommend this article for publication.

Reply. > We thank the reviewer for their helpful comments. Please find our responses to the additional comments in section '**Additional comments**', found below comment **C9**.

<

C1. Figure 3 could go to Supplementary.

Reply. > As suggested, we have placed the figure into the supplementary (c.f. supplementary **Figure S1a**).

<

C2. Figure 4 panel B Line 946: maybe include the 5th trait also in the second panel for consistency with the first panel Just a suggestion.

Reply. > Running MOLOC for 5 traits in a region of ~1000 SNPs took around 1hr per dataset whereas for 4 traits it took ~1 minute. This limited our use of MOLOC (when analysing 5 traits) to the analysis of a handful of datasets to summarise computational performance only. We have clarified this in the legend:

(**Figure 3**, page 44, line 16)

“MOLOC was restricted to $M \leq 5$ traits owing to the computational and memory burden of the MOLOC algorithm when $M > 5$. When $M = 5$, we summarise the computation time of MOLOC from 10 datasets - as it took around 1 hour to analyse a single dataset, in all other scenarios performance was summarised from 1000 datasets.”

<

C3. Is "Posterior Probability" what is called "PPFC" in HyperCOLOC? I would clarify in Figure or legend.

Reply. > Yes, PPFC stands for posterior probability of full colocalization. We have edited **Figure 3** (right panel) to include PPFC and have edited the legend to reflect this:

(**Figure 3**, page 44, line 22)

“Distribution of the posterior probability of colocalization between all traits, i.e. the posterior probability of full colocalization (PPFC)”

<

C4. Figure 5 only choose one sample size or one variance explained across sample sizes and place the rest in Supplementary.

Reply. > We appreciate the reviewer's suggestion, however we feel this plot is important in its entirety as it simultaneously conveys subtle differences in performance according to influences in sample size vs increases in variation explained by a causal variant – it is similar in spirit to Wallace's (Plos Genetics, 2020) distribution of expected posterior probabilities Fig 3.

To help readers, we have clarified the main comparisons to consider in the figure legend.

(**Figure 4**, page 45, line 10)

“Comparing performance across increasing study sample size and variance explained by the causal variant, power to detect all colocalized traits is reduced when including studies with smaller sample sizes (top row), however including these studies can still boost the probability of correctly identifying the shared causal variant irrespective of variance explained (middle row).”

<

C5. Figure 6 is way too cluttered and hard to follow.

I would split this into two Figures at least.

One Figure to include the first and second rows, with Accuracy, FP, TP and LD with comparison with pairwise COLOC.

Other Figure to include the "Number of clusters" (current Figure 6e without the COLOC pairwise comparison, since this is outside the scope of COLOC anyway), and "Number of Traits" (current Figure 6f, adding a greater gap across the clusters to reflect these are subgroups of Figure 6e), computed versus theoretical.

Reply. > We have followed the reviewer's recommendations for Figure 6 (in the latest version of the manuscript this is now supplementary **Figures S2**). We have split the figure into two separate figures: **Figure S2**, which includes Accuracy, FP, TP and LD comparison with COLOC; and **Figure 5**, which includes the number of clusters and number of traits within each cluster identified by HyPrColoc. We have taken the same approach to the similarly arranged Figure 7, which is now main text **Figure 6**.

In terms of adding a greater gap – we have introduced a partition in the plot to make clear the distinctions between results from the 3 scenarios (see **Figure 5b**).

<

C6. Possibly move Figure 6g to Supplementary.

Reply. > We have combined Figure 6g and supplementary Figure S5g into a new figure which illustrates results from the prior sensitivity analysis tool (**Figure 7**). Reviewer 2 was very keen that we provide clear guidance around sensitivity analyses to investigators. Therefore, we now include more guidance on the interpretation of these analyses and as a

result we feel that having a figure that illustrates results from sensitivity analyses in the main text is very useful. We would therefore prefer to keep this figure in the main text.
<

C7. Figures 6a-e and Figures 7a-e convey very similar information. I would choose one only and move the other to Supplementary.

Reply. > As suggested, we have retained figure 7 (now main text **Figure 6**) and have moved what was Figure 6 to the supplementary (c.f. supplementary **Figure S2**). These two figures highlight an important distinction between pairwise and multi-trait analyses and we now clarify this in the legend of **Figure 6**.

(**Figure 6**, page 46, line 11)

“The results highlight that on increasing the posterior threshold from 0.5 (c.f. supplementary **Figure S2**) to 0.7, HyPrColoc’s ability to cluster multiple traits together demonstrably improves accuracy and the true positive rate relative to pairwise analyses.”

<

C8. Figure 8: what is "Colocalization posterior explained"? Is "PPFC" in HyperCOLOC? I would clarify in Figure or legend.

Reply. > We apologise for the confusion, we have clarified what we mean in the legend of Figure 8:

(**Figure 8**, page 47, line 5)

“HyPrColoc identified rs713782 as a candidate causal variant explaining the shared association signal between CHD and the 5 related traits. The posterior probability of colocalization between the traits was 0.909 and rs713782 explained over 76% of this, i.e. the posterior probability of rs713782 being the shared causal variant is $0.909 \times 0.76 = 0.69$. The next candidate variant explained < 20%.”

<

C9. Line 247-248: "We compare this with results when all studies have a large sample size by additionally performing an analysis in which $N = 15k$ for all traits." -> reference to Supp Figure missing.

Reply. > We apologise that our explanation was unclear, these results are presented in the main figures along with results when the sample size varies between studies. We have amended the text to clarify this:

(**Branch and bound divisive clustering algorithm**, page 11, line 16)

“For comparison, we additionally present results when all studies have a large sample size by also performing an analysis in which $N_i = 15k$ for all traits.”

<

Additional comments from reviewer 4

Additional comment 1

Regarding point 1 (On prior specification): I agree with this reviewer that the text about the prior specifications in relation to the correlation across traits (in the first section "Description of the HyPrColoc method") needs to be clarified: it is not clear if the traits are exchangeable, since the authors describe their method as "allowing prior information about the relatedness of traits to be incorporated", then describe three types of priors:

p -> "probability that a variant is causal for one trait (equivalent to parameter p_1 in COLOC)";

γ r -> related traits: when is this actually used?

γ u -> why not just have the user specify p_{12} as in COLOC and compute the $p_{12}/(p_{12}+p_1)$?

If the correlations across traits is actually never used in the main software, this formulation is confusing.

Reply > We appreciate the reviewer's point and apologise that our previous formulation was confusing. As per our comment to both reviewers at the beginning, to remedy this we have re-written our prior setup in terms of the two non-exchangeable parameters $\{p, p_c\}$ only. These are the parameters we use in our main and sensitivity analyses, and we also provide guidance to investigators regarding their specification and use.

As the reviewer suggested, we now additionally allow users to specify p_{12} in HyPrColoc and have edited the manuscript to highlight this:

(Description of the HyPrColoc method, page 6, line 20)

"To help users of the COLOC² software, our software allows users to specify the parameter p and one of either (i) p_c ; or (ii) p_{12} , from which p_c is computed."

Additional comment 1 continued:

In the Supplementary, it is clear that the authors have explored the behavior of the method in the presence of correlation of traits. Also, the authors have tested their method under realistic context to help choose priors (and it is well described in Supplementary). I do not believe there are further analyses to be done. However, the formulation of the prior settings could be simplified. Also, the text should describe how the correlation across traits can influence the prior and posterior, and whether this is accounted for (are the traits actually "exchangeable"?). The text needs to be clarified to give the user a better intuition of what the default parameters actually mean.

Reply > We thank the reviewer for their positive comments and their suggestion to make clearer our findings in the supplement. We have taken their advice and have updated the main manuscript, summarising our findings to highlight how correlations between traits might affect posterior inference:

(Branch and bound divisive clustering algorithm, page 14, line 16)

“We further tested the algorithm using a variety of thresholds $\{P_R^*, P_A^*\}$ and two different prior frameworks (**Figures S9-S10**). We also assessed results in the presence of correlated traits and overlapping samples (**Supplementary Material**). We analysed these data in three ways: (a) ignoring all correlation, i.e. wrongly assuming non-overlapping participants between pairs of studies and ignoring known trait correlation when setting the configuration prior probabilities; (b) adjusting for correlation between the summary data in the computation of the likelihood only; and (c) adjusting for correlation in the computation of the likelihood and accounting for known trait correlation when setting the configuration prior probabilities. Our findings suggest that analyses which account for correlation in the computation of the likelihood should *also* account for any known trait correlation in the configuration prior probabilities: the posterior probability of colocalization between the truly colocalized traits in scenario (b), which ignored known correlation when setting the configuration prior, was significantly smaller than in scenario (c) – leading to a single large cluster of colocalized traits being split into smaller clusters (**Figure S11 and Table S2**). Our results indicated that scenario (a), i.e. ignoring all correlation by treating studies as independent and traits as a-priori exchangeable, even when there is complete sample overlap (i.e. participants are the same in all studies), gives reasonable results and in our assessment was comparable to scenario (c) (**Figure S10 Tables S2-S3**). We discuss the theoretical reasons for this in **Supplementary Material**.”

While we have not tested a situation in which some subsets of traits are related, other subsets are unrelated, we discuss how investigators might account for such a scenario at the level of the configuration priors in the supplementary material (**Supplementary Material, Accounting for non-exchangeability between traits in the configuration prior, page 20**). We have edited the manuscript to highlight this:

(Methods, page 36, line 4)

“If two or more traits in a cluster are known to be related, this information would ideally be included in analyses and we outline an extension to our prior setup which allows for non-exchangeability of traits to be included (**Supplementary Material**).”

Additional comment 2:

Regarding point 2 (On one causal SNP assumption): In cases where there are multiple variants, the posteriors will vary depending on LD between the causal variants and trait variance explained. Unfortunately, with only summary statistics (no LD information or raw data), I do not think this issue can be solved easily with a pre-established value any further than what the authors have done already using FP/FN under realistic simulation scenarios.

Reply > We agree with the reviewer, this is a difficult task which cannot be easily solved more than what we have already done. We acknowledge this in our reply to Reviewer 2's comment **B2**.

<

Additional comment 3:

Regarding point 3 (On added simulation studies): I agree with the reviewer that the goal of this section needs to be clarified and possibly split into two sections (see below). It is unclear whether the goal here is to compare methods or compare settings under multiple causal variants, and the author's statement "the CLPP of eCAVIAR is not equivalent to the posterior probability of HyPrColoc and COLOC" can be easily misinterpreted. I don't think the authors were referring to per-SNP versus regional probability (and actually the regional colocalization can be translated easily in SNP-level probability), rather, the way that the CLPP is computed in the presence of multiple causal variants and by taking account of the LD structure.

I suggest to split the paragraph in two:

One paragraph, "Violations of the single causal variant assumption", describing results from simulations of multiple causal variants;

a separate paragraph could describe results of the comparison of HyPrColoc with existing methods, COLOC and eCAVIAR (and now eCAVIAR can be directly compared to COLOC when assuming at most one shared causal SNP in each region).

Reply > We are very grateful for the reviewer's suggestion, which has helped to clarify our findings when the single causal variant assumption is violated as well as helping to deal with the difficult situation of making comparisons between methods which summarise findings differently. We have actioned the reviewer's suggestions, please see our reply to reviewer 2's comment **B3**. <

Reviewer #2 (Remarks to the Author):

I appreciate the authors carefully respond to my previous comments. I greatly value this scientific discussion and am grateful for the authors' efforts. As I will recommend acceptance to the editor after reviewing the revised manuscript, I would like to document some of my reservations for the proposed methods.

1. The lack of guidance of prior specification is a fundamental drawback of the proposed approach. I advocate for Bayesian methods and am extremely pleased to see the authors propose a Bayesian approach to tackle an important practical problem in genetics. I don't feel the authors sufficiently address my concerns on "conservative" priors. In particular, I'm afraid I have to disagree that sensitivity analysis is a principled way for prior selection. The post-hoc nature of the sensitivity analysis is against the paradigm of Bayesian inference. But more importantly, a sensitivity analysis can highlight the problem of prior-dependence, but, IMHO, it does not justify selecting any particular prior value. In the authors' analysis, I think the range of the $\{p_c\}$ can be questionable in some applications. Thus, I am not convinced by the authors' argument or their cited publications, e.g., Wallace, 2020. The bottom line is that there IS a natural conservative prior, which is used in eCAVIAR. It assumes all causal sites from different traits are a priori independent. Why can't the authors use the independent prior as their default conservative prior?

2. I think the authors are right on "HyPrColoc *can* return better fine-mapping results." But their added arguments on Page 17 (line 18) feels circular. "However, by borrowing information between multiple traits, HyPrColoc outperformed eCAVIAR when fine-mapping the shared causal variant." (Should the information be borrowed after confirmation of colocalization?) Furthermore, this statement is about fine-mapping. I don't think the authors have provided a rigorous examination of the effect of "(despite) not incorporating LD information."

Despite my criticisms, I think this work is a quality of scholarship. My biggest concern is that the method itself does not provide a safeguard for false-positive findings. If not applied carefully, it can prompt the practitioners to report more colocalization findings, which are most likely false positives.

I appreciate the authors carefully respond to my previous comments. I greatly value this scientific discussion and am grateful for the authors' efforts. As I will recommend acceptance to the editor after reviewing the revised manuscript, I would like to document some of my reservations for the proposed methods.

Reply > We would like to thank the reviewer once again for their helpful comments and constructive input. We also greatly value their contribution to the scientific discussion. <

1. The lack of guidance of prior specification is a fundamental drawback of the proposed approach. I advocate for Bayesian methods and am extremely pleased to see the authors propose a Bayesian approach to tackle an important practical problem in genetics. I don't feel the authors sufficiently address my concerns on "conservative" priors. In particular, I'm afraid I have to disagree that sensitivity analysis is a principled way for prior selection. The post-hoc nature of the sensitivity analysis is against the paradigm of Bayesian inference. But more importantly, a sensitivity analysis can highlight the problem of prior-dependence, but, IMHO, it does not justify selecting any particular prior value. In the authors' analysis, I think the range of the $\{p_c\}$ can be questionable in some applications. Thus, I am not convinced by the authors' argument or their cited publications, e.g., Wallace, 2020. The bottom line is that there IS a natural conservative prior, which is used in eCAVIAR. It assumes all causal sites from different traits are a priori independent. Why can't the authors use the independent prior as their default conservative prior?

Reply 1. > We respect the reviewer's comments about the specification of prior information. However, we would like to highlight that HyPrColoc's prior sensitivity analysis tool is not employed by default. We have included the tool within the software to augment post-hoc conclusions made by investigators in the expected case that an investigator does not have a strong prior belief regarding the specification of the colocalization prior – results from the primary (e.g. default) analysis can be reported along with a statement about any observed sensitivity. The objective is to assess robustness of posterior inference (as recommended by Wallace, 2020, and by us) and not as a means for prior selection. Of course, if an investigator does have a strong prior belief then such post-hoc assessments are not necessary.

We agree that employing a prior sensitivity analysis tool as a post-hoc means to identify a 'reasonable' prior is against the Bayesian paradigm, but this is certainly not how we are advocating use of the tool.

On the point about a natural conservative prior, we would again like to highlight that the choices of default prior parameters $\{p = 10^{-4}, p_c = 0.02\}$ for use in HyPrColoc were identified using recommendations in the colocalization literature, which are derived from many real-world applications, as well as extensive simulations. Wallace, 2020, provides an in-depth discussion around the specification of the COLOC/HyPrColoc prior p , stating that " $p \sim 10^{-4}$ is supported by the convergence of...three approaches" **(i)** empirical estimation from GTEx data; **(ii)** aggregating data from over 5000 GWAS studies; and **(iii)** setting a p-value threshold and using a range of designs (case/control or quantitative trait, with varying MAF and sample size). We are therefore confident our default prior values are indeed generally robust.

In an **additional note** at the end of this document I have added a note to show that users can set values for the HyPrColoc prior parameters $\{p, p_c\}$ that allow the HyPrColoc prior formulation to match the eCAVIAR formulation. We do not recommend this. By matching the formulation, we flag an issue and go some way to answering the reviewer's question: "Why can't the authors use the independent prior as their default conservative prior?". In short, setting $p = p_c$ (the independent causal site assumption) in COLOC returned very poor performance relative to the default strategy of assuming a small dependence between causal sites. It was shown that, while the independent causal site assumption returns low type 1 error rates it leads to a massive reduction in the power/true-positive-rate, almost universally masking true colocalized associations (Wallace, 2020, Figure 4).

<

2. I think the authors are right on "HyPrColoc *can* return better fine-mapping results." But their added arguments on Page 17 (line 18) feels circular. "However, by borrowing information between multiple traits, HyPrColoc outperformed eCAVIAR when fine-mapping the shared causal variant." (Should the information be borrowed after confirmation of colocalization?) Furthermore, this statement is about fine-mapping. I don't think the authors have provided a rigorous examination of the effect of "(despite) not incorporating LD information."

Reply 2. > On the reviewer's question "Should the information be borrowed after confirmation of colocalization?"

We apologise for any confusion. Information is 'borrowed' when colocalized traits are present in the sample, otherwise there is no "common aetiological factors" or "concurrency of patterns" (Otiende et al; PLOS ONE, 2020) shared between the traits which can strengthen inference. In the HyPrColoc model, the posterior probability of colocalization is computed by aggregating fine-mapping probabilities for each SNP. The upshot is that, any information borrowed between traits to strengthen inference about colocalization is information simultaneously borrowed to strengthen fine-mapping inference (and vice versa).

The reviewer rightly highlights that we do not know in advance which (if any) of the traits are colocalized. This is not a drawback. We have shown that when colocalized traits are present, HyPrColoc's joint multi-trait colocalization model returns improved performance relative to the alternative of performing multiple rounds of pairwise analysis (these models cannot detect "concurrency of patterns" across more than two traits). In general, the divisive clustering algorithm places no condition on having advance knowledge of any colocalized traits in the sample, nor does it place a requirement that colocalized traits need to be present in the sample.

One approach to boost the possibility of borrowing information is to select traits in a 'supervised' way, based on the prior assumption that colocalization between (subsets of) traits is biologically plausible, e.g. by analysing families of related traits. A striking example, which highlights the benefit of such an approach, is seen in Supplementary Figure S13

where the sub-threshold CHD signal is boosted by being jointly analysed with three colocalized lipid traits. <

3. Despite my criticisms, I think this work is a quality of scholarship. My biggest concern is that the method itself does not provide a safeguard for false-positive findings. If not applied carefully, it can prompt the practitioners to report more colocalization findings, which are most likely false positives.

Reply 3. > We very much appreciate the reviewer's kind comment. We agree that practitioners should apply the method carefully and employ good scientific principles before drawing strong conclusions, such as attempting to externally validate results where possible. <

Additional Note

Here we show that users can specify values for the HyPrColoc prior parameters $\{p, p_c\}$ that allow the HyPrColoc prior formulation to match the eCAVIAR formulation.

The eCAVIAR prior configuration probabilities are computed by specifying a single prior parameter, which we denote by γ_{cav} . In order to map the two-parameter prior setup of HyPrColoc to the single parameter model of eCAVIAR we can apply the following formula (under the default analysis which approximates each posterior using $2m + 1$ hypotheses):

$$p_c = 1 - (1 - p)^{\frac{1}{m-1}},$$

where p_c is the colocalization prior, p is the prior probability that a SNP is causal in the region and m denotes the number of traits under investigation. The HyPrColoc prior framework is now constrained to be a one parameter model in which the prior probability of a causal configuration in the colocalization set H_m is the same as the prior probability of a causal configuration in the non-colocalization set $H_{(m-1;1)}$ – which approximates the eCAVIAR prior framework in which causal sites from different traits are a priori independent. Finally, setting

$$p = \frac{\gamma_{cav}}{1 - \gamma_{cav}} \quad (1)$$

matches the prior configuration probabilities between the methods, i.e. using the above formulae the prior probability of a causal configuration in which each trait has at most 1 causal variant, divided by the prior probability of the 'null' configuration (note that, the terms in the COLOC/HyPrColoc model are presented in w.r.t. Bayes factors), is identical between methods.

On the reviewer's point: "why can't the authors use the independent prior as their default conservative prior?" We summarise some reasons for not doing this in light of the above.

The default eCAVIAR setting is $\gamma_{cav} = 0.01$, whereas in COLOC/HyPrColoc $p = 10^{-4}$. Using the matching formula (equation 1) above, we reveal that the COLOC/HyPrColoc prior is *two* orders of magnitude smaller than eCAVIAR. It is clear from Wallace's results (Wallace, Plos Gen, 2020) that a value of $p = \frac{\gamma_{cav}}{1-\gamma_{cav}} = \frac{0.01}{0.99} \approx 10^{-2}$ is not supported in practice / not appropriate for use in COLOC/HyPrColoc – however the eCAVIAR authors demonstrated that setting $\gamma_{cav} = 0.01$ is reasonable. We can alternatively ask: will eCAVIAR maintain performance when $\gamma_{cav} \approx 10^{-4}$? Reconciling why there is such a large discrepancy between the default specifications of these parameters is interesting and is worthy of a future investigation.

I suspect the most important line of investigation is not whether we can match prior strategies, as above, but rather, to focus efforts on quantifying differences between the modelling strategies (used to compute the likelihood) in order to decide whether matching the prior formulations is practical. The main difference between the methods is that eCAVIAR models LD between the summary data and COLOC does not. Accounting for LD can help attenuate issues associated with computing 'large' posterior probabilities for configurations which include variants in strong LD with a true causal variant – in fact this is likely the reason why setting $\gamma_{cav} = 0.01$ (a large value relative to COLOC's default p) is reasonable.

Under the single causal variant assumption – which does not account for LD – causal configurations which visit variants in strong LD with a true causal variant can have the effect of masking a true colocalization signal (as the posterior probabilities are similar in magnitude). An appropriate choice of model for the causal configuration prior probabilities can help attenuate the masking of the colocalization signal. The COLOC model does this (as the reviewer notes) by allowing for a small dependence between causal sites from different traits – which necessarily means setting the marginal probability p smaller than the colocalization probability p_c . In light of this, a great deal of time and effort has gone into identifying reasonable choices for these parameters by both Wallace and the HyPrColoc authors. Setting $p = p_c$ (the independent causal site assumption) leads to a massive reduction in the true positive rate (Wallace, 2020, Figure 4) almost universally masking true colocalization associations, (modest) increasing of p_c relative to p led to much improved overall performance.